# Will rivers become more intermittent in France? Learning from an extended set of hydrological projections

Tristan Jaouen[1], Lionel Benoit[2], Louis Héraut[1], Eric Sauquet[1]

[1]INRAE Lyon-Grenoble Auvergne-Rhône-Alpes, RiverLy, Villeurbanne, France
[2]INRAE PACA, Biostatistique et processus SPatiaux, Avignon, France

*Correspondence to*: Tristan Jaouen (trjaouen@laposte.net) and Eric Sauquet (eric.sauquet@inrae.fr)

**Abstract**

This study aims to assess the changes in the intermittence of river flows across France in the context of climate change. Projections of flow intermittence are derived from the results of the Explore2 project, which is the latest national study that
proposes a wide range of potential hydrological futures for the 21st century. The multi-model approach developed within the Explore2 project enables to characterize uncertainties in future flow intermittence. Combined with discrete observations of flow states, hydrological projections are post-processed to compute the daily probability of flow intermittence (*PFI*) on each element of the partition of France in Hydro-EcoRegions (HER2).

The post-processing consists in calibrating logistic regressions between the historical flow states of the National Low-Flow
Observatory (ONDE) network and the flow data simulated by the hydrological models involved in Explore2 run with the SAFRAN atmospheric reanalysis as inputs. After calibration, these regressions are used to project daily *PFIs* for the entire 21st century, based on flow simulations from five hydrological models driven by up to 17 climate projections under RCP 2.6, 4.5, and 8.5 climate change scenarios.

The results show good agreement among the hydrological models regarding the increase in flow intermittence under RCP 4.5
and 8.5. The projected increase in mean daily *PFI* between July and October as well as the shift of the first and last days when *PFI* exceeds 20% both suggest a gradual intensification and extension of dry spells throughout the century. The southern regions of France are likely to experience greater increases in runoff intermittence than the northern regions, and mountainous regions such as the Alps and the Pyrenees are likely to experience changes in their dynamics of intermittence with a reduction of winter intermittence and the apparition or increase of summer intermittence. The uncertainty of these projected changes is
larger in northern France due to greater intermodel variability in this region.

## 1 Introduction

Intermittent Rivers and Ephemeral Streams (IRES) are watercourses characterized by the absence of continuous year-round water flow (Sefton et al., 2019). While water management practices can exacerbate flow intermittence by altering runoff

patterns, climatic factors - particularly aridity - remain the primary determinants of these systems (Addor et al., 2018; Hammond and Fleming, 2021; Sauquet et al., 2021; Zipper et al., 2021). In recent decades, climate change has intensified these dynamics, significantly impacting the global water cycle: summer precipitations tend to decrease, and rising temperatures have increased evapotranspiration and driven more frequent heatwaves (Douville et al., 2021). These changes have led to more severe low-flow periods and more frequent and widespread dry spells in Western Europe (Delso et al., 2017; Tramblay et al., 2020; Vicente-Serrano et al., 2020). Meanwhile, IRES play a critical role as interfaces between terrestrial and aquatic ecosystems. Altered patterns of flow interruption can therefore compromise water quantity and quality in downstream perennial rivers, increase extinction risks for specialized species, and disrupt organic matter, nutrient, and sediment cycles (Bertassello et al., 2022; Gallart et al., 2012; Giezendanner et al., 2021; Finn et al., 2011; Meyer et al., 2007; Sarremejane et al., 2017; Larned et al., 2010). Hence, a better understanding of IRES behaviours and their interaction with perennial streams is needed to assess the impacts of ongoing and future changes on river ecosystems (Döll and Schmied, 2012; Jaeger et al., 2014; Pumo et al., 2016; Leigh and Datry, 2017), and to address water management challenges (Acuña et al., 2014).

Globally, IRES account for nearly two-thirds of river networks (Schneider et al., 2017; Messager et al., 2021), with this proportion reaching 39% in France (Snelder et al., 2013). Their spatial and temporal dynamics can be characterized using conceptual models of network contraction (Shaw et al., 2017; Botter and Durighetto, 2020) or through semi-conceptual and machine learning frameworks that integrate hydroclimatic and physiographic factors (Lapides et al., 2021; Durighetto et al., 2022). The strongest predictors of flow intermittence are aridity indices (Vicente-Serrano et al., 2019 ; Jaeger et al., 2019). However, local variations in permeability, lithology and topography also influence intermittence patterns in specific regions, including West Africa (Yu et al., 2018; Belemtougri et al., 2021), northern Spain (González-Ferreras and Barquín, 2017) and northwestern Australia (Bourke et al., 2021). Studying intermittence at a large scale therefore requires considering the diversity of hydrological processes at play, particularly when multiple watersheds exhibit diverse sensitivities to hydroclimatic conditions. This variability highlights the need for robust methodologies to assess intermittence at national to (semi-)continental scales (Addor et al., 2018; Hammond and Fleming, 2021; Sando et al., 2022; Döll et al., 2024).

Understanding the drivers of river intermittence is also crucial for analysing historical trends and projecting future changes in IRES behaviour. Climate change has already expanded the spatial and temporal extent of intermittent flows worldwide (Zipper et al., 2021; Jaeger et al., 2019; Sando et al., 2022; Gudmundsson et al., 2021), and the Mediterranean basin is particularly affected (Tramblay et al., 2020; De Girolamo et al., 2022). However, interpreting climate projections and emission scenarios at the scale of headwater basins remains rare (Schneider et al., 2013). Indeed, many climate models have a coarse spatial resolution and high uncertainties emerge from multi-model ensembles (MMEs), which makes accurate predictions of future IRES changes difficult.

To address these challenges, France has implemented the Explore2 project (https://entrepot.recherche.data.gouv.fr/dataverse/explore2; Sauquet et al., in prep.) that uses a multi-model, multi-scenario approach to simulate hydrological conditions throughout for the 21st century. It assesses uncertainties at each step of the modelling process. However, the resolution of the climate projections involved in Explore2 is too coarse to accurately capture IRES dynamics. Hence, this study extends the results of Explore2 for purposes of generating flow intermittence projections for French headwater streams. To this end, we follow a statistical approach that estimates the daily Probability of Flow Intermittence (*PFI*) in small streams based on streamflow data simulated by Explore2 for larger perennial rivers. The estimated *PFI* serves as a proxy indicator for the intensity and duration of dry periods. The method is calibrated using field observations carried out regularly at more than 3200 upstream river sites prone to drying.

To achieve this, the present work builds on two key studies: Beaufort et al. (2018), which demonstrated consistent performance of *PFI* estimation across the heterogeneous climate of France using daily discharge, and Sauquet et al. (2021), which projected *PFI* under a limited set of climate projections with coarse spatial resolution. In contrast, this study incorporates a finer spatial resolution, a broader range of climate scenarios, and a wider variety of hydrological models thanks to the integration of Explore2 projections. It therefore addresses a critical research gap by generating projections of small stream intermittence across France and throughout the 21st century while quantifying the associated uncertainties.

The remainder of the paper is organized into five sections. The data used are outlined in the second section. Section 3 describes the statistical method linking *PFI* to hydrological projections. Section 4 presents the results, which focus on the mean daily *PFI* between July and October ($mPFI_{7-10}$) and the median of the first and last days (respectively, *Tf* and *Tl*) when *PFI* exceeds 20%. Finally, Section 5 discusses these results and Section 6 concludes the study.

## 2 Data

### 2.1 Monitoring river flow intermittence

The French Biodiversity Office (OFB, https://ofb.gouv.fr/) initiated in 2012 the National Low-Flow Observatory (ONDE) to gain a better understanding of low flows and intermittent rivers. ONDE is a stream intermittence monitoring network comprising 3248 observation sites strategically distributed across the French river network with the aim of characterizing the occurrence and the intensity of summer (low-) flows (Nowak and Durozoi, 2012). The network has been designed to focus on streams with a Strahler order ranging from 1 to 4, which are prone to natural and/or anthropogenic intermittent flow conditions. Most sites (85%) are located on "small streams" with a drainage area ≤ 100 km$^2$, 75% are situated on streams with a Strahler index of 1 or 2, and 20% of the sites have a drainage area ≤ 10 km$^2$ (Appendix Fig. G1 and G3).

A key feature of the ONDE network is that it involves the flow state of watercourses, which results from the visual inspection
of streams by OFB observers at each location, instead of a sensor-based measurement of flow rates. In the present study, two
binary categories are defined for ONDE observations: (1) "visible flows" and (2) "dry conditions", which gather non-visible
flows (standing water in isolated pools) and totally dry conditions (dry riverbed at or near the ONDE site). We use only data
from regular and structured field campaigns (referred to as "usual campaigns" in the ONDE nomenclature), during which the
ONDE sites are monitored systematically in mainland France around the 25th day of each month from May to September.
Here we use ONDE data from 2012 to 2022 for model calibration and evaluation.

Based on ONDE data, the *PFI* is defined as the proportion of small streams under drying conditions at the regional scale. It is
used as a proxy indicator for the intensity and duration of dry periods. Long-term analysis of the *PFI* characterises the spatial
and temporal trends of drought across regions and provides valuable information about IRES intermittence behaviour.

## 2.2 Delineating areas with homogeneous hydrological behaviour: the Hydro-EcoRegions

Hydro-EcoRegions (HER) are spatially homogeneous areas defined over France based on natural drivers involved in river
ecosystem functioning, such as geology, topography, and climate. There are 85 level-2 HER (HER2) across France, and they
are derived from the sub-division of 22 level-1 HER (HER1) (Wasson et al., 2002), which were used in the previous study
about the impact of climate change on *PFI* in France (Sauquet et al., 2021). In contrast, the present study investigates flow
intermittence at the scale of HER2 regions in order to model *PFI* at a higher spatial resolution. The statistical approach
developed hereafter requires sufficient observations of flow states for calibration purposes (see Sect. 3.1), which led us to
perform five groupings of HER2 in order to ensure that enough observations are available for model calibration. When merging
HER2 regions we checked that they belonged to the same level-1 HER to ensure that they share similar environmental
characteristics at the large scale. In the end, a partition of France into 75 entities (HER2 or pool of HER2) was considered for
this study (Fig. 1).

These 75 HER2s have a median surface area of 4990 km² and more than 20% (17/75) have a surface area greater than 10000
km². The HER2s were the subject of ONDE field campaigns from May to September between 2012 and 2022, thus resulting
in 4125 campaigns. A total of 78 campaigns are excluded from the analysis due to their failure to cover more than 75% of the
monitoring stations within a given HER2. Thus, the data exclusion rate from the ONDE network is less than 2%, leaving 4047
field campaigns with usable data.

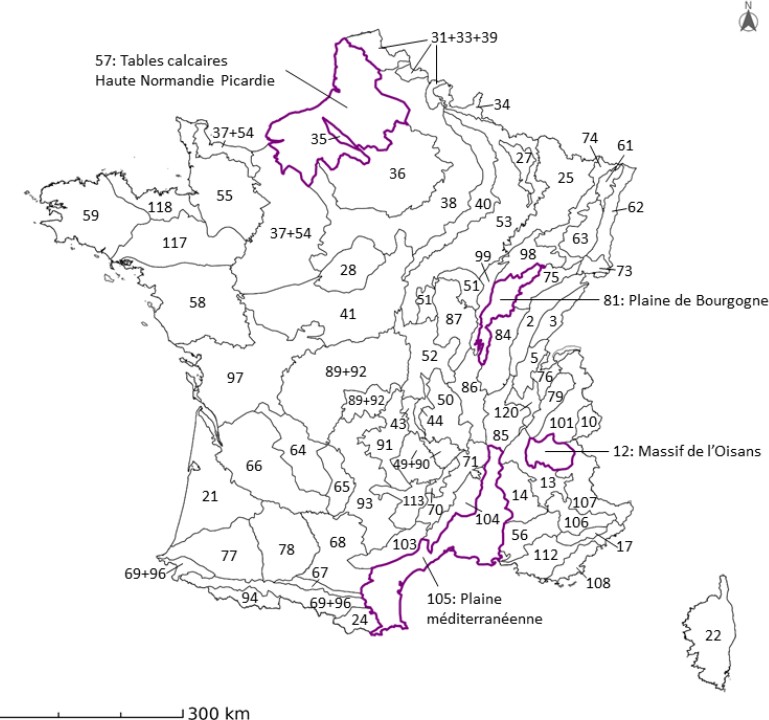

**Figure 1: Delineation of level 2 Hydro-EcoRegions (HER2) across France. The four HER2 that are used for illustration in this study are highlighted in purple (see Sect. 3.7).**

## 2.3 Future hydrological projections over France: the Explore2 project

The assessment of changes in the *PFI* over the 21$^{st}$ century is based on the large set of hydrological projections produced by the Explore2 project (Sauquet et al., in prep). The daily discharge projections were obtained at about 4000 simulation points distributed along the French river network, with a constraint on the minimum drainage area imposed by the spatial resolution of climate projections (64 km²) and another constraint of an even coverage across France.

Explore2 is a multi-model simulation experiment involving nine different hydrological models, but some of them are restricted to regional applications (on average each simulation point has discharge projections simulated by four hydrological models). Finally, with the objective of predicting flow intermittence at the national scale, we select the five models with the largest simulation domain: CTRIP (Decharme et al., 2019), GRSD (De Lavenne et al., 2016), J2000 (Morel et al., 2023), ORCHIDEE (Huang et al., 2024), and SMASH (Jay-Allemand et al., 2020). The Explore2 project considers only natural flows, although some of the hydrological models involved in the project have the capability of including human-induced influences. The evaluation of hydrological models was performed using an extended dataset of near-pristine catchments (Strohmenger et al., 2023). The simulations used for model evaluation were carried out for the period 1976-2022 using the French near-surface SAFRAN meteorological reanalysis (Vidal et al. 2010) as input. In addition, the hydrological models included in Explore2

have been evaluated on the common period of availability 1976-2019 (Sauquet et al., in prep). After validation, the hydrological models are forced by an ensemble dataset derived from 17 pairs of Global and Regional Climate Models (GCM-RCM) corrected by the statistical adjustment method ADAMONT (Verfaillie et al., 2018) for both historical (1976-2004) and future climate scenarios RCP 2.6, 4.5 and 8.5 (2005-2100; Appendix Sect. A). In the end, these bias-corrected climate projections were used as inputs for the hydrological models: all 17 GCM-RCM projections are used under RCP8.5 scenario, whereas 10 of these projections are used under RCP2.6 and 9 under RCP4.5. The baseline period is set to 1976-2005 and used thereafter for analysing changes in IRES behaviour. Overall, the hydrological projections used thereafter to predict *PFI* result from combinations RCP-GCM-RCM-Hydrological Model.

## 3 Modelling framework

### 3.1 Linking intermittence to stream discharge at the HER2 scale

The empirical method suggested by Beaufort et al. (2018; Eq. (4)) is applied here to estimate the Probability of Flow Intermittence (*PFI*) at the scale of HER2 regions. This method follows a two-step process that links observed flow states at ONDE sites to daily streamflows (either gauged or modelled) using logistic regressions.

First, observations from ONDE sites are used to determine the percentage of sites with "dry conditions" for each HER2 ($h$) and each campaign date ($j$). Flow intermittence evolves over relatively long time scales, with seasonal transitions between flowing and dry states occurring gradually within the year, except in cases of significant rainfall events that can temporarily trigger flow conditions. Given this inertia, the estimation of flow intermittence remains appropriate even if not all sites within a HER2 are monitored on the exact same day, with a shift of only a few days during a given campaign. To further ensure the temporal consistency across campaigns the monitoring is systematically conducted at fixed sites, and data from a given ONDE campaign and HER2 are only considered if at least 75% of the ONDE sites within the HER2 are monitored during the campaign. This approach minimizes potential biases associated with incomplete observations, and ensures that the percentage of dry ONDE sites referred hereafter as $PFI_h(j)$ can be considered representative of the *PFI*.

Subsequently, a logistic regression model is calibrated for each HER2 in order to link the *PFI* values for day $j$ to streamflow conditions (Eq. (1), Fig. 2):

$$PFI_h(j) = \frac{e^{\beta_{0 \cdot HER2_h} + \beta_{1 \cdot HER2_h} \times F_{Q \cdot HER2_h}(j)}}{1 + e^{\beta_{0 \cdot HER2_h} + \beta_{1 \cdot HER2_h} \times F_{Q \cdot HER2_h}(j)}}, \tag{1}$$

where $\beta_{0 \cdot HER2_h}$ and $\beta_{1 \cdot HER2_h}$ are respectively the logistic regression intercept and slope coefficient associated with the predictor $F_{Q \cdot HER2_h}$.

The model uses the mean non-exceedance frequencies of discharge $F_{Q \cdot HER2_h}(j)$ as predictor for $PFI_h(j)$ (Eq. (2)). $F_{Q.HER2h}$ is regarded as a proxy characterising the current wet versus dry hydrological conditions at the regional scale. It is derived from the non-exceedance frequency curves of streams, which describe the probability that a given discharge value is not exceeded. For each specific ($j$) corresponding to an ONDE field campaign and each HER2 ($h$), the daily empirical non-exceedance

frequencies of discharge are spatially averaged over all available $n$ streams whose drainage areas intersect the HER2 of interest (weighted by their drainage area) and temporally averaged over the period $[j-6;j]$. This seven-day window allows the inclusion of non-simultaneous response times caused by heterogeneous propagation times in the underground and the river network (the choice for a seven-day window is the result of an optimisation process, see Appendix Sect. B). In the end, $F_{Q \cdot HER2_h}$ is defined by:

$$F_{Q \cdot HER2_h}(j) = \frac{1}{7 \times n} \sum_{k=j-6}^{j} \sum_{s=1}^{n} F_Q(k,s) \tag{2}$$

where $F_Q(k,s)$ represents the non-exceedance frequency of each stream $s$ among the $n$ streams whose drainage areas intersect the HER2 $h$. This frequency is weighted by the drainage area of stream $s$. The summation is computed over each day $k$ within the $[j-6;j]$ period around campaign $j$.

Once calibrated, the logistic regressions are used to convert the hydrological projections from the Explore2 project into *PFI* projections.

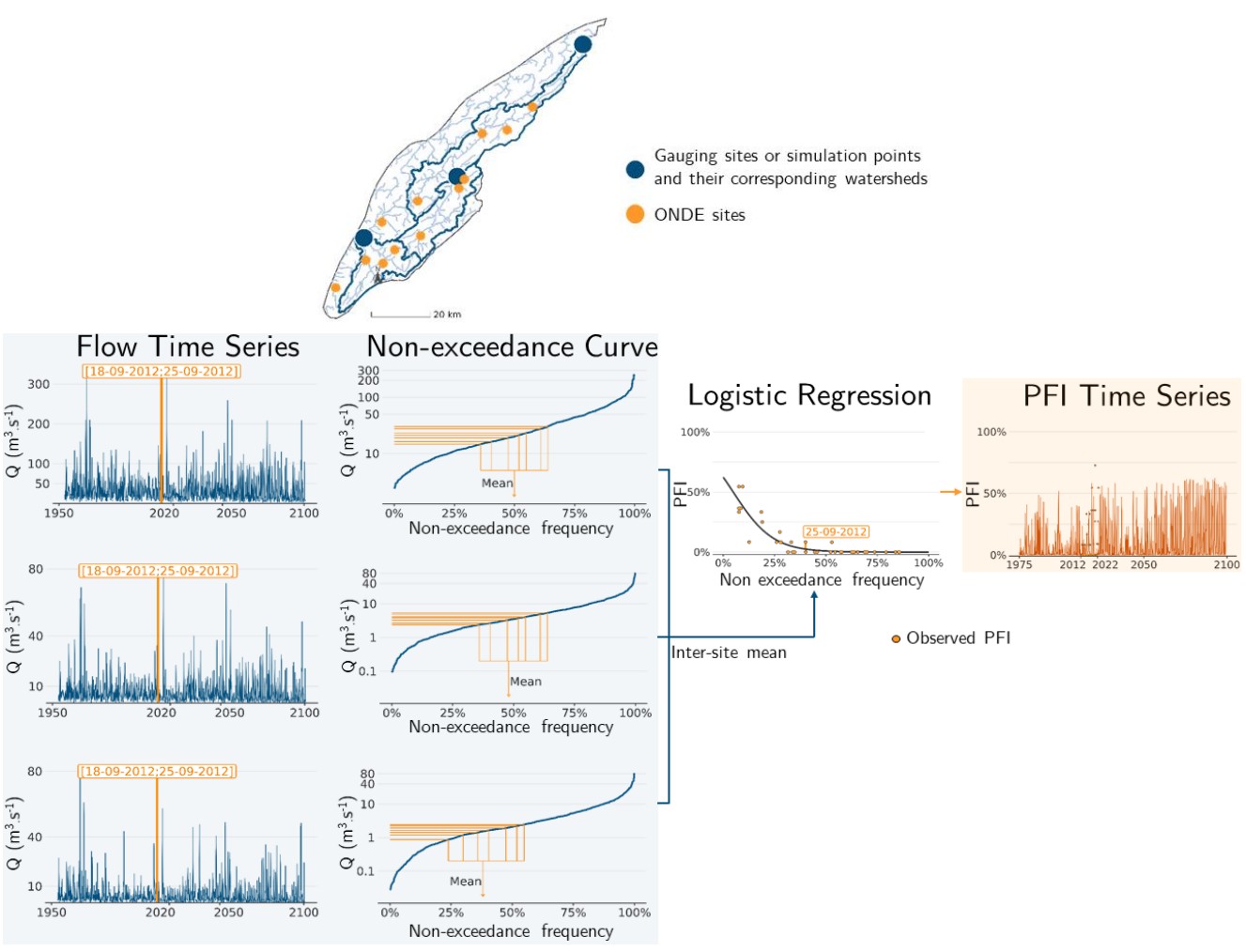

**Figure 2: Schematic view of the approach adapted to derive the regional Probability of Flow Intermittence (*PFI*) from ONDE sites using daily flows either gauged or modelled within a given HER2. The daily flows simulated by Explore2 at three simulation points generate flow time series (column 1) and their corresponding non-exceedance frequency curves (column 2). A logistic regression is then used to link daily flows to ONDE observations, as illustrated here for the campaign of September 25, 2012. For this campaign, daily discharge values simulated between September 18 and September 25, 2012, are extracted for each simulation point whose drainage area intersects the HER2. These seven flow values are matched to their respective non-exceedance frequency using the non-exceedance frequency curve. The mean non-exceedance frequency across all simulation points is then associated with the proportion of ONDE sites exhibiting "dry conditions" during this campaign (observed *PFI*, column 3). The logistic regression is calibrated using data from all ONDE campaigns. Once calibrated, these models allow for the conversion of daily discharge time series into daily *PFI* time series for the HER2 (column 4).**

### 3.2 Calibrating *PFI* logistic regression models using observed daily discharge data

Logistic regression models are preliminarily fitted between 2012 and 2022 using observed discharge data. The daily discharge data used in the calculation of $F_{Q.HER2h}$ were compiled using gauging station records from the French reference hydrologic network HydroPortail (Leleu et al., 2014) (www.hydro.eaufrance.fr). The set of gauging stations were selected on the basis of three different criteria:

- Daily discharge data must be available during the ONDE field campaigns, i.e., within the [j-10;j+10] interval centred on 25th of each month from May to September, for the years 2012 to 2022.
- The human disturbance to flow must be minimal at the gauging stations.
- All selected gauging stations must have a drainage area of less than 2000 km². Large gauged basins with a high Strahler order have been discarded since they are unlikely to behave in the same way as small streams.

This selection process results in a final dataset including 1008 stations (Fig. 3). The non-exceedance frequency curve of each gauging station is computed using its entire available discharge time series, leading to variations in the time periods covered by the discharge data across stations. Nevertheless, limiting the analysis to the period 2012-2022 would likely exclude critical information on hydrological extremes and fail to fully capture the variability of high and low flow events, which are pivotal for understanding a river's behaviour during extreme conditions.

### 3.3 Assessing the performance of *PFI* logistic regression models under current climate

Several cross-validation schemes are then considered to test the robustness of these models. A leave-one-year-out cross-validation is first carried out by excluding one year at a time from the training dataset, learning from the data of the remaining years and then making estimations for the excluded year. The robustness under different climate conditions is also assessed through differential split-sample test (DSST; Dakhlaoui et al., 2017), which is a k-fold cross-validation performed on three distinct groups of hydrological years categorized as dry, intermediate, or wet (Appendix Sect. C). The years are discriminated according to the aridity index (Barrow, 1992), which is one on the main drivers of flow intermittence at the global scale (Sauquet et al., 2021).

The model performance is assessed using several skill scores: the bias (to detect underestimation or overestimation), the mean absolute error (MAE, to quantify the magnitude of prediction errors), and the root-mean-square error (RMSE, which provides

a quadratic estimation that assigns relatively higher weight to large errors). Additionally, the Nash-Sutcliffe model efficiency
coefficient (NSE) is computed to compare the variance of the estimation error to the variance of the observed time series (Nash
and Sutcliffe, 1970). The Kling-Gupta Efficiency (KGE) complements the NSE metric as KGE is the Euclidean distance
between observed and estimated *PFI*, computed using the coordinates of bias, standard deviation, and correlation (Gupta et
al., 2009). The Leave One Year Out analysis results are obtained by averaging the validation metrics computed for each year.

### 3.4 Selecting simulation points for *PFI* projections under climate change

The modelling framework suggested by (Beaufort et al., 2018; Sauquet et al., 2021) was applied to the 75 HER2s under
regional climate projections to assess the potential impact of climate change on flow intermittence dynamics.
Discharge data derived from the Explore2 database are used as input for the logistic regression models introduced in Sect. 3.1.
However, not all the stations in the HydroPortail database have been simulated by the hydrological models. To overcome this
co-location problem, the choice is made to identify the simulation points closest to the 1008 gauging stations selected for the
diagnosis in Sect. 3.2 (Fig. 3). The conditions of the application experiment are thus similar to the ones of the diagnostic
experiment. A constraint is applied to ensure that no simulation point is assigned multiple times, unless there is no alternative
point within a 50 km distance from the gauging station (Appendix Sect. D). This selection is made independently for each
hydrological model. Only 483 points are assigned for the J2000 model because this model simulates streamflow only for the
Loire and Rhône river basins.

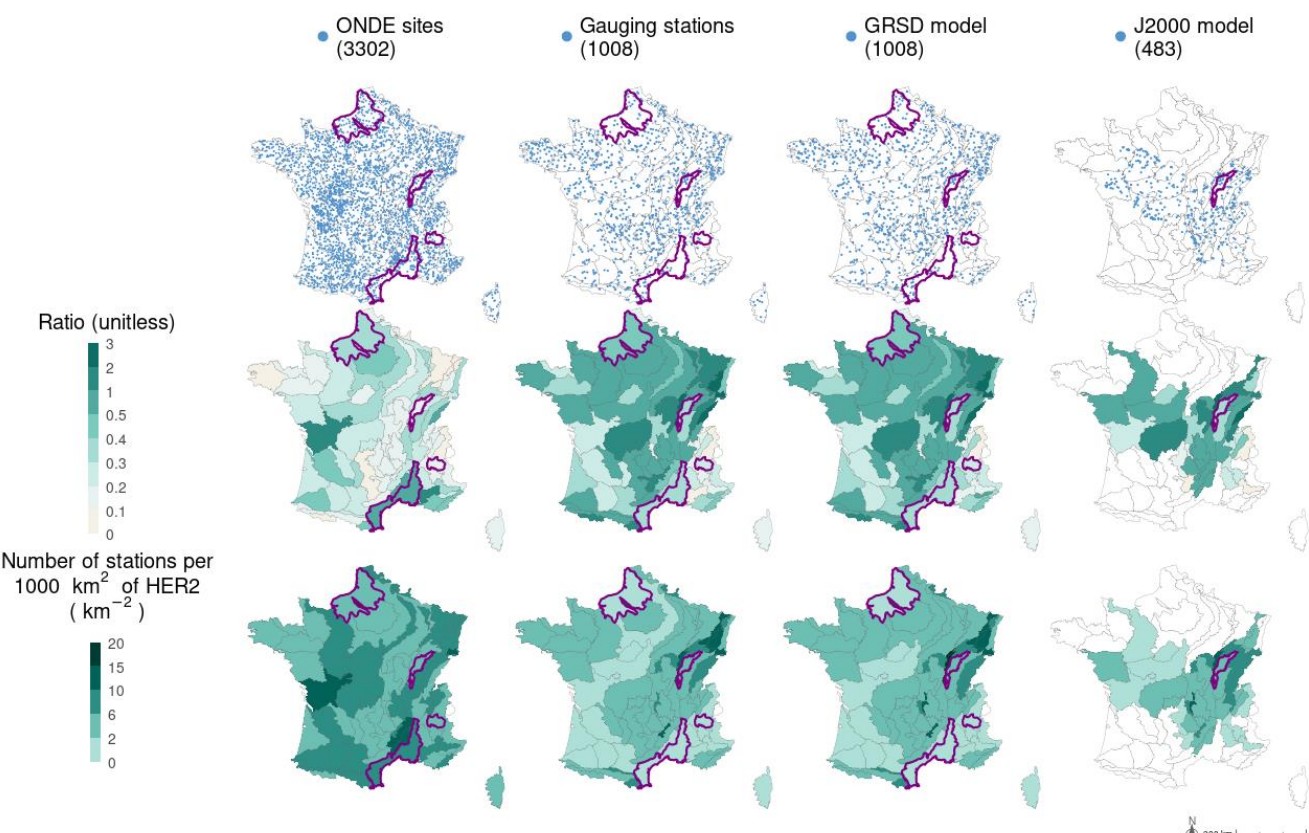

**Figure 3: Location of the 3302 ONDE sites, the 1008 gauging stations selected from the HydroPortail database, and the Explore2 simulation points selected for GRSD and J2000 hydrological model (line 1). The maps on the second line show the ratio between the sum of the catchment areas defined by the gauging stations or the simulation points and the surface area of the HER2 they intersect (line 2). These catchment areas may overlap and their sums are computed without considering their spatial distribution. The maps**

**on the third line show the density of gauging stations or simulation points (line 3). The same results are available for the other hydrological models in Appendix Sect. E.**

### 3.5 Constructing synthetic non-exceedance frequency curves for *PFI* projections

In our framework, non-exceedance frequency curves must be computed for each simulation point prior to *PFI* estimation. To avoid problems of extrapolating non-exceedance frequency of discharge values outside the range of historical flows when

performing *PFI* projection under changing climate, we decided to build synthetic flow time series with a period length of 169 years that combine the historical discharge data simulated with SAFRAN over the period 1976-2022 with discharge data simulated by the different modelling chains RCP-GCM-RCM-Hydrological Model over the period 1976-2100. This concatenation is made independently for each modelling chain and simulation point. As a consequence, while the assessment of the performance of *PFI* logistic regression models under current climate is conducted using observed discharge data that

can be unbalanced (i.e., covering different observation periods) due to heterogeneous data availability across stations (see Sect. 3.3), the calibration used for projecting *PFI* under future climate scenarios relies on standardized and balanced synthetic time series which ensures the consistency of the modelling across simulation points and modelling chains.

Finally, each synthetic discharge time series generated by a specific RCP-GCM-RCM-Hydrological Model chain results in a non-exceedance frequency curve which is used thereafter to calibrate a logistic regression (Eq. (1)).

## 3.6 Recalibrating logistic regressions and projecting *PFI* under climate scenarios

Before being applied to future climate conditions, the logistic regression model of Eq. (1) needs to be recalibrated with past discharges simulated by each hydrological model (here discharges simulated by the different hydrological models using SAFRAN reanalysis data available during the period 2012-2022 are used). The resulting logistic regressions are then re-evaluated for each hydrological model using the median of the bias, MAE, RMSE, NSE, and KGE skill scores over all RCP-GCM-RCM modelling chains.

Finally, after re-calibration, these logistic regressions are used to project *PFI* based on discharge data simulated for both the historical period (1976-2004) and the projection period (2005-2100) under various climate scenarios (RCP 2.6, 4.5 and 8.5).

## 3.7 Assessing changes in *PFI* and associated uncertainties

We compare *PFI* between a reference period (1976-2005), and medium-term (2041-2070) as well as long-term (2070-2099) horizons. Results are presented at the HER2 scale across France and discussed in more details for four HER2s (Fig. 1), deliberately selected to represent contrasting hydro-climatic conditions and behaviours: "Haute Normandie Picardie" (HER2-57, North, oceanic temperate climate), "Plaine de Bourgogne" (HER2-81, Northeast, continental temperate climate), "Massif de l'Oisans" (HER2-12, Alps, mountainous climate), and "Plaine méditerranéenne" (HER2-105, Southeast, Mediterranean climate). In the rest of the article, "dry periods" of HER2s are defined as the period of the year when *PFI* is greater than 20%. For each HER2, the changes in intensity and seasonality of flow intermittence for 2041-2070 and 2070-2099 relative to 1976-2005, are characterized using the mean daily *PFI* between July and October ($mPFI_{7-10}$), and the median of the first and last days (respectively, *Tf* and *Tl*) of the year with *PFI* exceeding 20%.

The analysis of the uncertainty propagation in the modelling framework is examined using the QUALYPSO method, applied to the national average of $mPFI_{7-10}$ ($mPFI_{7-10}[France]$), weighted by HER2 areas and obtained under RCP 2.6, 4.5 and 8.5 climate change scenarios (Evin et al., 2019; Evin et al., 2021). The QUALYPSO method is used to characterise the changes in $mPFI_{7-10}[France]$ by estimating its ensemble mean over all the projections, as well as the total uncertainty associated with this set of projections, the contribution of the various sources of uncertainty (RCP, GCM, RCM, hydrological models, residual and internal climate variability) to the total uncertainty and the main effect of each individual model and its contribution to the total uncertainty. The contribution of each component to the total uncertainty is estimated as its fixed effect in a linear regression model, which adjusts for imbalances in the number of simulations across components within modelling chains (Appendix Sect. F).

A multi-model index of agreement (*MIA*) is also computed on $mPFI_{7-10}$ time series to highlight convergence in changes over the projections (Tramblay and Somot, 2018):

$$MIA = \frac{1}{n}\sum_{k=1}^{n} i_k,$$ (3)

where $n$ is the number of projections, $i_k = 1$ for a significant positive trend of the $mPFI_{7-10}$ according to the Mann-Kendall test (with $\alpha = 0.1$), $i_k = -1$ for a significant negative trend and $i_k = 0$ for no significant trend.

# 4 Results

## 4.1 Data pre-processing

The 3302 ONDE sites are located in small streams while gauging stations in France are monitoring medium size catchments (Van Meerveld et al., 2020). In addition, all Explore2 simulation sites have a drainage area larger than 64 km². Unsurprisingly, there is little overlap between the distribution of areas drained by ONDE sites (median: 24 km²; quartiles Q1-Q3: 12-50) and those drained by the two sets of 1008 gauging stations (median: 173 km²; Q1-Q3: 85-396) and 1008 simulation points (median: 178 km²; Q1-Q3: 95-400; Appendix Fig. G1). ONDE sites, gauging stations and simulation points are located at similar elevations (overall median: 168 m; Q1-Q3: 75-308; Appendix Fig. G2). The drainage areas of ONDE sites, gauging stations and simulation points represent respectively a median coverage of 21% (Q1-Q3: 15-35), 54% (Q1-Q3: 31-94) and 55% (Q1-Q3: 32-96) of the HER2 areas. This corresponds to a median number of 6.1 sites (Q1-Q3: 5-8), 3.6 stations (Q1-Q3: 1.9-5.7) and 3.7 simulation points (Q1-Q3: 2-6) per 1000 km² of HER2 (Fig. 3). These statistics highlight the question of representativeness of the ONDE sites despite the even coverage of the ONDE network.

## 4.2 Model performance

The logistic regressions calibrated using observed flows accurately predict *PFI* under current climate conditions (Fig. 4; Appendix Fig. H1). The median performance across HER2s is, for example, 0.85 for KGE (Q1-Q3: 0.77-0.89) and 0.79 for NSE (Q1-Q3: 0.64-0.84) when the leave-one-year-out cross-validation is considered. The KGE skill score exceeds 80% in 50 out of 75 HER2 and the bias is very low. The best performances are observed in sedimentary plains and in Aquitaine while lower performances are obtained in mountainous areas such as the Alps, the Massif Central and the Pyrenees. The mean absolute error and RMSE show that the largest absolute deviations between observations and predictions occur in the southeastern part of France. This result is consistent with previous findings (e.g. Fig. 3 in Sauquet et al., 2021): (1) the performance of the logistic regressions is partly correlated with the level of intermittence, and (2) no-flow conditions which mainly occur in winter due to freezing in the Alps are not captured by the ONDE network. Observed values of $mPFI_{7-10}$ range between 3 and 37% (median: 14.3%) across France while values of $mPFI_{7-10}$ obtained with SAFRAN range between 2 and 37 % (median: 14.4%). The modelling approach is able to reproduce the spatial pattern of flow intermittence (Fig. 4), although the results obtained with the DSST show less satisfactory performance (Appendix Fig. H2 and Table H1). The two skill scores

KGE and NSE are sensitive to wet, intermediate and dry conditions while the other skill scores show no major change. The

315 values of RMSE and MAE skill scores do not increase much when the calibration dataset is stratified based on climate

conditions, which indicates that the proposed model is quite robust to climate fluctuations. The bias, however, indicates an

underestimation of *PFI* in the southwest and in the north during dry years. This lower performance is partly due to a reduced

number of years used for calibration purposes, and to the difficulty to extrapolate the logistic regression equations under

unobserved climate conditions.

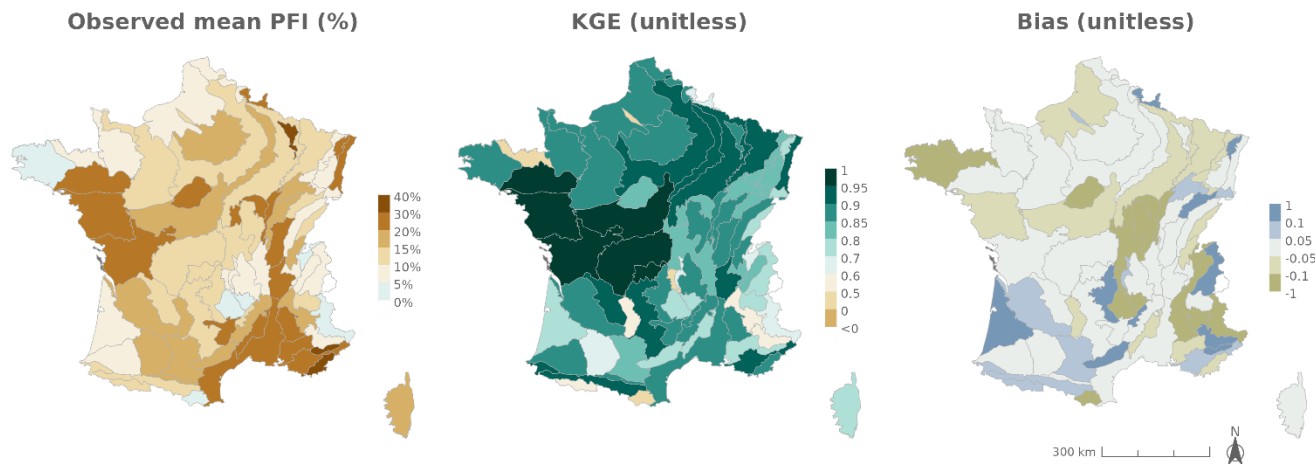

**Figure 4: Leave-One-Year-Out cross-validation assessing the calibration of the logistic regressions using discharge measurements from gauging stations. From left to right: observed mean *PFI* measured during the calibration period (2012-2022, left), Kling-Gupta**
**Efficiency (KGE, centre) and absolute bias of *PFI* predictions (right).**

The logistic regressions fitted with discharge data simulated by hydrological models also accurately reconstruct *PFI* from May

to September. Compared to the results obtained with observed discharge data, the performance is slightly lower due to the

inherent imperfections of the SAFRAN reanalysis compared to the observed climate (Fig. 5, 6; Table 1; Appendix Sect. I).

The inter-HER2 and inter-projections median estimated $mPFI_{7-10}$ across France is 11.3-13.5% depending on hydrological

models, which is close to the 14.3% value (Q1-Q3: 8.4-20.3) derived from the ONDE network. The KGE is slightly lower but

exceeds 80% in more than half of the HER2 using the CTRIP (38/75), SMASH (38/75), and GRSD (43/75) models. The inter-

HER2 median of the NSE skill score (calculated for all RCP-GCM-RCM modelling chains and for each of these three

hydrological models) indicates that between 71% and 74% of the variance in $mPFI_{7-10}$ is explained by the logistic regressions.

The other skill scores do not show a loss of performance (KGE between 0.80 and 0.81; MAE between 0.04 and 0.05, and

RMSE between 0.06 and 0.07) when comparing the results of the Leave-One-Year-Out cross-validation obtained using

observed and simulated discharge data. Lower performance is observed with the ORCHIDEE and J2000 models, as the KGE

exceeds 80% in only 7/75 and 5/37 HER2, respectively. Their median KGE (0.60 and 0.69, respectively), NSE (0.49 and 0.62), MAE (0.06), and RMSE (0.08 and 0.09) are lower than the values obtained during calibration using observed flow data, 340 indicating that these two hydrological models seem less sensitive.

Regardless of the hydrological model of interest, the maps of skill scores show spatial patterns that are consistent with the results obtained with observed discharge data (Fig. 6; Appendix Sect. I). Both NSE and KGE remain high in the sedimentary plains and the southern part of the country, while they are less favourable in mountainous regions (Alps, Massif Central, and 345 Pyrenees). An overestimation persists in the southwestern part of France, while underestimations of *PFI* values are found in the northwestern, northern, and southeastern parts of France, and are especially pronounced during dry years. The models also successfully reproduce the inter-annual variability of *PFI*, particularly the alternation of dry (e.g., 2019) and wet (e.g., 2015) years (Fig. 5). Thus, the modelling chains demonstrate their ability to simulate *PFI* over the period 2012–2022 and are therefore deemed reliable to attempt projecting changes in flow intermittence under modified climate conditions. However, note that 350 *PFI* values are slightly underestimated when the logistic regressions are calibrated on the driest years and then applied to wetter years. This suggests that our future projections are likely to be biased in the same way, since the calibration is based on current conditions before applying the regression models to a potentially drier future climate.

.

**Table 1: Evaluation of the logistic regressions calibrated using observed discharge data (results of the cross validation) and using** 355 **flows simulated by the CTRIP, GRSD, J2000, ORCHIDEE and SMASH hydrological** models**. Statistics of the skill scores are summarized by medians and the first and the third quartiles (Q1-Q3, into brackets) across all HER2s.**

| Hydrological model | NSE (unitless) | KGE (unitless) | MAE (unitless) | RMSE (unitless) |
|---|---|---|---|---|
| **Observed discharge data** **Leave One Year Out (2012-2022)** | 0.79 ( Q1-Q3: 0.64-0.84) | 0.85 ( Q1-Q3: 0.77-0.89) | 0.10 ( Q1-Q3: 0.06-0.13) | 0.07 ( Q1-Q3: 0.05-0.09) |
| **CTRIP** | 0.74 ( Q1-Q3: 0.61-0.80) | 0.80 ( Q1-Q3: 0.69-0.85) | 0.05 ( Q1-Q3: 0.04-0.07) | 0.07 ( Q1-Q3: 0.05-0.10) |
| **GRSD** | 0.74 ( Q1-Q3: 0.64-0.80) | 0.81 ( Q1-Q3: 0.71-0.86) | 0.04 ( Q1-Q3: 0.03-0.06) | 0.06 ( Q1-Q3: 0.04-0.08) |
| **J2000** | 0.62 ( Q1-Q3: 0.52-0.71) | 0.69 ( Q1-Q3: 0.63-0.79) | 0.06 ( Q1-Q3: 0.04-0.09) | 0.09 ( Q1-Q3: 0.07-0.13) |
| **ORCHIDEE** | 0.49 ( Q1-Q3: 0.39-0.63) | 0.60 ( Q1-Q3: 0.50-0.72) | 0.06 ( Q1-Q3: 0.04-0.09) | 0.08 ( Q1-Q3: 0.06-0.11) |
| **SMASH** | 0.71 ( Q1-Q3: 0.65-0.80) | 0.80 ( Q1-Q3: 0.72-0.86) | 0.04 ( Q1-Q3: 0.03-0.06) | 0.06 ( Q1-Q3: 0.05-0.08) |

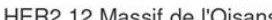

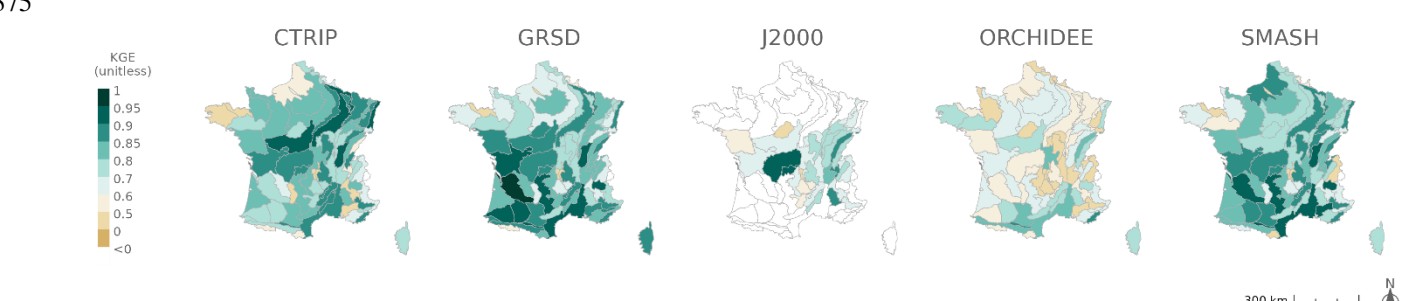

**Figure 5: Time series of *PFI* for the four HER2s selected for illustrative purposes. Dots and lines are observed *PFI* derived from the ONDE network and estimated *PFI* computed by the logistic regressions using simulated discharge by each hydrological model, respectively.**

**Figure 6: Kling-Gupta Efficiency (KGE) assessing the calibration of logistic regression using discharge simulated by each hydrological model with SAFRAN-based simulations available during the 2012-2022 period**

## 4.3 Flow intermittence projections

The logistic regressions fitted for each HER2 are applied to derive the *PFI* values from the historical (1976-2004) and future
(2005-2100) discharges simulated by the hydrological models forced by RCP-GCM-RCM climate projections over the period
1976-2100.

Figure 7 shows the evolution of $mPFI_{7-10}$ and of the start date *(Tf)* and end date *(Tl)* of the dry period for 2041-2070 and 2070-
2099 compared to 1976-2005. The results are illustrated in Fig. 7 with the median projection given by the GRSD model (see
Appendix Fig. J1-J4 for other models) under RCP2.6, RCP4.5 and RCP8.5 climate scenarios. Table 2 details the same results
for the four HER2 under RCP8.5.

Overall, the average *PFI* shows an increase, indicating a gradual intensification of dry periods over the 21[st] century under the
RCP 4.5 and RCP 8.5 scenarios. The inter-HER2 and inter-projections median of $mPFI_{7-10}$ calculated at the national scale
($mPFI_{7-10}[France]$) for the baseline period 1976-2005 ranges from 10 to 12% depending on the hydrological model. However,
the median projection of $mPFI_{7-10}$ during 1976-2005 is higher than 20% in 12 to 20 of the 75 HER2s (depending on the
hydrological model), although it does not exceed 42% in any HER2. Under RCP 8.5, $mPFI_{7-10}[France]$ could reach values
between 13 and 21% in the mid-term of the century 2041-2070 and between 16 and 29% by the end of the century 2070-2099.
By this time, six HER2s could have inter-projections median of $mPFI_{7-10}$ exceeding 50% according to at least two hydrological
models: four in the south-east of France; one in the southwest; and one in the northwest of France. Under RCP 4.5, changes
are more moderate by the end of the century 2070-2099, with $mPFI_{7-10}[France]$ ranging between 14 and 20%. The spatial
pattern of the changes is not uniform but looks similar between RCP 8.5 and RCP4.5. A strengthening divide between the
south-west and the north-east of the country is projected. Regions already prone to intermittence are expected to experience
an increase in this phenomenon under both emission scenarios. The ratio of the ensemble median of $mPFI_{7-10}$ at the end of the
century compared to the baseline period evolves around 1.4 for RCP 4.5 and 1.9 for RCP 8.5. Mountainous regions see the
intensity of summer dry periods increase but remain relatively spared.

Shifts of the start and the end of the dry period are partly correlated with changes in $mPFI_{7-10}$ (Fig. 7). It appears that climate
change results in both earlier and later dry periods in a fairly symmetrical manner: dry periods are advanced and extended, but
with different sensitivities structured along a north-south gradient. Under RCP 8.5 climate conditions, the dry periods are
projected to get (on median) longer in southern France at the end of the century. According to at least two hydrological models,
the highest shifts of *Tf* or *Tl* could exceed 5 weeks in several mountainous HER2 (four HER2 in the Alps, three in the Pyrenees,
two in the Jura). In contrast in the northern part of France, the dry periods could be advanced and extended by only one or two
weeks, as observed in HER2-81 "Plaine de Bourgogne" (Table 2). HER2 28 could be one of the most affected in the north due
to its impermeable clay-sandy formations, which differs from the neighbouring sedimentary formations. One can note a change

in seasonality for the HER2-12 "Massif de l'Oisans" located in the Alps (Fig. 8, example with the GRSD hydrological model). No-flow events were concentrated in winter (retention of water in the snow cover) during the historical period. Under climate change, temperature will be higher leading to less snowfall and more runoff in the river network during winter. The river flow regime will be more sensitive to losses by evapotranspiration and finally no-flow conditions will likely occur in summer by the end of the 21[st] century.

| HER2 | Period | $mPFI_{7-10}$ (%) | | | $Tf$ (date) | | | $Tl$ (date) | | | Nb Days PFI > 20% | | |
|---|---|---|---|---|---|---|---|---|---|---|---|---|---|
| | | Min | Med | Max | Min | Med | Max | Min | Med | Max | Min | Med | Max |
| Haute Normandie Picardie (57) | 1976-2005 | 8 | 13 | 15 | 30/07 | 22/08 | 05/10 | 22/09 | 30/10 | 04/12 | 3 | 33 | 40 |
| | 2041-2070 | 7 | 14 | 23 | 15/07 | 24/08 | 11/10 | 11/10 | 10/11 | 16/12 | 1 | 38 | 98 |
| | 2070-2099 | 5 | 16 | 34 | 04/06 | 21/08 | 23/11 | 16/10 | 12/11 | 23/12 | 0 | 37 | 185 |
| Plaine de Bourgogne (81) | 1976-2005 | 12 | 19 | 22 | 10/07 | 31/07 | 18/08 | 18/09 | 27/10 | 16/12 | 31 | 65 | 83 |
| | 2041-2070 | 11 | 23 | 34 | 09/07 | 02/08 | 01/09 | 02/10 | 04/11 | 29/12 | 31 | 79 | 140 |
| | 2070-2099 | 10 | 27 | 47 | 21/06 | 22/07 | 03/09 | 04/10 | 08/11 | 31/12 | 31 | 93 | 189 |
| Massif de l'Oisans (12) | 1976-2005 | 6 | 11 | 17 | 26/07 | 23/08 | 06/10 | 18/09 | 19/11 | 17/03 | 23 | 32 | 128 |
| | 2041-2070 | 11 | 21 | 38 | 27/06 | 01/08 | 01/09 | 23/09 | 30/10 | 03/12 | 25 | 65 | 139 |
| | 2070-2099 | 12 | 29 | 52 | 29/05 | 12/07 | 21/08 | 28/09 | 02/11 | 15/12 | 38 | 92 | 197 |
| Plaine méditerra-néenne (105) | 1976-2005 | 14 | 25 | 29 | 22/06 | 17/07 | 16/08 | 22/09 | 23/10 | 17/12 | 37 | 85 | 103 |
| | 2041-2070 | 9 | 31 | 46 | 29/05 | 11/07 | 28/08 | 25/09 | 01/11 | 25/12 | 28 | 107 | 168 |
| | 2070-2099 | 9 | 41 | 57 | 02/04 | 16/06 | 30/08 | 02/10 | 13/11 | 01/01 | 33 | 155 | 228 |

Table 2: Statistics of flow intermittence characteristics for the four illustrative HER2s under RCP 8.5. The minimum (min), median (med) and maximum (max) are the ensemble minimum, median and maximum across all the projections and hydrological models. *Nb Days PFI > 20%*: Number of days in dry periods (period of the year when *PFI* exceeds 20%).

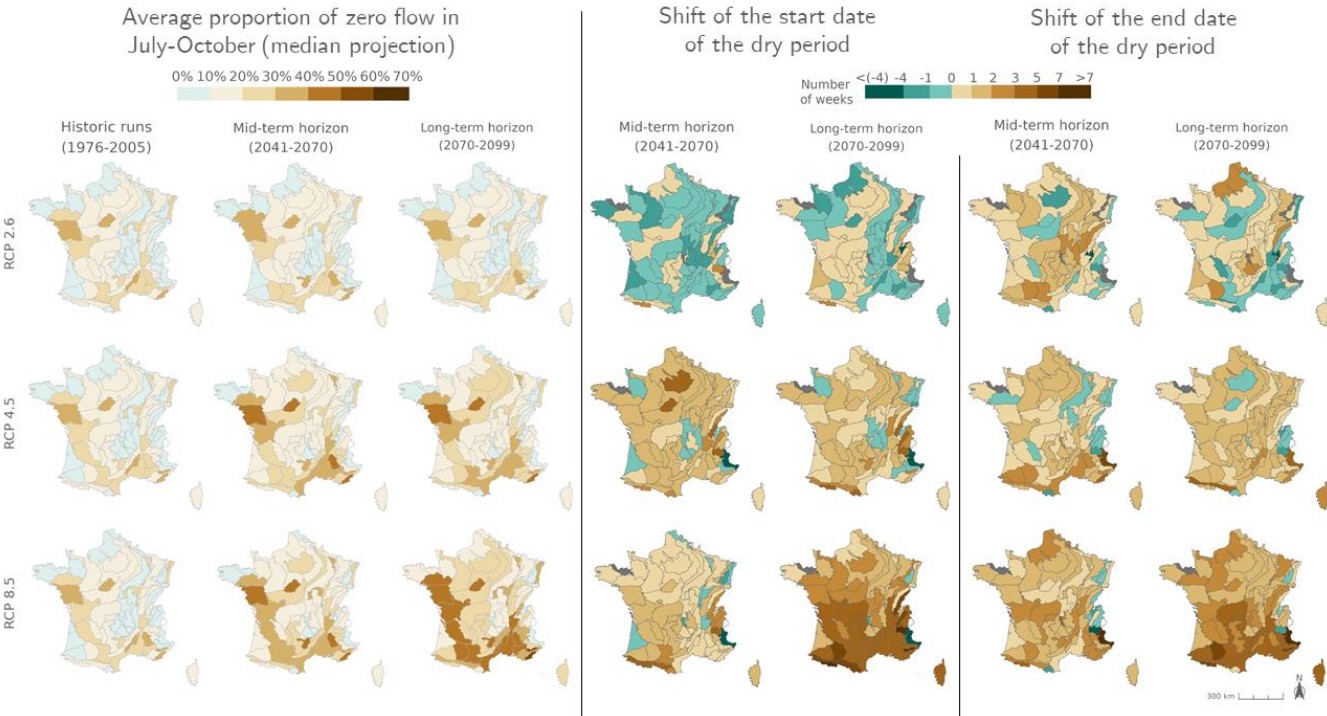

Figure 7: Ensemble median *mPFI$_{7-10}$* (columns 1 to 3), and ensemble median shift of the start date *Tf* (columns 4 and 5) and the end date *Tl* (columns 6 and 7) of the dry period over the two periods 2041-2070 and 2070-2099 under the three RCPs for the GRSD hydrological model, relative to the baseline period 1976-2005. The shift, expressed in week, takes a positive value when the duration of the dry period increases. Grey HER2s had no period with a *PFI* >20% during the reference period. The same results are available for the other hydrological models in Appendix Fig. J1-J4.

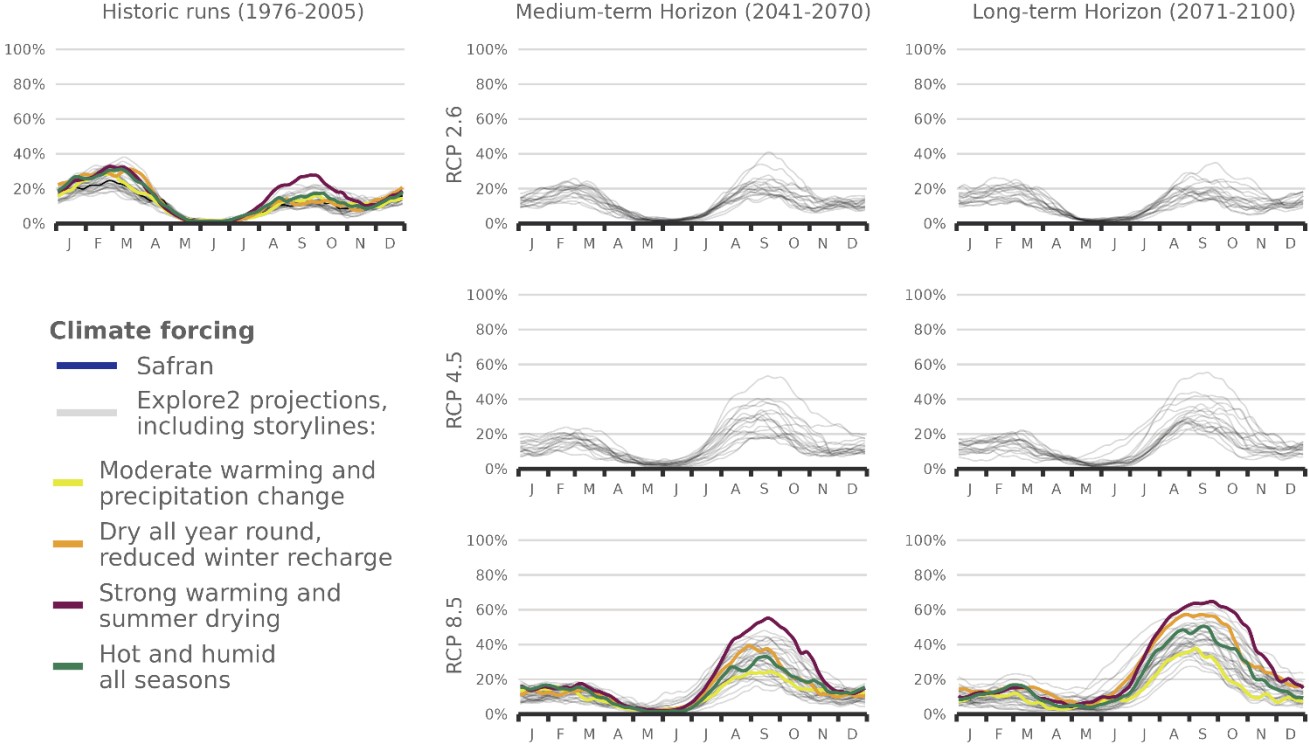

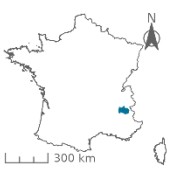

**Average proportion of zero flows smoothed over 5 days
Hydrological model: GRSD - HER2 12**

Historic runs (1976-2005)          Medium-term Horizon (2041-2070)          Long-term Horizon (2071-2100)

**Climate forcing**
— Safran
— Explore2 projections, including storylines:
— Moderate warming and precipitation change
— Dry all year round, reduced winter recharge
— Strong warming and summer drying
— Hot and humid all seasons


**Figure 8: Seasonal pattern of *PFI* for the periods 1976-2005, 2041-2070 and 2070-2099 under RCP 2.6, 4.5 and 8.5 scenarios for GRSD hydrological model in the Alps (HER2 12). The storylines highlighted in this figure are introduced in Appendix Sect. J. The same results are available for HER2s 57, 81 and 105, and for the other hydrological models in Appendix Fig. J5-21.**

**4.4 Uncertainties in hydro-climatic projections and their impact on *PFI* projections**

Figure 9 illustrates the results of the uncertainty analysis, showing the uncertainty range of *mPFI*$_{7-10}$*[France]* throughout the 21$^{st}$ century. These results confirm that dry conditions may occur more frequently in a changing climate, with 22% projected increase in *mPFI*$_{7-10}$*[France]* by the end of the century. The total uncertainty of *mPFI*$_{7-10}$*[France]* over all projections results from the accumulation of uncertainties related to RCP scenarios, GCMs and RCMs, hydrological models, residual variability, and internal variability. The confidence interval, and therefore the extent of change, increases over the course of the 21$^{st}$ century

due to the divergent results of the modelling chain. This finding is consistent with studies using a similar methodology to assess the uncertainty of panels of flow projections (Evin et al., 2019; Aitken et al., 2023).

The contribution of each component to the total uncertainty varies over time. The fraction of total variance due to residual variability is small and remains stable over time. The main part of the uncertainty in climate change responses is thus explained by RCP, GCM, RCM and hydrological models. In addition, the contribution of the internal variability is highly predominant and represents more than 75% of the total uncertainty until the mid-century. This contribution then decreases with time as the total uncertainty increases. It becomes less than 50% of the total uncertainty by the end of the century but remains the greatest contributor to the total uncertainty over the whole simulation period.

Regarding the spatial distribution of the different contributions to the total uncertainty, it appears that climate models are predominant source of uncertainty, and that they are distributed in a spatially heterogeneous manner over France. The uncertainty related to the RCP scenarios increases over time and becomes predominant in the south, surpassing uncertainties related to other steps of the modelling chain (GCM, RCM, and hydrological models). In the mountainous regions, differences between results obtained with RCP4.5 and RCP8.5 are clearer by the end of the century, leading to a contribution of RCP to the total uncertainty reaching 20% at the end of the century. In northern France, the contribution of RCP to the uncertainty is lower compared to hydrological models, GCM, or RCM contributions. Uncertainty related to hydrological models is also spatially structured, with greater uncertainty in the northwestern part of France.

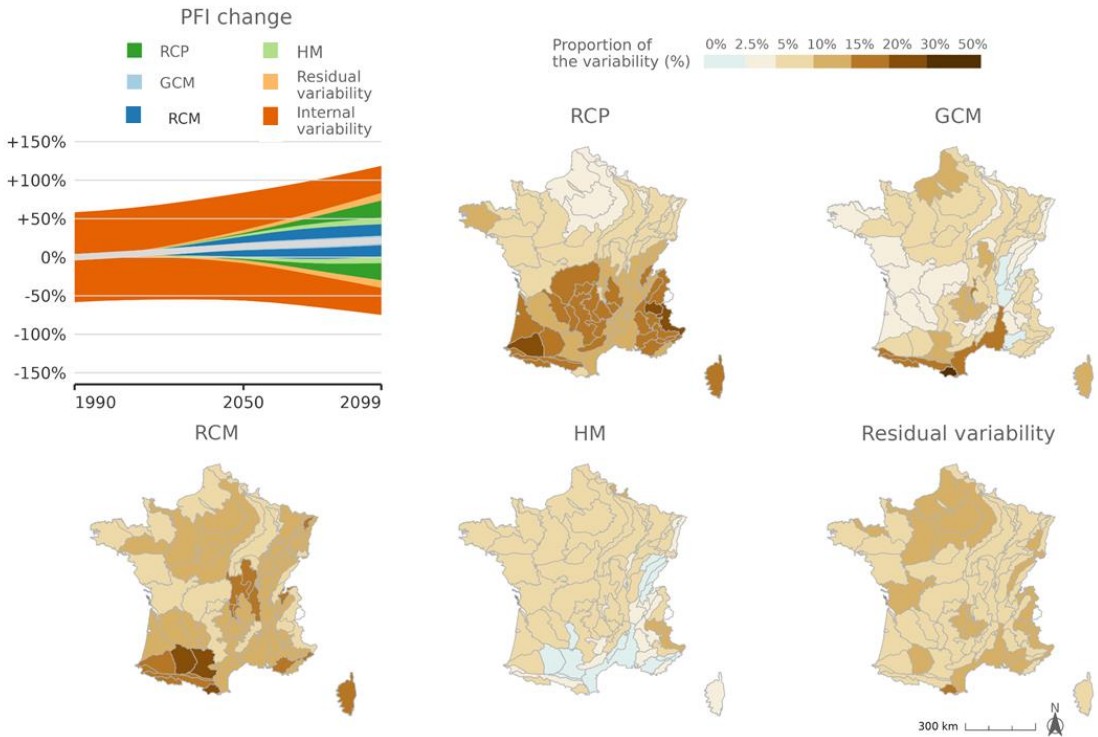

**Figure 9: Decomposition of the effects contributing to the variance of projections for relative changes (unitless) of *mPFI$_{7-10}$[France]*. In the top-left graph, the grey line and the coloured areas respectively show the climate change response and the contribution of time for each step of the modelling chain to the 90% confidence interval. For each model uncertainty and internal variability component, the vertical extent of the corresponding area is proportional to the fraction of total uncertainty explained by the component.**

**4.5 Agreement between changes**

Despite uncertainties about the intensity of change, *PFI* projections still show a good degree of convergence in the direction
of change when considering all climate projections driving each hydrological model individually, as well as in the overall
convergence of *PFI* projections (Fig. 10). *MIA* values are first calculated independently for each hydrological model, before
all projections are combined for multi-model assessment.

Figure 10 highlights spatial contrasts of the *MIA* multi-model approach: unsurprisingly, the projections of $mPFI_{7-10}$ are more
uncertain in the northern part of France. The climate change signal on the proportion of dry periods is not homogeneous across
France, with significant uncertainties remaining for the northern part.

The agreement of projections is high for the GRSD and SMASH models across France (*MIA* values close to +100% under
RCP8.5 in Fig. 10 and +80% under RCP4.5, see Appendix Sect. K). In contrast, CTRIP, J2000, and ORCHIDEE suggest
contrasted regional impacts of climate change on *PFI* under RCP4.5, with a reduction of the $mPFI_{7-10}$ in the northern part of
France, indicating a differentiated sensitivity of hydrological models to climate changes. This trend is also observed for
ORCHIDEE under RCP8.5, while the other models agree on an increasing drying.

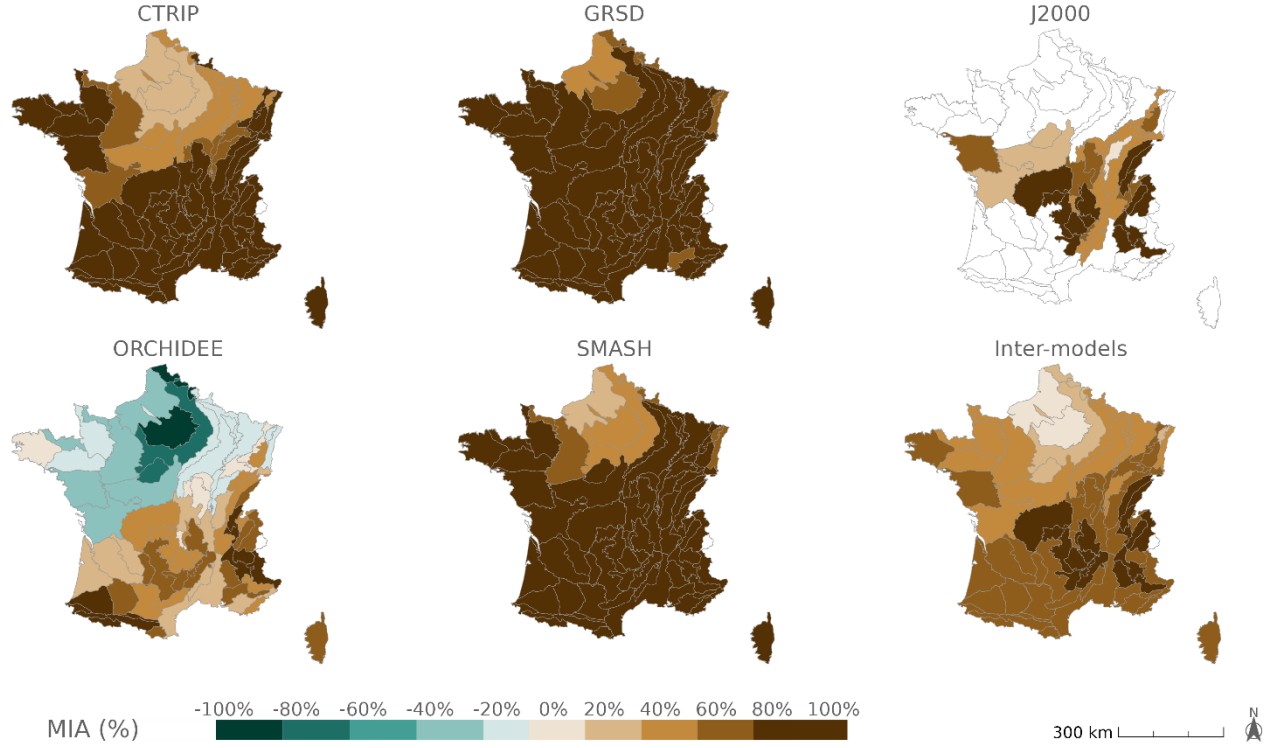

**Figure 10: Agreement between projections of $mPFI_{7-10}$ for each hydrological model and inter-model agreement on the change signal
of $mPFI_{7-10}$ under the RCP 8.5 scenario.**

## 5 Discussion

### 5.1 Modelling framework and assumptions

This study presents projections of the *PFI* in small streams by HER2 in France over the 21$^{st}$ century. It extends the previous analysis of Sauquet et al. (2021) on the 2012-2018 period by exploiting an extensive national dataset of stream intermittence observations collected on low-order streams (3302 observation sites), which are the most affected by shifts from perennial to intermittent flows (Reynolds et al., 2015; Dhungel et al., 2016). For the first time, we project the future evolution of the *PFI* using discharge data obtained from 5 hydrological models with conceptual (GRSD, SMASH), surface (CTRIP, ORCHIDEE), or process-oriented distributed (J2000) structures, under a multitude of possible climate scenarios informed by 17 pairs of CMIP5 GCM-RCM models.

The results of this study should be interpreted within the context of its underlying assumptions. Firstly, the study focuses on unregulated streams to characterize the "natural" hydrology (i.e., without considering water abstraction by anthropogenic activities or the impact of hydraulic engineering structures). This assumption was made in the Explore2 project, which provides the input streamflow simulations. Nevertheless, global water model simulations including direct human impacts by Döll and Zhang (2010) or Gudmundsson et al. (2021) concluded that ecologically relevant flow characteristics will be more altered by climate change than by withdrawals and dams. However, we believe that quantitatively estimating the extent to which flow intermittence is due to direct anthropogenic stressors is crucial. Such an estimation could improve our projections of flow intermittence and inform decision-making aimed at regulating and preventing water stress situations for populations.

Secondly, groundwater levels are not incorporated into the model, although they could potentially enhance the accuracy of the projections. Indeed, similar logistic regression models were used in Beaufort et al. (2018) and they incorporated groundwater levels measured by piezometers. Likewise, in Beaufort et al. (2019), climate, hydrological and morphological data were supplemented with groundwater levels to predict flow intermittence at a local scale. In these studies incorporating subsurface water in the regression models was thought to improve performance, in particular because the contribution of subsurface processes is known to play a decisive role in maintaining baseflow and mitigating flow intermittence. However, information about groundwater levels has not been included in the present study because hydrological projections incorporating groundwater levels have not yet been conducted across entire France (for instance, within the Explore2 project, projections of groundwater levels have only been produced for a restricted set of areas in France).

Thirdly, with only five annual discrete observations of streamflow intermittence over eleven years for the calibration of the logistic regression models, our ability to capture the full range of extremes and variability in these regression models remains imperfect. Yet, visual monitoring remains the most common technique for observing non-perennial streams. An alternative approach would be to consider citizen science to augment our database, although concerns about data reliability persist, in particular because past studies have shown participant agreement rates ranging from 46% to 70% (Scheller et al., 2024). Data scarcity necessitated conducting this analysis at the HER2 scale, which nonetheless represents a significant improvement of the spatial resolution by increasing the number of modelling subdomains from 22 to 75 compared to previous studies (Sauquet

et al., 2021). In future work, a downscaling process could enhance the usability of *PFI* projections for local stakeholders in water management.

Fourthly, while projections of no-flow events often exhibit significant uncertainties when derived directly from discharge simulated by hydrological models (Evin et al., 2024; Aitken et al., 2023), this study illustrates how categorical and discrete (in space and time) field observations can be combined with conventional stream gauge data, to enhance the understanding of small stream drying dynamics. In this way, despite the divergences between projections induced by climate data, the agreement within this multi-model approach is relatively strong, indicating a consensus toward increasing *PFI* throughout the 21$^{st}$ century.

## 5.2 A consistent signal across projections

This study confirms that logistic regressions properly capture the relationships between flows in large watersheds and the *PFI* of small streams. These regressions are calibrated and validated using discharge measurements from gauging stations, and subsequently using discharge simulated from SAFRAN climate data. Following this second calibration, projections based on regressions consistently indicate an increase in average *PFI* and a shift in the start and end dates of dry periods under RCP 4.5 and RCP 8.5 climate projections, suggesting a progressive intensification and extension of dry periods over the 21$^{st}$ century.

These results are consistent with previous studies indicating a transition of many streams from perennial to intermittent regimes (Jaeger et al., 2014; Reynolds et al., 2015; Dhungel et al., 2016; Schneider et al., 2013). In regions already affected by intermittence, an increase in the intensity and duration of drying periods is likely, a phenomenon also anticipated in other areas around the world (Jaeger et al., 2014; Dai et al., 2011). Increasing intermittence and decreasing flow rates are observed in catchment-scale studies: the fraction of time with zero discharge increases from 0.05% to 4.30% in a Swiss Alpine catchment between 2020–2040 and 2080–2100 (Halloran et al., 2023) while the no-flowing phase could extend by up to 12 days between 1980-2009 and 2030-2059 for a catchment in southern Italy (De Girolamo et al., 2022). This trend is also noticeable on a larger scale, as monitoring of gauging stations in five regions with Mediterranean climates around the world between 1980 and 2019 (Carlson et al., 2024) and over 452 rivers on the European continent between 1970 and 2010 (Tramblay et al., 2020) both show that approximately 30% of them have already experienced drier conditions due to climate change, with modified flow regimes or extended periods of drying.

## 5.3 Uncertainties in northern France

In the northern part of France, discrepancies between GCM-RCM-Hydrological Models projections result in higher uncertainties in *PFI* projections and pronounced geographical contrasts on the multi-model-based *MIA* map (Fig. 10). Part of these uncertainties can be attributed to the underestimation of *PFI* in this region, particularly during dry years (Sect. 4.2, Appendix Fig. H2 and Table H1). However, the primary source of these uncertainties likely stems from uncertainties in future rainfall patterns in this region where the majority of Explore2 projections indicate an increase in winter rainfall and winter mean flows (for 8 out of 9 hydrological models) and a decrease in summer precipitation (Evin et al., 2024; Sauquet et al., 2024). The annual precipitation by the end of the century remains uncertain, due to the compensatory effect between increased

winter recharge and an increased evapotranspiration (Ribes et al., 2022; Douville et al., 2021). As a result, some hydrological models predict a shortening of the dry period for certain northern HER2 regions, including under RCP 8.5. Similar uncertainties are observed along the east coast of the USA, where precipitation changes could turn intermittent streams into perennial ones (Dhungel et al., 2016) while alternative climate change scenarios projected that by the 2040s, approximately half of the streams in Washington state would shift from snow-fed to rain-fed, resulting in reduced annual discharge (Reidy Liermann et al., 2012).

## 5.4 Transformation of the snowmelt regime in the Alps

The mountain ranges could be moderately affected by the increased probability of dry conditions, consistent with our previous modelling efforts (Sauquet et al., 2021), but our projections anticipate two specific phenomena in these regions. First, the Pyrenees are more affected than the other mountain ranges, with significant changes impacting the massif and the dependent basins in southwestern France. Additionally, Alpine HER2s will probably undergo hydrological regime changes. Higher winter temperatures will lead to reduced snowfall and snow-related intermittence. This snowpack reduction will also reduce groundwater recharge by spring melt. In addition, since soils tend to retain less water in summer due to more intense and brief rainstorms (Rutkowska et al., 2023), an increased summer intermittence is expected despite increasing summer precipitations. Using 16 hydrographic variables describing the magnitude, frequency, timing, duration, and rate of change of the flow regime at 59 primarily selected sites with a Strahler order of 5, Dhungel et al. (2016) also observed the reduction in the snowmelt regime at 2 Rocky Mountain sites under climate change. Halloran et al. (2023) also conclude that groundwater will play an increasingly important role in ensuring flow in alpine streams and that the shift from perennial to intermittent could occur for alpine streams over the course of the current century.

Furthermore, the reduction of the snowmelt regime in the Alps indicate that we can still make projections consistent with the literature beyond the calibration period (May to September). More generally, the hypothesis of temporal transferability of the models is a strong assumption in data exploitation, as it assumes that climate models, statistical adjustment methods and hydrological models can simulate the behaviour of the systems they represent in a future hydro-climatic context that is very different from the one in which they were developed (Evin et al., 2024). In this context, it is important to recall the risk that the projections may be underestimated, as demonstrated by the validation of logistic regressions calibrated on wetter years compared to dry years.

## 6 Conclusion

This study assesses the changes in the intermittency of river flows across France in the context of climate change. For the first time, multi-model and multi-scenario hydro-climatic projections are used as a predictor to explore the possible evolution of the daily probability of flow intermittency at the scale of (level 2) Hydro-EcoRegions. Leveraging monthly monitoring of small streams over ten summers, we calibrated logistic regressions to transform hydrological projections of large watersheds into regional proportions of flow intermittence for the 21$^{st}$ century. Under both RCP 4.5 and 8.5 scenarios, robust signals

indicate an intensification of dry events, marked by increased probability of flow intermittence and longer dry periods throughout the year. These changes are projected to be more pronounced in southern France, with greater uncertainty in the northern half of the country. Mountain areas could remain relatively spared from summer dry periods but shifts in hydrological regimes are anticipated. By the end of the century under RCP8.5, dry stream phenomena along the Atlantic coast could surpass those currently observed in the Mediterranean region by the usual monitoring campaigns. The evolution of droughts and reduced water availability suggested by these results could lead to significant ecological impacts, including alterations in the structure and function of freshwater ecosystems (such as changes in microbial activity and habitat loss), shifts in soil chemistry, increased carbon and solute fluxes, and sediment mobilization (Geris et al., 2015).

## Appendix A: Explore2 modelling

| Global Climate Model | Regional Climate Model | Histo-rical | RCP 2.6 | RCP 4.5 | RCP 8.5 |
|---|---|---|---|---|---|
| CNRM-CERFACS-CNRM-CM5 | CNRM-ALADIN6 | × | × | × | × |
| CNRM-CERFACS-CNRM-CM5 | MOHC-HadREM3-GA7-05 | × | - | - | × |
| ICHEC-EC-EARTH | KNMI-RACMO22E | × | × | × | × |
| ICHEC-EC-EARTH | SMHI-RCA4 | × | × | × | × |
| ICHEC-EC-EARTH | MOHC-HadREM3-GA7-05 | × | × | - | × |
| MOHC-HadGEM2-ES | CNRM-ALADIN63 | × | - | - | × |
| MOHC-HadGEM2-ES | CLMcom-CCLM4-8-17 | × | - | × | × |
| MOHC-HadGEM2-ES | ICTP-RegCM4-6 | × | × | - | × |
| MOHC-HadGEM2-ES | MOHC-HadREM3-GA7-05 | × | × | - | × |
| IPSL-IPSL-CM5A-MR | DMI-HIRHAM5 | × | - | - | × |
| IPSL-IPSL-CM5A-MR | SMHI-RCA4 | × | - | × | × |
| MPI-M-MPI-ESM-LR | CLMcom-CCLM4-8-17 | × | × | × | × |
| MPI-M-MPI-ESM-LR | ICTP-RegCM4-6 | × | × | - | × |
| MPI-M-MPI-ESM-LR | MPI-CSC-REMO2009 | × | × | × | × |
| NCC-NorESM1-M | DMI-HIRHAM5 | × | - | × | × |
| NCC-NorESM1-M | GERICS-REMO2015 | × | × | × | × |
| NCC-NorESM1-M | IPSL-WRF381P | × | - | - | × |

**Table A1: Global and Regional Climate Models combinations driving the Explore2 Hydrological Models selected for *PFI* simulation in the 21st century**

**Appendix B: Sensitivity analysis of the duration of flow measurement interval used as input for the logistic regression**
**to calculate the *PFI***

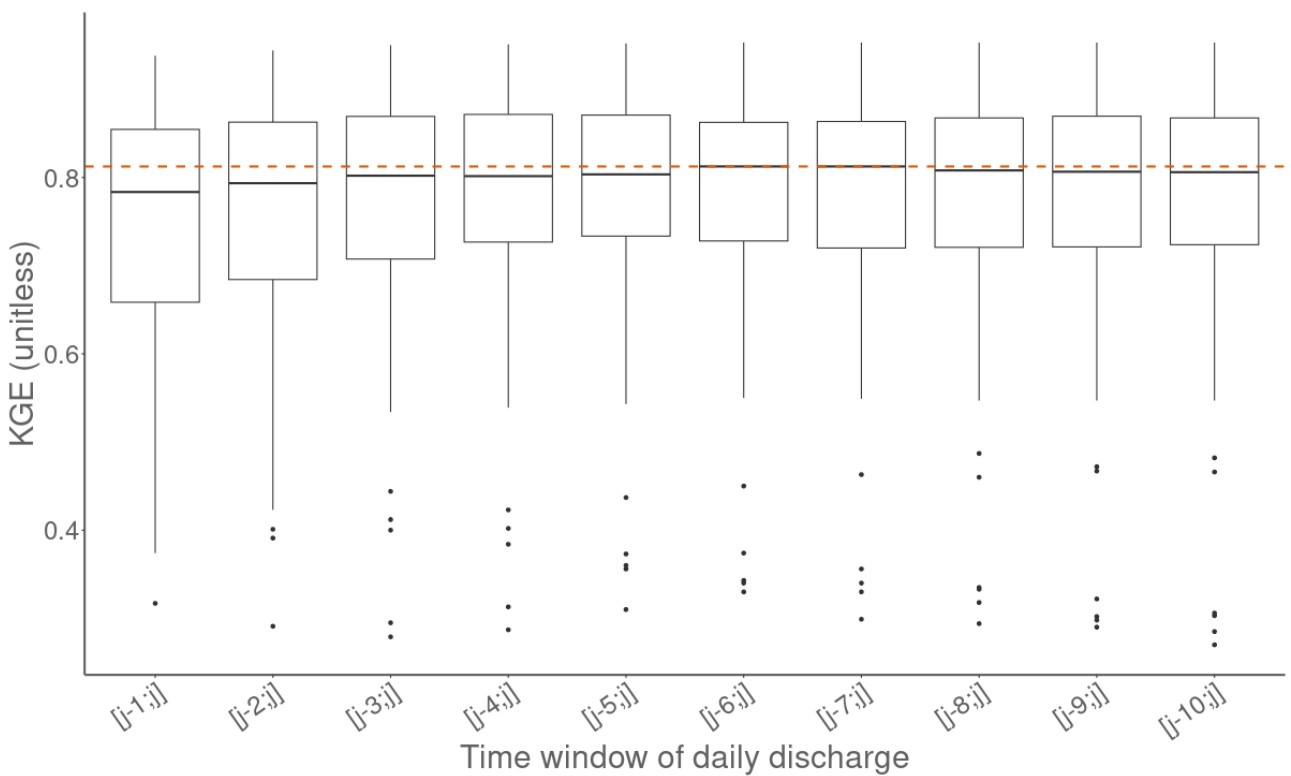

**Figure B1: Kling-Gupta Efficiency (KGE) computed on results obtained by Leave One Year Out validations, testing the sensitivity to the time window of daily discharge used for calibrating the logistic regressions. Boxes represent the quartiles Q1 and Q3, the whiskers extend up to 1.5 times the interquartile range above Q3 and below Q1 and points located beyond the whiskers are displayed**
**individually. The dashed line represents the median KGE of the [j-6;j] window**

A sensitivity analysis was performed to fix the time window of daily discharge that optimizes the calibration of logistic regressions presented in Sect. 3.1. The calibration and validation were performed using the Leave One Year Out method, detailed in Sect. 3.3. The model performance was assessed using the Kling-Gupta Efficiency (KGE) (Gupta et al., 2009). The
KGE values for the different HER2 are summarized using boxplots for each tested window size. The [j-6;j] window was selected because it corresponds to the highest median KGE score, the narrowest interquartile range, and the highest minimum KGE.

**Appendix C: Separation of dataset based on dry, intermediate and wet years**


The data from ONDE sites used for validation were collected between May and September over 11 years from 2012 to 2022. The robustness of the model was assessed through a validation process involving three sets of dry, intermediate, and wet years. For each test, model calibration was performed using the years excluded from the validation set.

The hydrological years are distributed into three equal groups of hydrological years (dry years, intermediate years and wet years) according to the annual aridity index, calculated at the national scale. The aridity index AI was given by the ratio between the total annual precipitation and potential evapotranspiration from August 1 of the previous year to July 31 of the current year (Barrow, 1992; Figure A2.1). A set of dry years was formed using hydrological years where potential evapotranspiration exceeded annual precipitation ($AI < 1$ for 2017, 2019, and 2022). Two sets were then created: the four years

with AI greater than 1.4 were classified as wet years (2012, 2015, 2020 and 2021), and the remaining four years with AI between 1.15 and 1.37 were classified as intermediate years (2013, 2014, 2016, 2018).


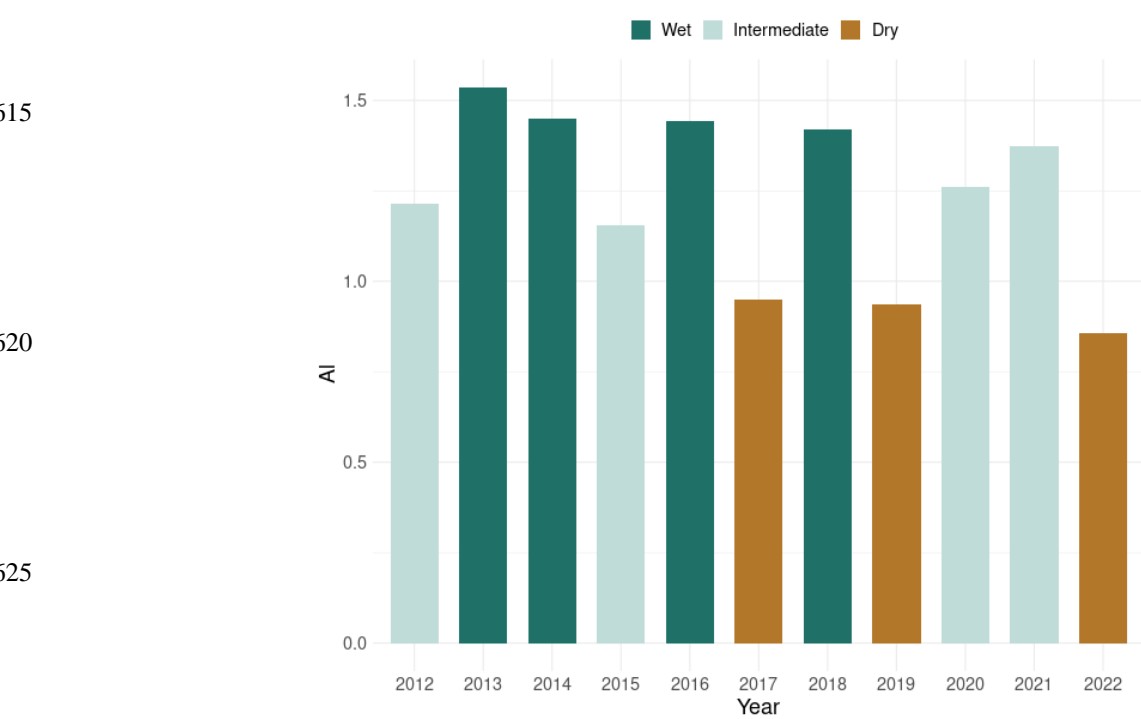



**Figure C1: Aridity index for the hydrological years 2012 to 2022**


**Appendix D: Assignment of simulation points**

After establishing the models' reliability through validation at secure monitoring stations, the next step involved extrapolation, which entailed linking these monitoring sites to the nearest simulation points in Explore 2 to project future hydrological scenarios while preserving the existing data structure. The proximity between the gauging station A and a simulation point in the Explore2 project is measured by the distance:

$$Dist(A,B) = \sqrt{\left(X(A) - X(B)\right)^2 + \left(Y(A) - Y(B)\right)^2 + (\alpha \times \Delta S_{rel})^2} \tag{D1}$$

Here, $(X(A), Y(A))$ and $(X(B), Y(B))$ are coordinates of $A$ and $B$ (in km), respectively, and $Surf(A)$ and $Surf(B)$ are the drainage areas of $A$ and $B$, respectively that are used to compute the relative difference between the drainage areas (absolute value).

$$\Delta S_{rel} = 2 \times \frac{|Surf(A) - Surf(B)|}{|Surf(A) + Surf(B)|} \tag{D2}$$

The coefficient $\alpha$ is used to balance the importance of geographical distance and the relative difference in surface and was set to 100.

 **Appendix E: Location of the ONDE sites, the gauging stations and the Explore2 simulation points selected for each hydrological model**

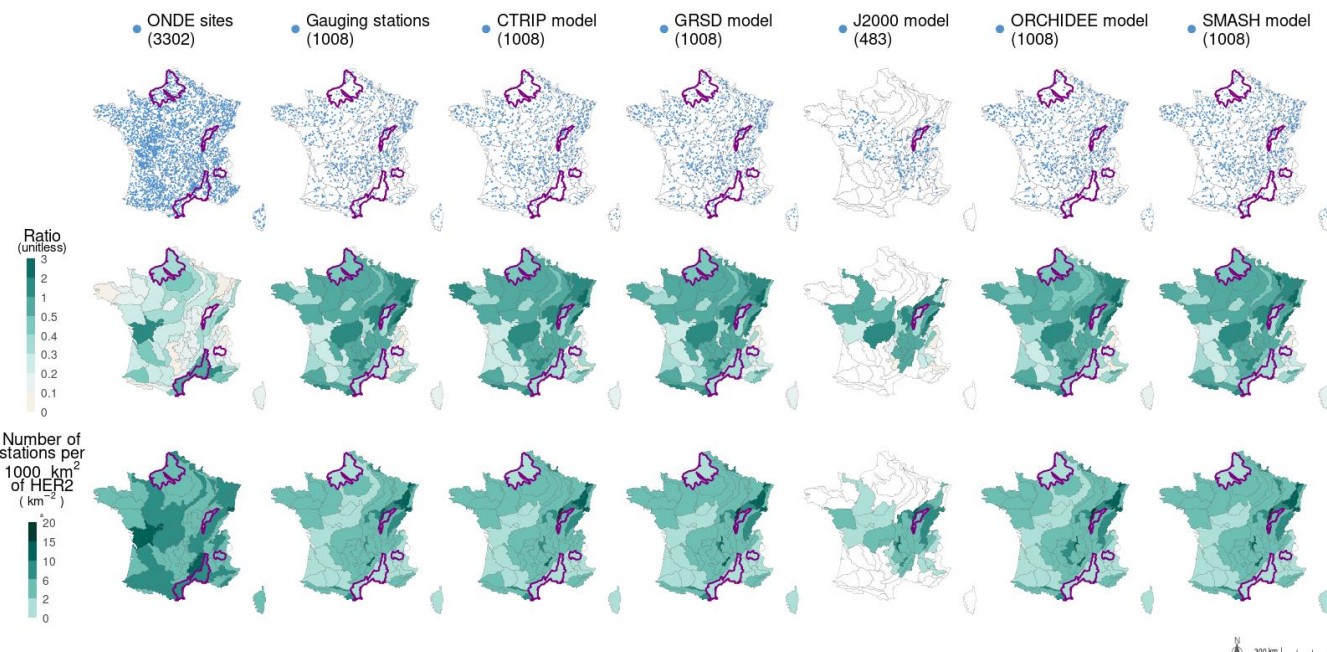

 **Figure E1: Location of the 3302 ONDE sites, the 1008 gauging stations selected from the HydroPortail database, and the Explore2 simulation points selected for each hydrological model (line 1). Maps on the second and third line respectively show the ratio between the sum of the catchment areas defined by the gauging stations or the simulation points and the surface area of the HER2 they intersect (line 2), and the density of gauging stations or simulation points (line 3).**

## Appendix F: QUALYPSO method


QUALYPSO method is applied to the multi-scenario multi-model ensembles of projections of the national average of $mPFI_{7-10}$ ($mPFI_{7-10}[France]$), weighted by HER2 areas and obtained under RCP 2.6, 4.5 and 8.5 climate change scenarios. According to Evin et al. (2019), we assume that the variable $mPFI^*_{7-10\,i,j,k,l}(t)$ characterizes the change of $mPFI_{7-10}$ between a year $t$ and the control year $c$ (e.g., $c = 1990$ in this study), for a given combination of RCP scenario $i$, GCM $j$, RCM $k$,

hydrological model $l$:

$$mPFI^*_{7-10\,i,j,k,l}(t) = mPFI_{7-10\,i,j,k,l}(t) - mPFI_{7-10\,i,j,k,l}(c)$$

The change of $mPFI_{7-10}$ can be split up into:

$$mPFI^*_{7-10\,i,j,k,l}(t) = \varphi^*_{i,j,k,l}(t) + \eta^*_{i,j,k,l}(t)$$

Where

$\varphi^*_{i,j,k,l}(t)$ is the climate change response of the RCP/GCM/RCM/Hydrological model combination,

$\eta^*_{i,j,k,l}(t)$ is the deviation from the climate change response for this RCP/GCM/RCM/Hydrological model combination, as a result of internal variability, representing natural and stochastic variability in the climate system.

For each time $t$, the climate change response characterized by $\varphi^*_{i,j,k,l}(t)$ of any RCP/GCM/RCM/Hydrological model combination can be expressed as:

$$\varphi^*_{i,j,k,l}(t) = \mu(t) + \alpha_i(t) + \beta_j(t) + \gamma_k(t) + \theta_l(t) + \varepsilon_{i,j,k,l}(t)$$

Where

$\mu(t)$ is the ensemble mean climate change response,

$\alpha_i(t)$ is the main effect of emission scenario $i$ (i.e., the mean deviation of RCP scenario $i$ from $\mu(t)$),

$\beta_j(t)$ is the main effect of GCM $j$,

$\gamma_k(t)$ is the main effect of RCM $k$,

$\theta_l(t)$ is the main effect of the hydrological model $l$,

$\varepsilon_{i,j,k,l}(t) = \varphi^*_{i,j,k,l}(t) - \mu(t) - \alpha_i(t) - \beta_j(t) - \gamma_k(t) - \theta_l(t)$ correspond to residual terms. For each year $t$, $\varepsilon_{i,j,k,l}(t)$ are

assumed to be independent and identically distributed over all scenarios, GCMs, RCMs, and hydrological models, and to follow normal distributions, with mean 0 and variance $\sigma^2(t)$.

If $\varphi^*_{i,j,k,l}(t)$ and $\eta^*_{i,j,k,l}(t)$ are assumed to be independent, the total variance of the change variable $mPFI^*_{7-10\,i,j,k,l}(t)$ is given by:

$$Var\left(mPFI^*_{7-10\,i,j,k,l}(t)\right) = Var\left(\varphi^*_{i,j,k,l}(t)\right) + Var(\eta^*_{i,j,k,l}(t))$$

Where

$Var(\eta^*_{i,j,k,l}(t))$ is the uncertainty associated to internal variability of $mPFI^*_{7-10\,i,j,k,l}(t)$.

$Var(\varphi^*_{i,j,k,l}(t))$ is the uncertainty in the climate change response, calculated as the sum of different uncertainty components:

$$Var(\varphi^*_{i,j,k,l}(t)) = Var(\mu(t)) + Var(\alpha_i(t)) + Var(\beta_j(t)) + Var(\gamma_k(t)) + Var(\theta_l(t)) + Var(\varepsilon_{i,j,k,l}(t))$$

RCP uncertainty $Var(\alpha_i(t))$ quantifies the variability in $mPFI^*_{7-10\,i,j,k,l}(t)$ values across the RCP scenarios. Specifically, $\alpha_i(t)$ represents the differences between the means of $mPFI^*_{7-10\,i,j,k,l}(t)$ involving each scenario $i$ and the overall mean of $mPFI^*_{7-10\,i,j,k,l}(t)$ across all scenarios. This component illustrates how differences between emission scenarios contribute to the total uncertainty in the projected $mPFI^*_{7-10\,i,j,k,l}(t)$.

Similarly, GCM, RCM and hydrological model contributions to $mPFI^*_{7-10\,i,j,k,l}(t)$ uncertainties are characterized by the variances for GCMs ($Var(\beta_j(t))$), RCMs ($Var(\gamma_k(t))$), and hydrological models ($Var(\theta_l(t))$).

Residual variability ($Var(\varepsilon_{i,j,k,l}(t))$) captures interaction effects and unexplained variability not accounted for by the main effects.

**Appendix G: Metadata**

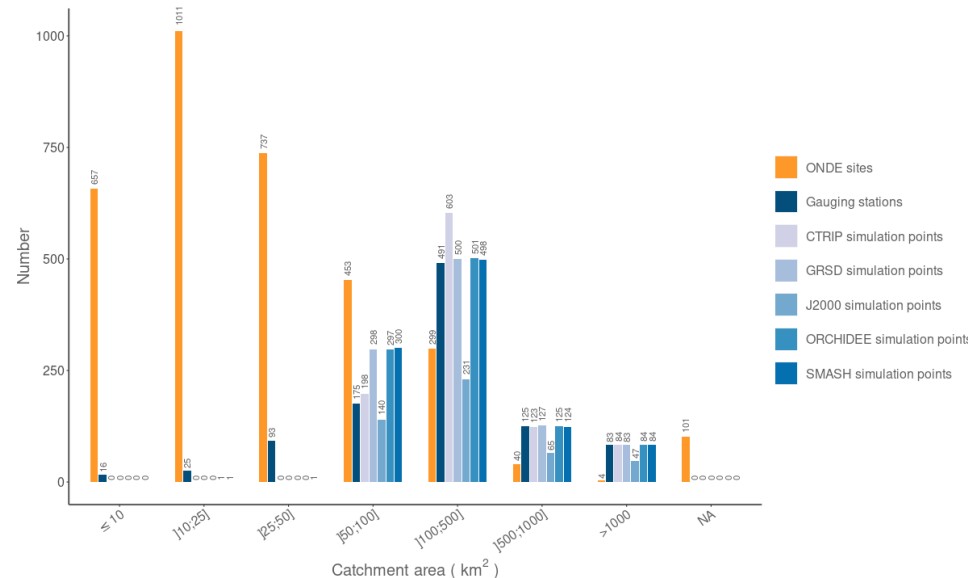

**Figure G1: Frequency distribution of the number of ONDE sites, the 1008 gauging stations selected from the Hydroportail database,**
**and the 1008 simulation points from Explore2 as a function of catchment area**

**NA: Missing values**

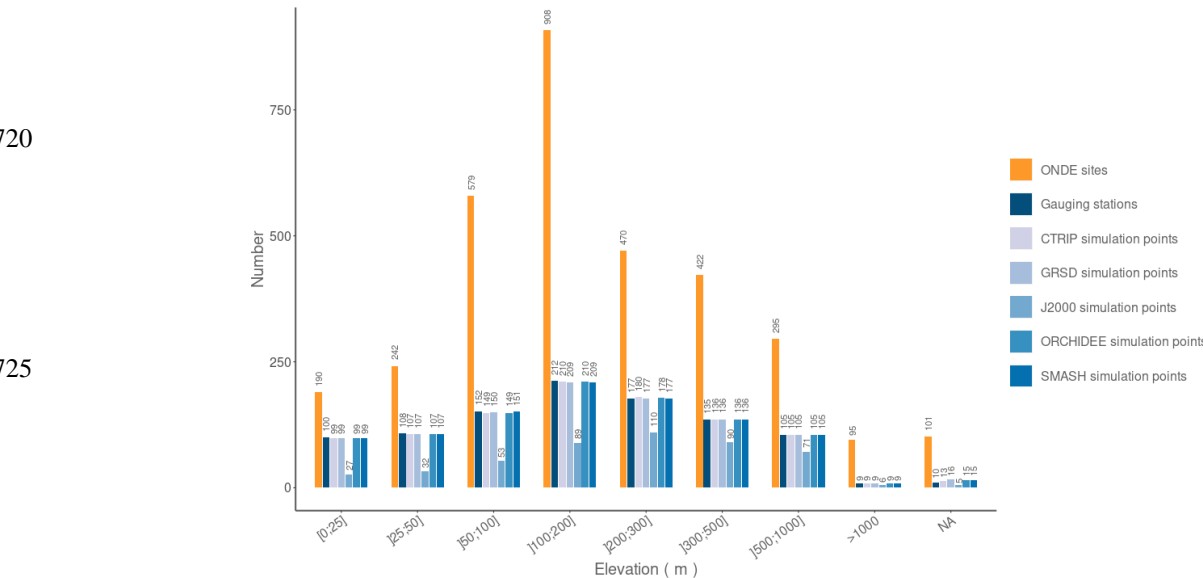

**Figure G2: Frequency distribution of the number of ONDE sites, the 1008 gauging stations selected from the Hydroportail database,**
**and the 1008 simulation points from Explore2 as a function of elevation**

**NA: Missing value**

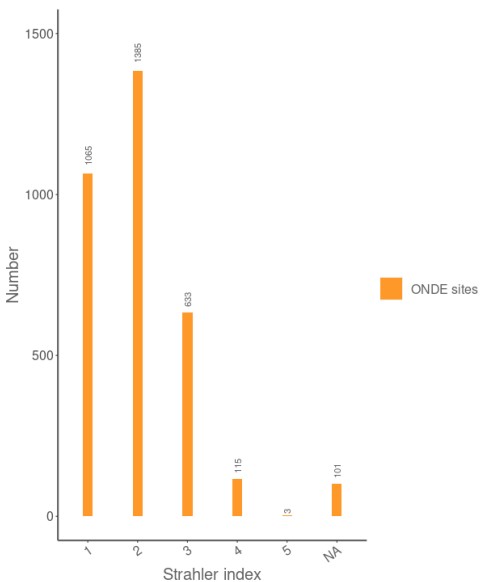

**Figure G3: Strahler index of ONDE sites**

**NA: Missing value**

**Appendix H: Model performance using observed discharge data from gauging stations as predictors**

Over the observation period (2012-2022), logistic regression models estimate the *PFI* values at HER2 scale (Fig. 4). The median observed *PFI* across all HER2 and all campaigns is 14.3% (Q1-Q3: 8.4-20.3) while the logistic regression models yield a median value of 14.4% (Q1-Q3: 8.5-20.4). With results of the leave-one-year-out cross-validation, the models explain 73% of the *PFI* variability according to the NSE (Q1-Q3: 64-84%) (Table 1; Fig. H2), the KGE exceeds 80% in 50 out of 75 HER2 (Fig. H2) and the bias is very low.

The models are also able to describe the inter-annual variability with the alternation of dry and wet years (Fig. H1). The median KGE and NSE scores remain above 0.71 during k-fold validations considering dry, intermediate, and wet years for calibration. RMSE and MAE values do not increase much when the calibration dataset is stratified based on climate conditions, which indicates that the proposed model is quite robust to climate variations under current conditions.

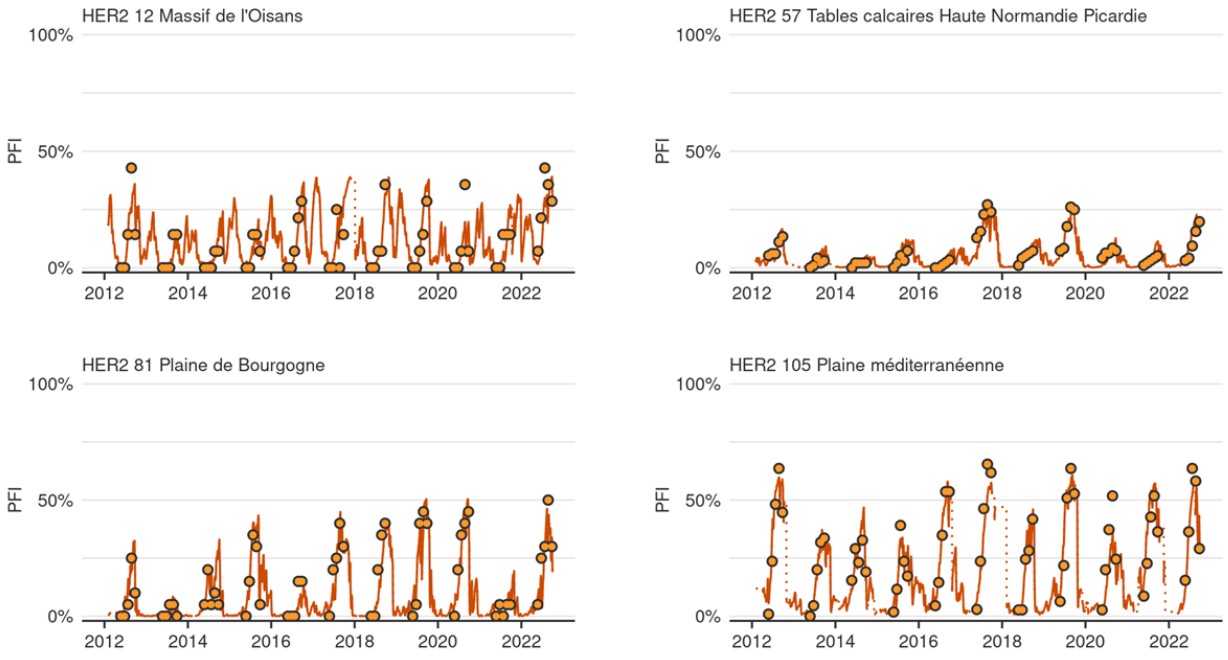

**Figure H1: Time series of *PFI* in HER2 13, 57, 81 and 105. Points and lines respectively represent the *PFI* derived from the ONDE network and the *PFI* estimated by the logistic regression models using discharge data from gauging stations during the calibration period (2012-2022).**

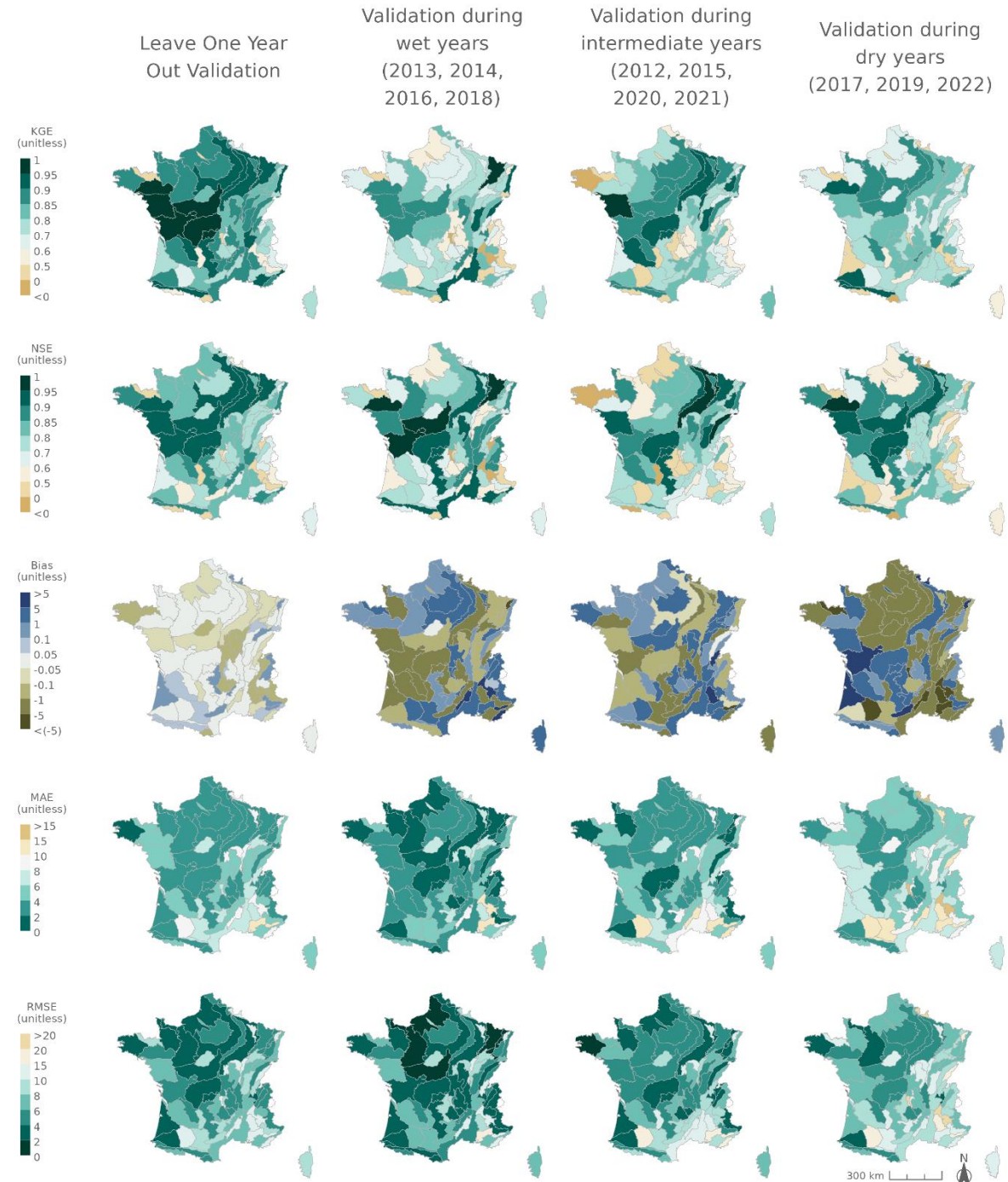

**Figure H2: Kling-Gupta Efficiency (KGE, line 1), Nash–Sutcliffe model efficiency coefficient (NSE, line 2), Bias (line 3), Mean Absolute Error (MAE, line 4), Root-mean-square error (RMSE, line 5) over the calibration period (2012-2022)**

| Validation years | PFI observed on ONDE network (%) | Predicted PFI (%) | Nash Sutcliffe Efficiency (NSE, unitless) | Kling Gupta Efficiency (KGE, unitless) | Bias (unitless) | Mean Absolute Error (MAE, unitless) | Root Mean Square Error (RMSE, unitless) |
|---|---|---|---|---|---|---|---|
| Leave One Year Out (2012-2022) | 14.3 Q1-Q3 8;20 min-max 3;37 | 14.4 Q1-Q3 9;20 min-max 2;37 | 0.79 Q1-Q3 0.6;0.8 min-max 0.2;0.9 | 0.85 Q1-Q3 0.8;0.9, min-max 0.3;1.0 | -0.02 Q1-Q3 -0.08;0.05 min-max -0.68;0.70 | 0.10 Q1-Q3 0.06;0.13 min-max 0.02;0.32 | 0.07 Q1-Q3 0.05;0.09 min-max 0.03;0.15 |
| Wet years (2013, 2014, 2016, 2018) | 8.3 Q1-Q3 5;15 min-max 1;29 | 8.7 Q1-Q3 5;13 min-max 0;35 | 0.76 Q1-Q3 0.6;0.9 min-max -1.7;1.0 | 0.71 Q1-Q3 0.6;0.8 min-max -4.0;1.0 | 0.17 Q1-Q3 -1.23;1.25 min-max -7.09;7.32 | 0.05 Q1-Q3 0.03;0.09 min-max 0.01;0.23 | 0.05 Q1-Q3 0.03;0.08 min-max 0.02;0.15 |
| Intermediate years (2012, 2015, 2020, 2021) | 12.9 Q1-Q3 7;19 min-max 1;35 | 13.6 Q1-Q3 7;19 min-max 2;31 | 0.76 Q1-Q3 0.6;0.8 min-max -0.7;1.0 | 0.76 Q1-Q3 0.7;0.9 min-max -0.2;1.0 | 0.21 Q1-Q3 -1.06;1.41 min-max -8.85;5.76 | 0.07 Q1-Q3 0.04;0.09 min-max 0;0.25 | 0.06 Q1-Q3 0.04;0.09 min-max 0;0.18 |
| Dry years (2017, 2019, 2022) | 21.5 Q1-Q3 14;30 min-max 2;51 | 21.4 Q1-Q3 14;32 min-max 4;49 | 0.7 Q1-Q3 0.5;0.8 min-max -0.1;1.0 | 0.75 Q1-Q3 0.6;0.8 min-max -0.6;0.9 | -0.71 Q1-Q3 -2.90;1.49 min-max -15.13;12.78 | 0.09 Q1-Q3 0.06;0.12 min-max 0.01;0.43 | 0.08 Q1-Q3 0.06;0.12 min-max 0.03;0.24 |


**Table H1: Validation results of drying probability predictions at the HER2 scale using observed flows from the 1008 gauging stations from the HYDRO database. The results correspond to the inter-HER medians, quartiles, minimum and maximum values. The Leave One Year Out analysis results are obtained by averaging the validation metrics computed for each year.**

**Drying probability: Percentage of ONDE sites in a dry state computed for each HER2, averaged by month; Q1-Q3:**

**first and third quartiles; Min: minimum; Max: maximum**

**Appendix I: Model performance using discharge data simulated with SAFRAN**

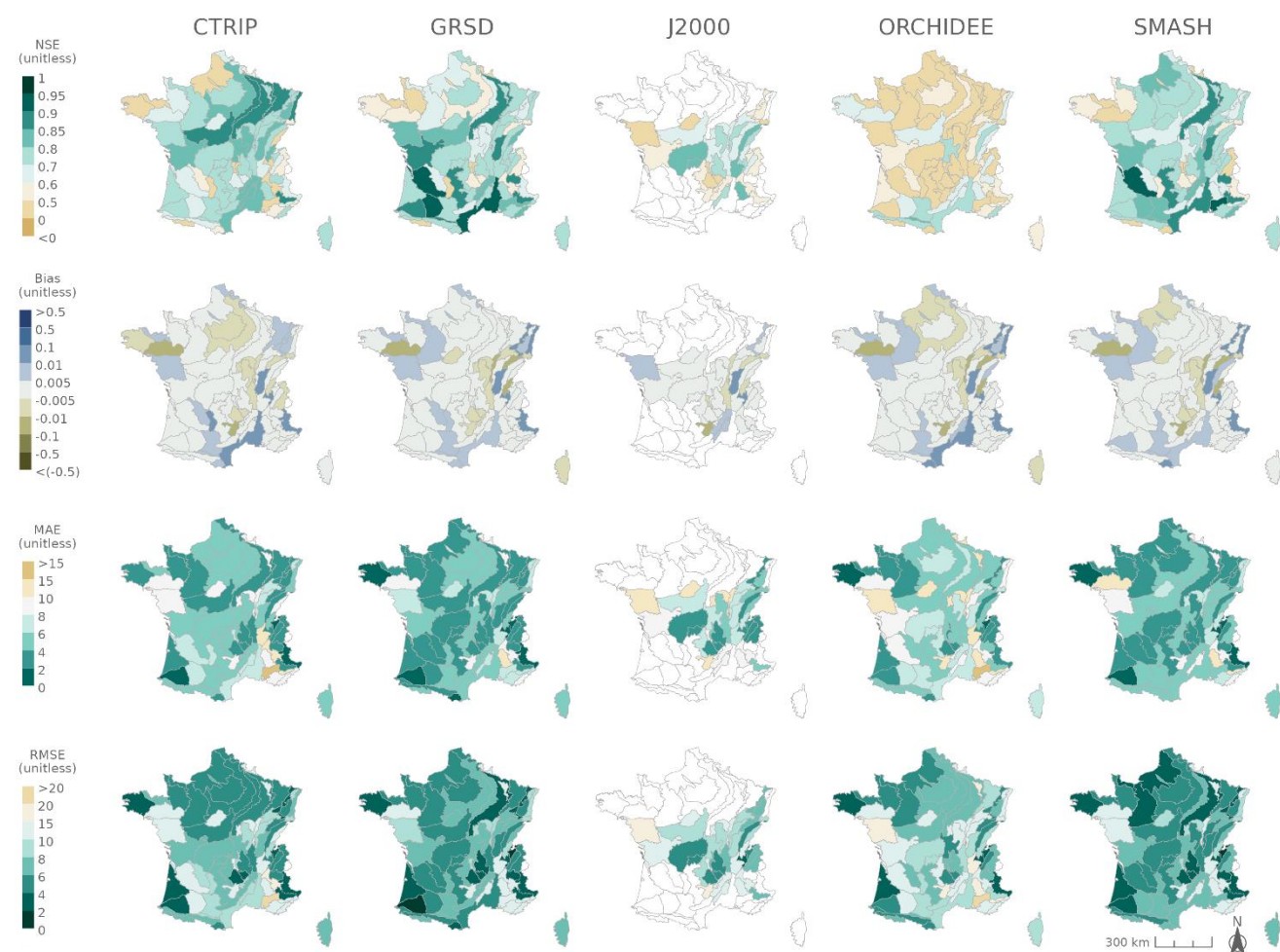

**Figure I1: Nash–Sutcliffe model efficiency coefficient (NSE, line 1), bias (line 2), Mean Absolute Error (MAE, line 3), Root-mean-square error (RMSE, line 4) assessing the calibration of logistic regression using discharge data simulated with SAFRAN data available during the calibration period (2012-2022)**


**Appendix J: Application to 21st century *PFI* modelling**

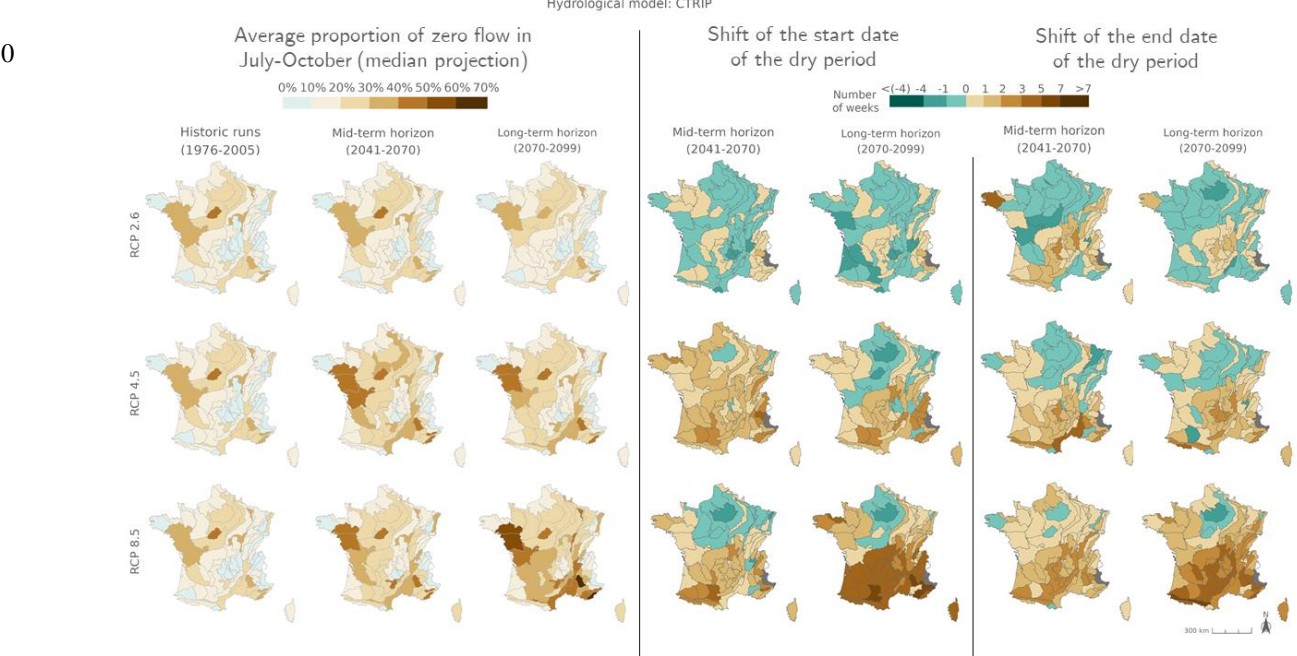

**Figure J1: Ensemble median *mPFI₇-₁₀* (columns 1 to 3), and ensemble median shift of the start date *Tf* (columns 4 and 5) and the end date *Tl* (columns 6 and 7) of the dry period over the two periods 2041-2070 and 2070-2099 under the three RCPs for the CTRIP hydrological model, relative to the baseline period 1976-2005. The shift, expressed in week, takes a positive value when the duration of the dry period increases. Grey HER2s had no period with a *PFI* >20% during the reference period.**

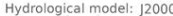

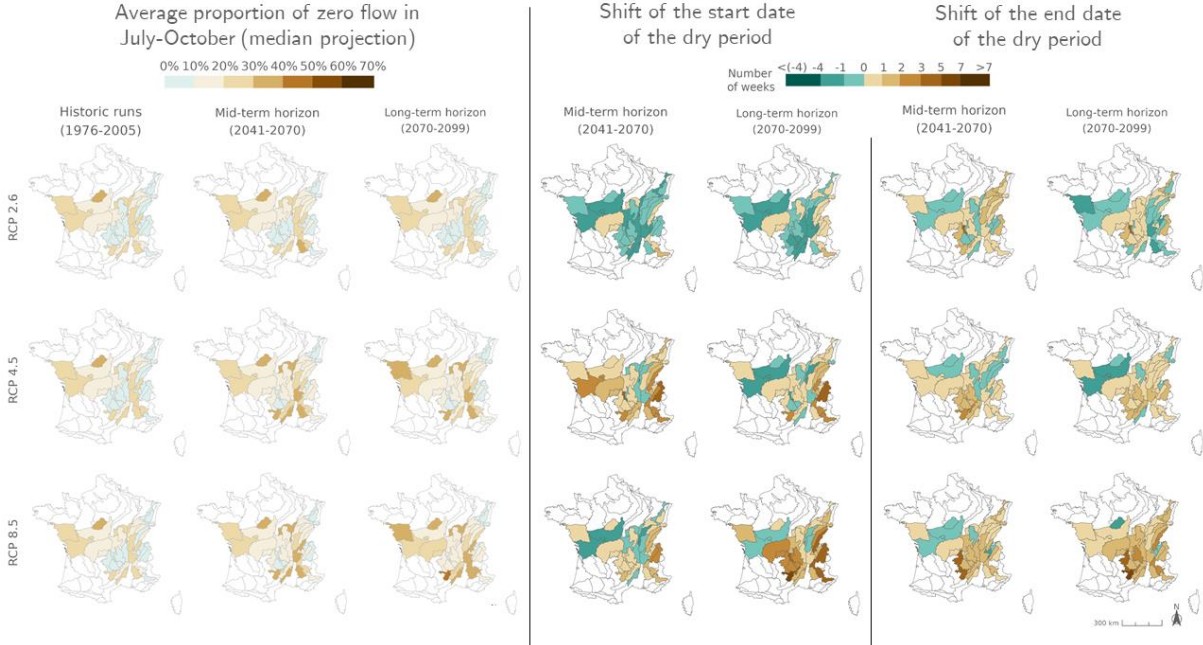


**Figure J2: Ensemble median *mPFI7-10* (columns 1 to 3), and ensemble median shift of the start date *Tf* (columns 4 and 5) and the end date *Tl* (columns 6 and 7) of the dry period over the two periods 2041-2070 and 2070-2099 under the three RCPs for the J2000 hydrological model, relative to the baseline period 1976-2005. The shift, expressed in week, takes a positive value when the duration of the dry period increases. Grey HER2s had no period with a *PFI* >20% during the reference period.**




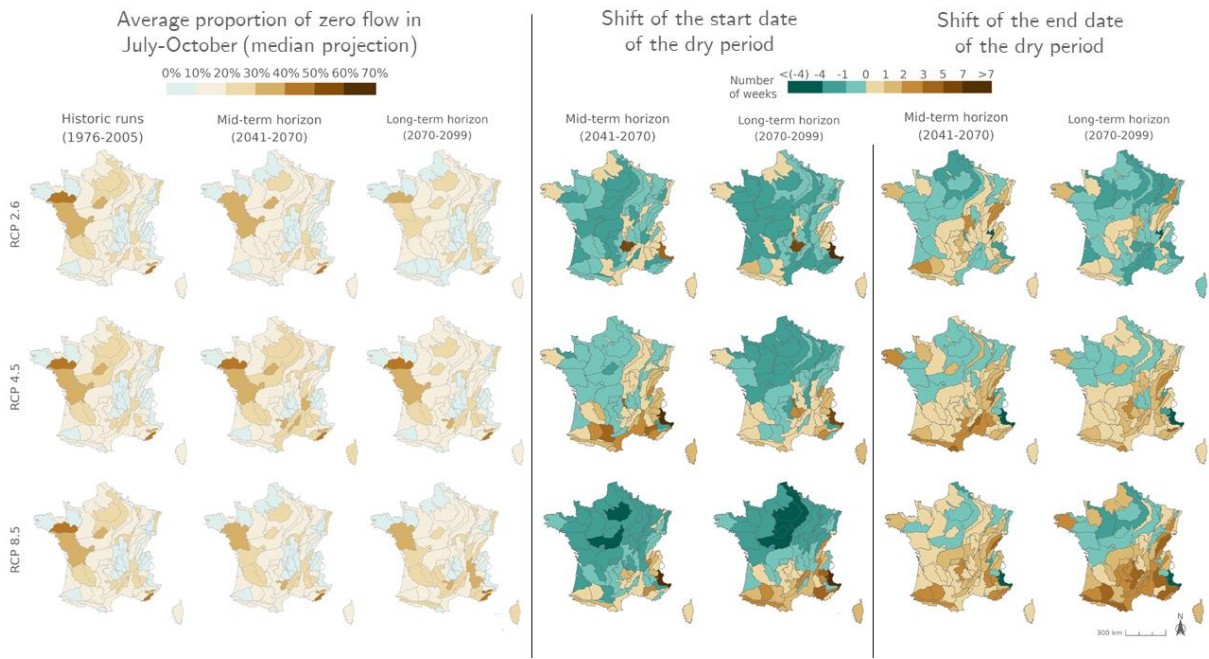

**Figure J3: Ensemble median *mPFI$_{7-10}$* (columns 1 to 3), and ensemble median shift of the start date *Tf* (columns 4 and 5) and the end date *Tl* (columns 6 and 7) of the dry period over the two periods 2041-2070 and 2070-2099 under the three RCPs for the ORCHIDEE hydrological model, relative to the baseline period 1976-2005. The shift, expressed in week, takes a positive value when the duration of the dry period increases. Grey HER2s had no period with a *PFI* >20% during the reference period.**

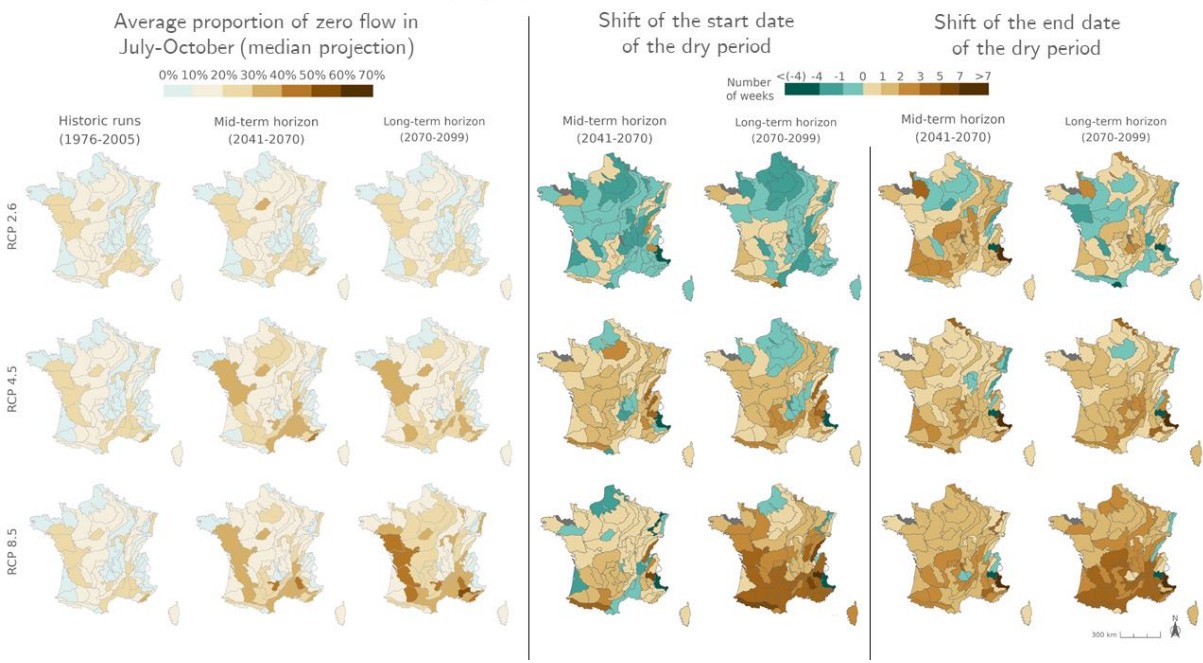

**Figure J4: Ensemble median $mPFI_{7-10}$ (columns 1 to 3), and ensemble median shift of the start date $Tf$ (columns 4 and 5) and the end date $Tl$ (columns 6 and 7) of the dry period over the two periods 2041-2070 and 2070-2099 under the three RCPs for the SMASH hydrological model, relative to the baseline period 1976-2005. The shift, expressed in week, takes a positive value when the duration of the dry period increases. Grey HER2s had no period with a $PFI$ >20% during the reference period.**

**Narratives**

The following figures present time series of the evolution of *PFI* dynamics at different horizons and under different RCP scenarios for each hydrological model in HER2 12, 57, 81 and 105. Four contrasting scenarios were highlighted among the Explore2 climate projections to illustrate a diversity of potential changes under RCP 8.5. These story lines range from "Strong warming and strong summer (and annual) drying", selected as one of the most extreme climate projections, to "Moderate warming and precipitation change" with less pronounced alterations, with two alternative projections named "Dry all year,

reduced winter recharge" and "Hot and humid all seasons". They are also illustrated here to present distinct hydrological nuances.


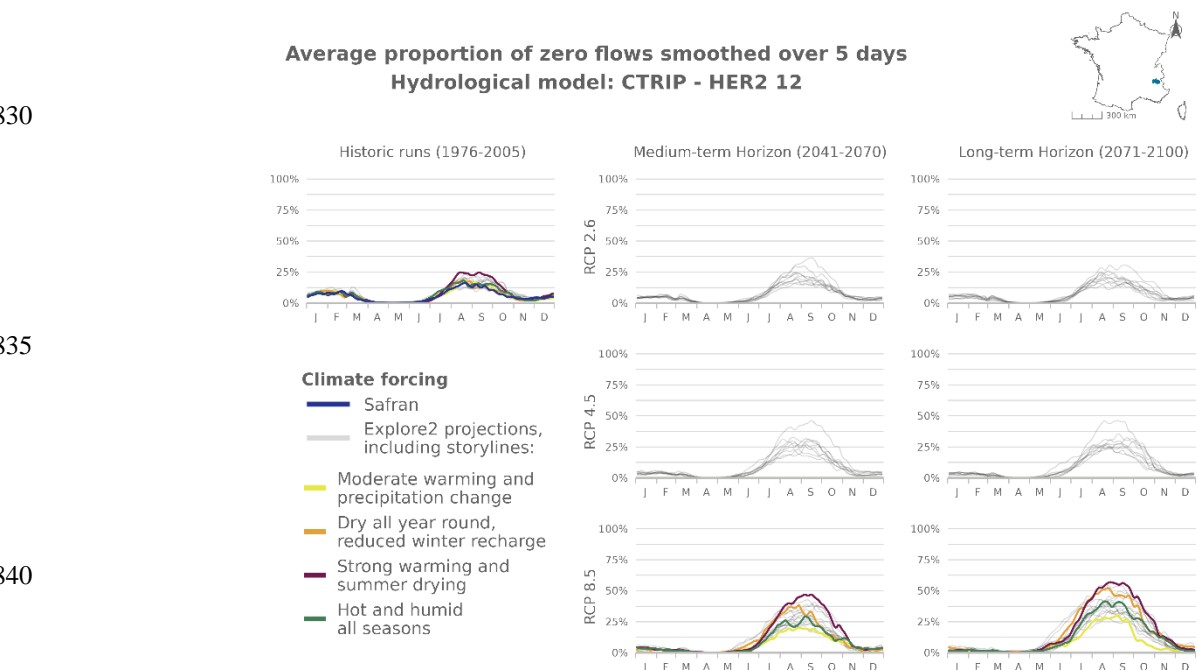



**Figure J5: Time series of the evolution of *PFI* dynamics at different horizons under different RCP scenarios for CTRIP hydrological model in HER2 12**


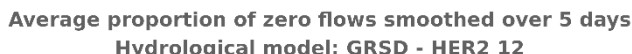
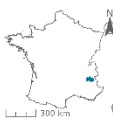

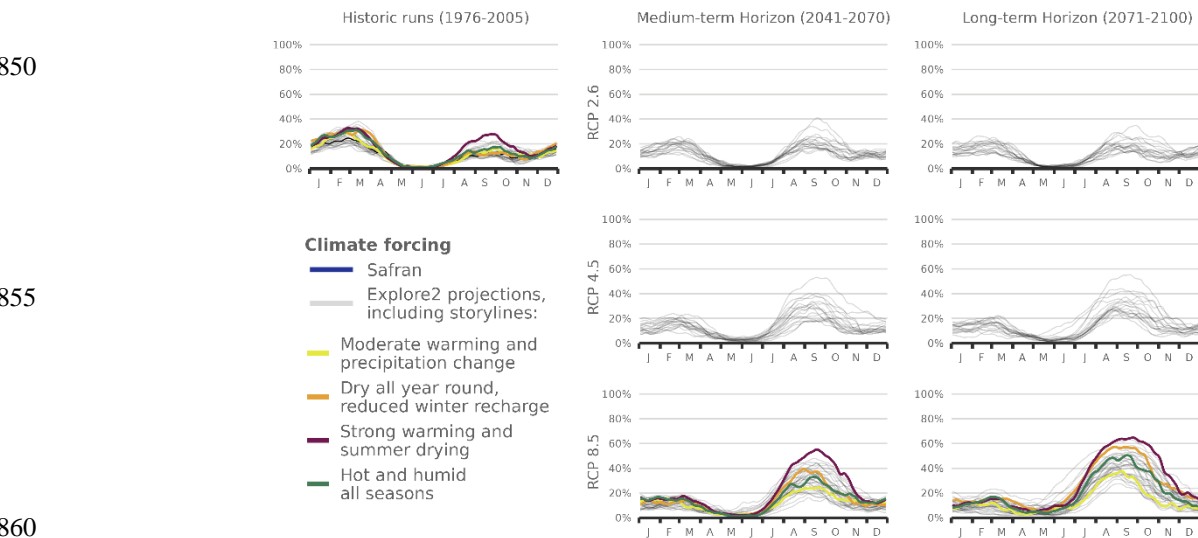

**Figure J6: Time series of the evolution of *PFI* dynamics at different horizons under different RCP scenarios for GRSD hydrological model in HER2 12**

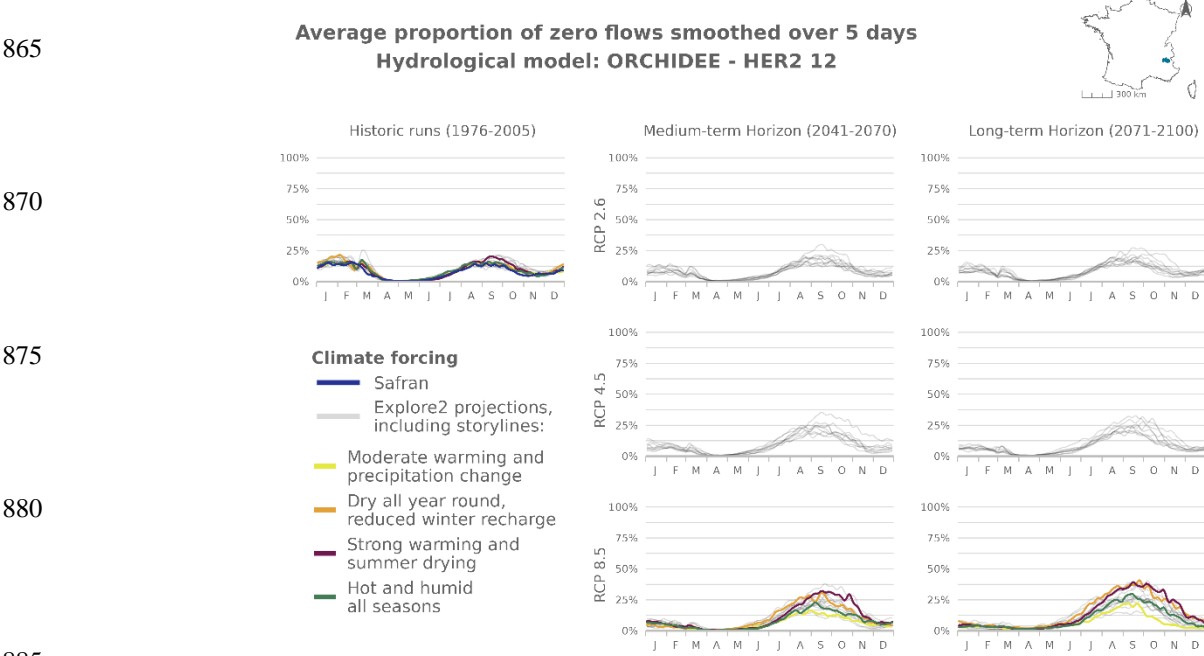

**Figure J7: Time series of the evolution of *PFI* dynamics at different horizons under different RCP scenarios for ORCHIDEE hydrological model in HER2 12**

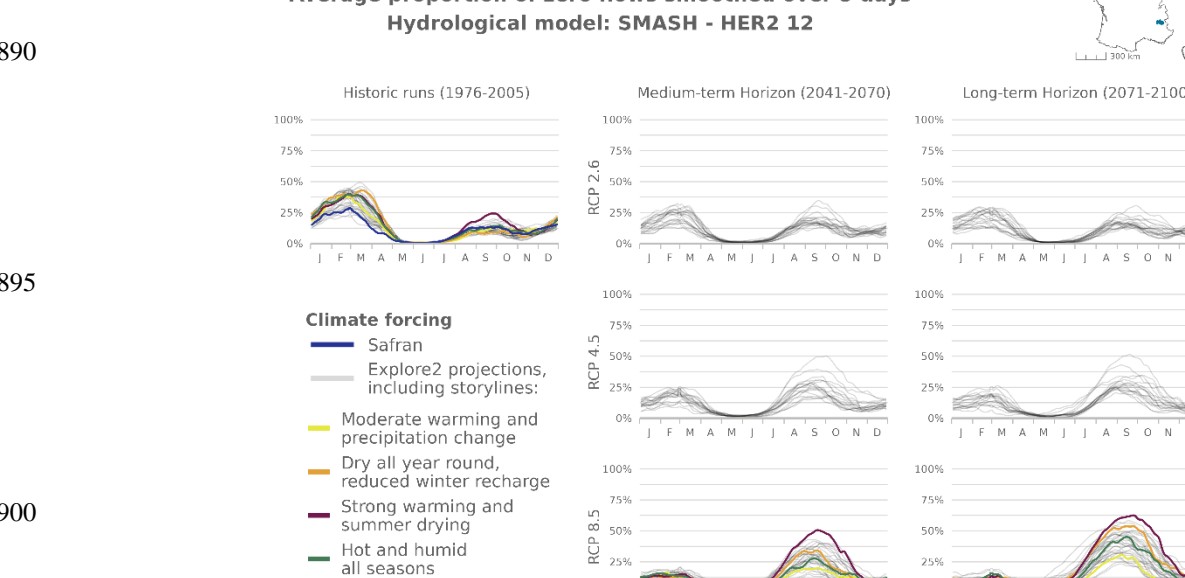

**Figure J8: Time series of the evolution of *PFI* dynamics at different horizons under different RCP scenarios for SMASH hydrological model in HER2 12**

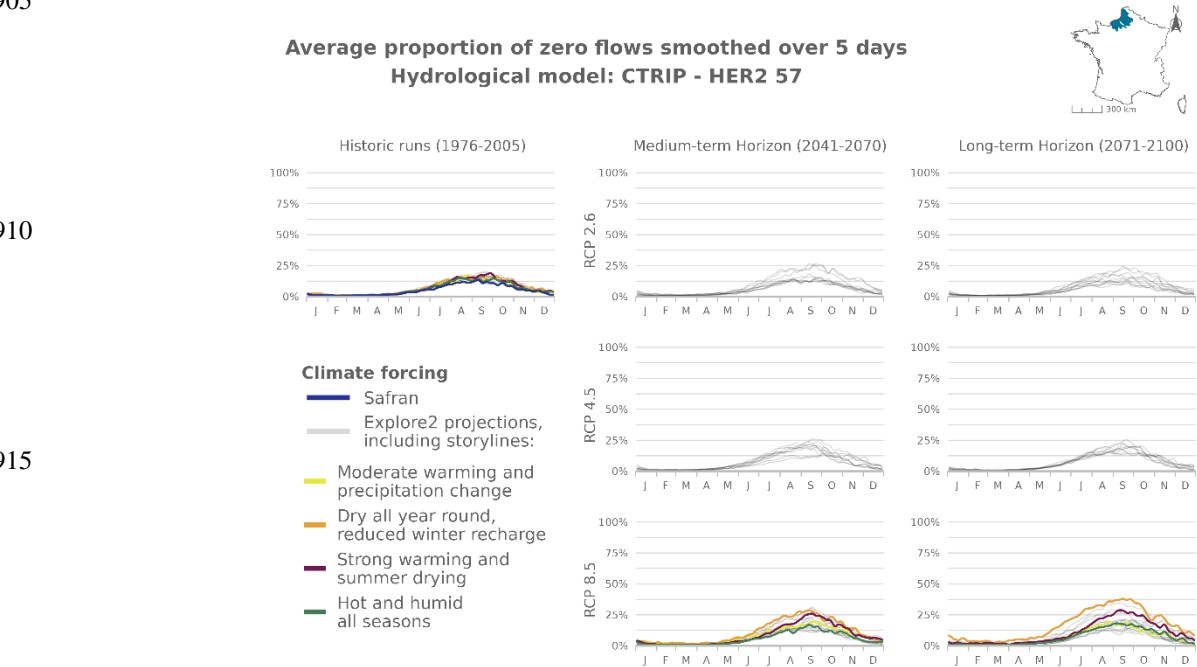

**Figure J9: Time series of the evolution of *PFI* dynamics at different horizons under different RCP scenarios for CTRIP hydrological model in HER2 57**

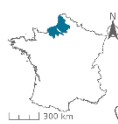

**Average proportion of zero flows smoothed over 5 days**
**Hydrological model: GRSD - HER2 57**

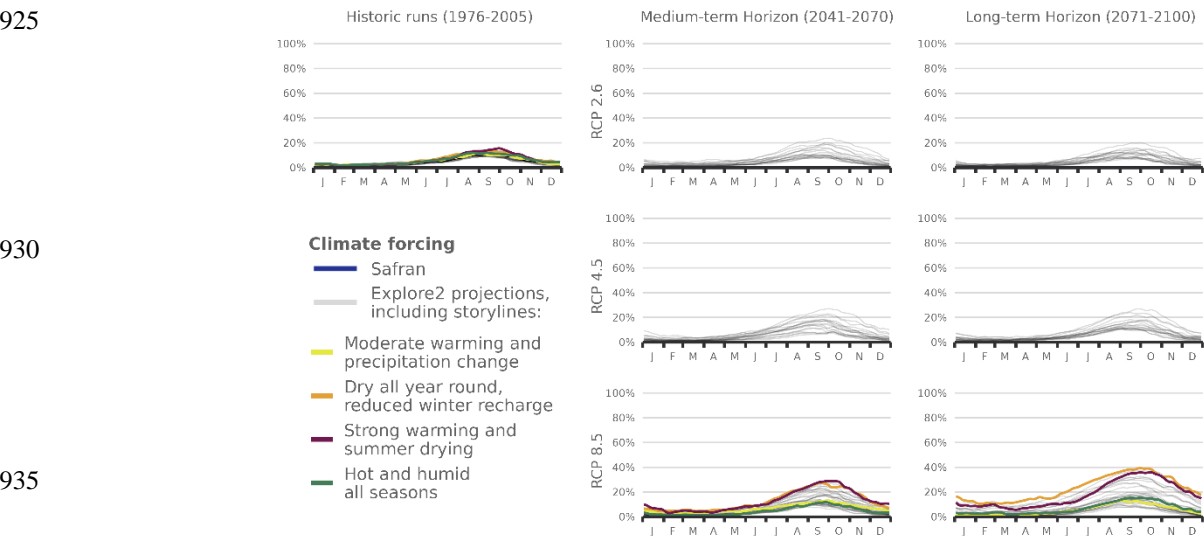

**Figure J10: Time series of the evolution of *PFI* dynamics at different horizons under different RCP scenarios for GRSD hydrological model in HER2 57**

**Average proportion of zero flows smoothed over 5 days**
**Hydrological model: ORCHIDEE - HER2 57**

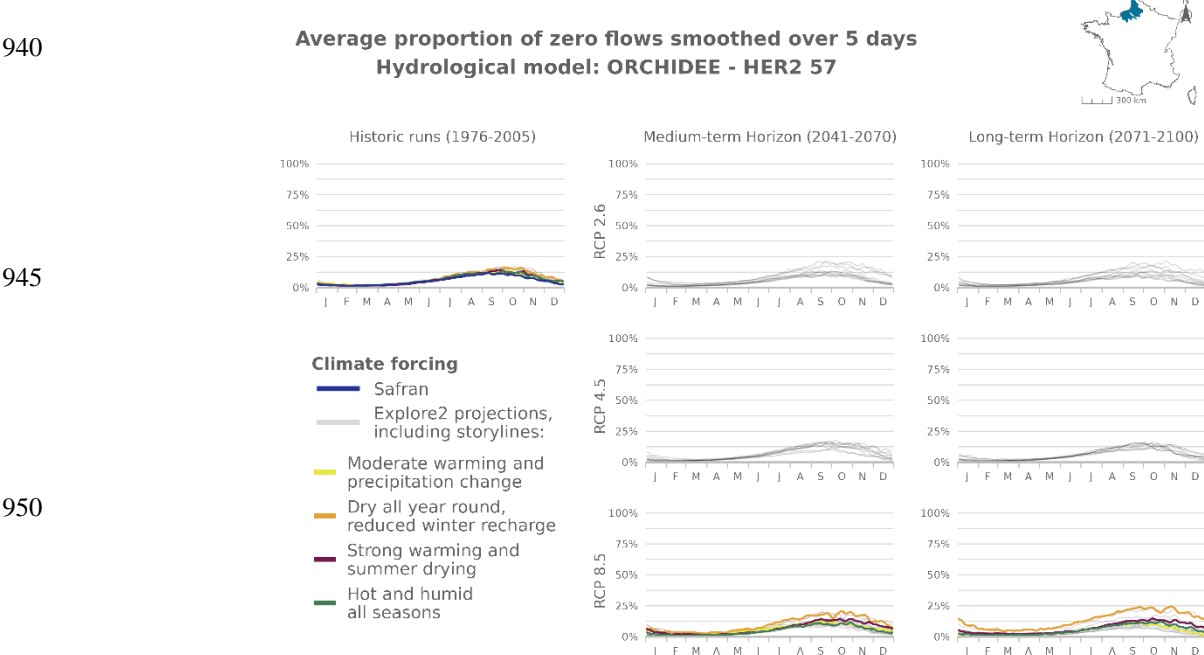

**Figure J11: Time series of the evolution of *PFI* dynamics at different horizons under different RCP scenarios for ORCHIDEE hydrological model in HER2 57**

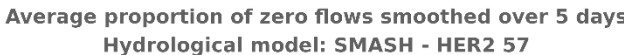
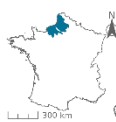

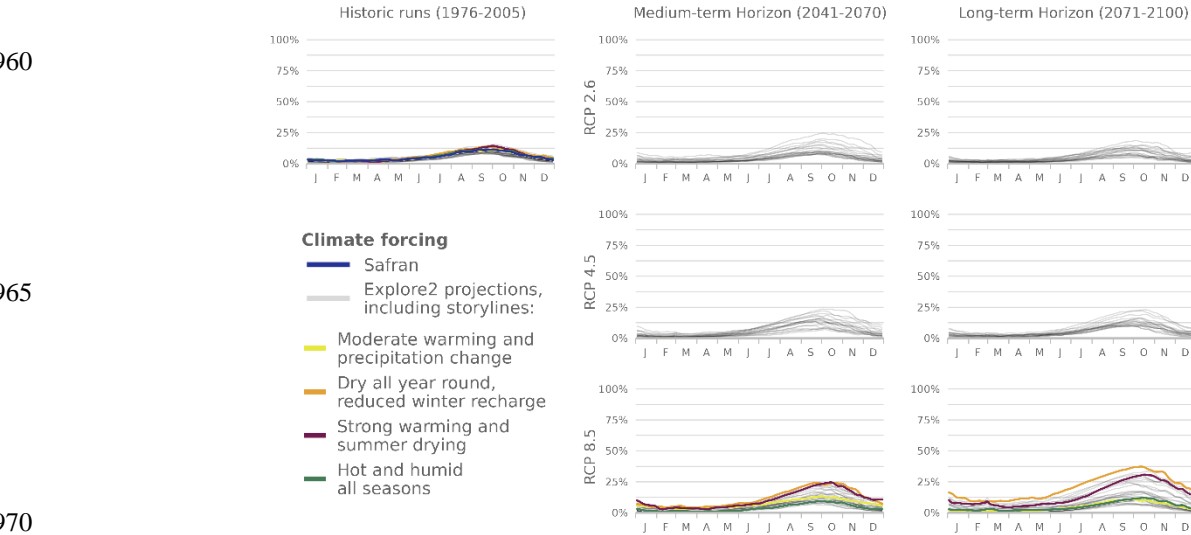

**Figure J12: Time series of the evolution of *PFI* dynamics at different horizons under different RCP scenarios for SMASH hydrological model in HER2 57**

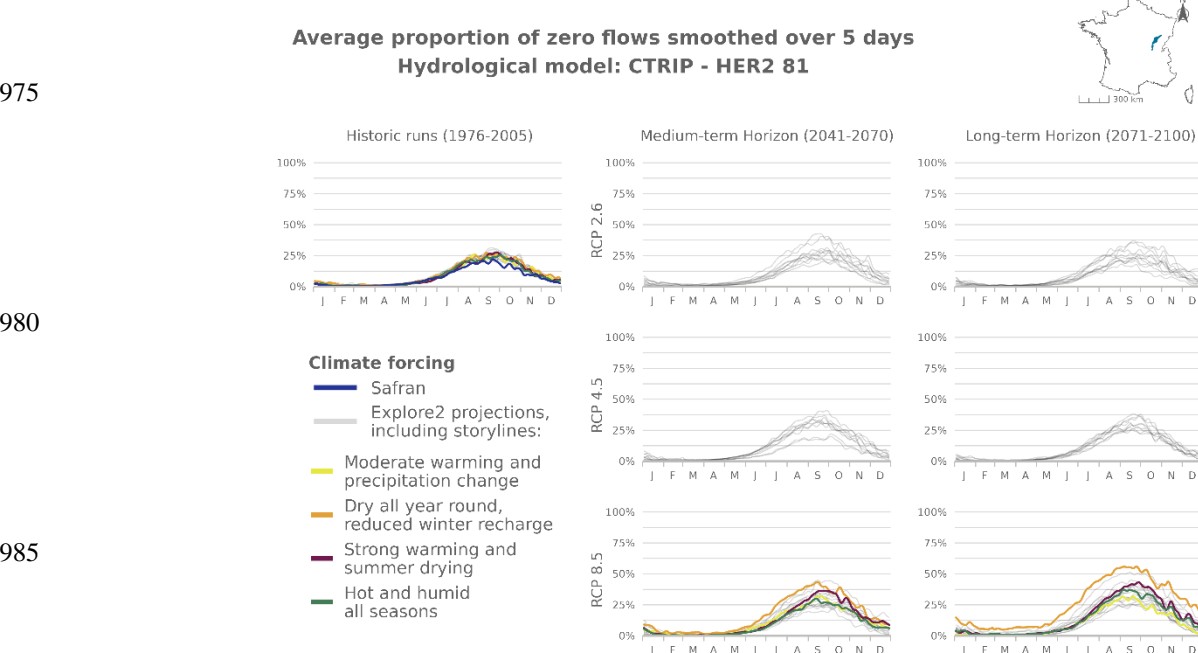

**Figure J13: Time series of the evolution of *PFI* dynamics at different horizons under different RCP scenarios for CTRIP hydrological model in HER2 81**

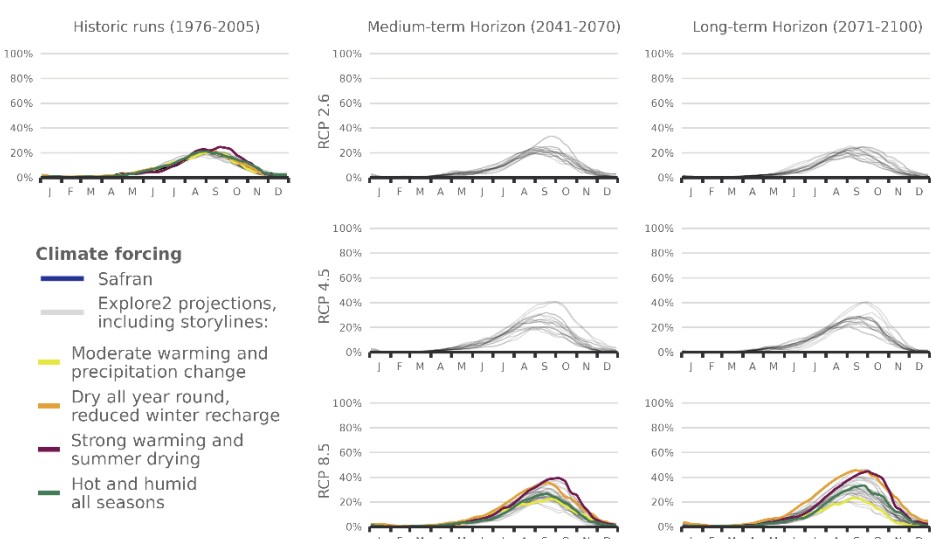

**Figure J14: Time series of the evolution of *PFI* dynamics at different horizons under different RCP scenarios for GRSD hydrological model in HER2 81**

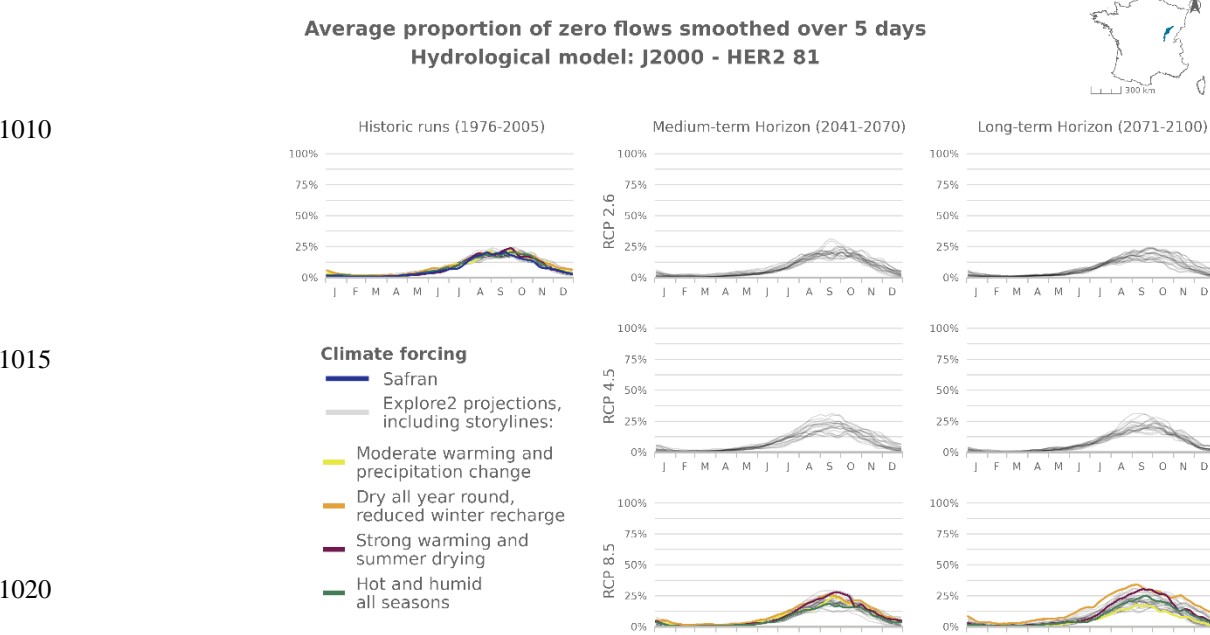

**Figure J15: Time series of the evolution of *PFI* dynamics at different horizons under different RCP scenarios for J2000 hydrological model in HER2 81**

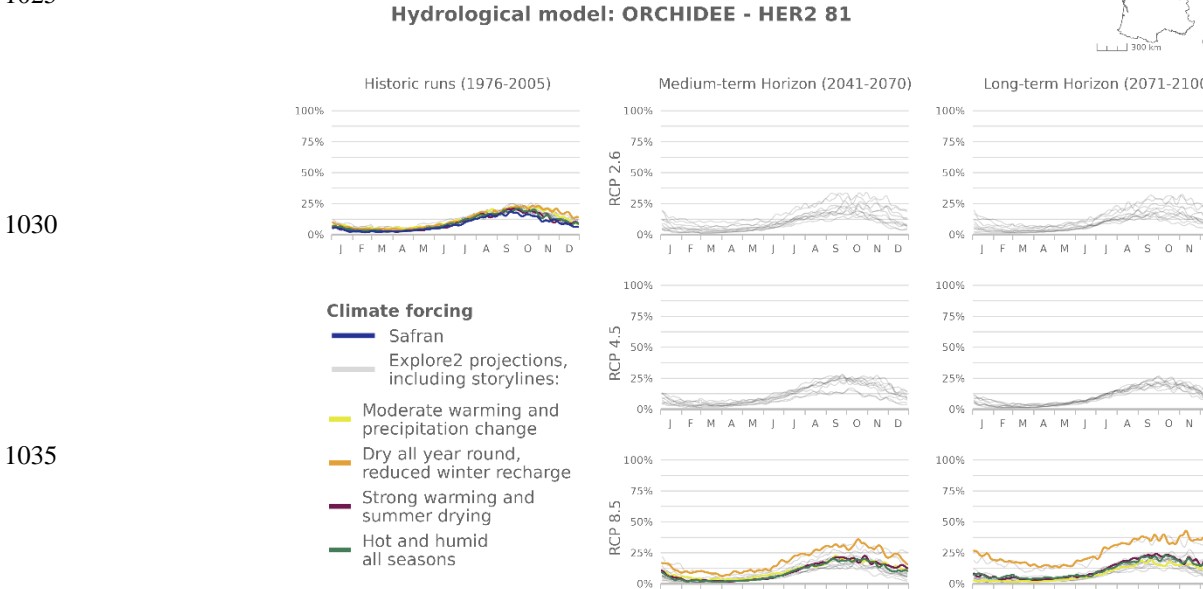

**Figure J16: Time series of the evolution of *PFI* dynamics at different horizons under different RCP scenarios for ORCHIDEE hydrological model in HER2 81**

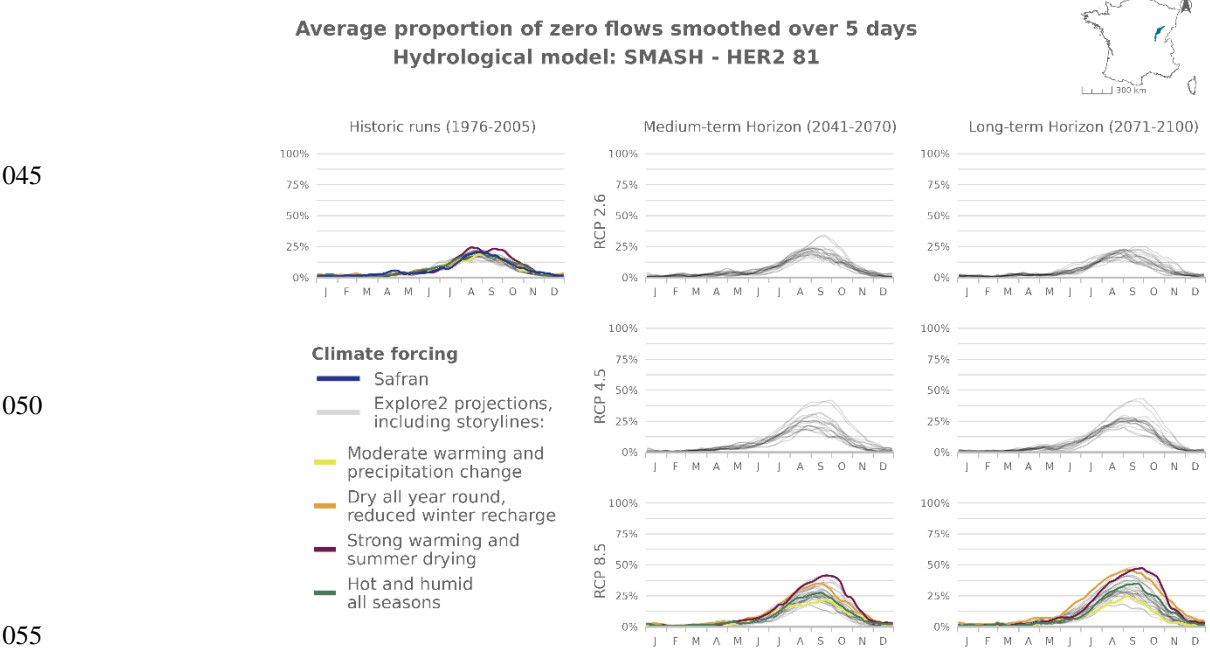

**Figure J17: Time series of the evolution of *PFI* dynamics at different horizons under different RCP scenarios for SMASH hydrological model in HER2 81**

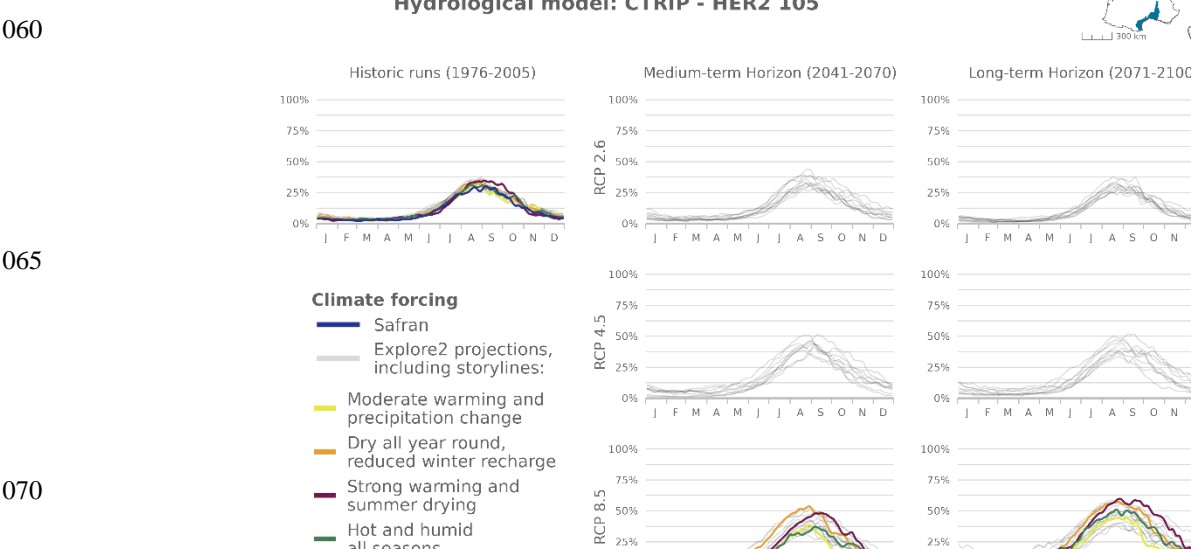

**Figure J18: Time series of the evolution of *PFI* dynamics at different horizons under different RCP scenarios for CTRIP hydrological model in HER2 105**

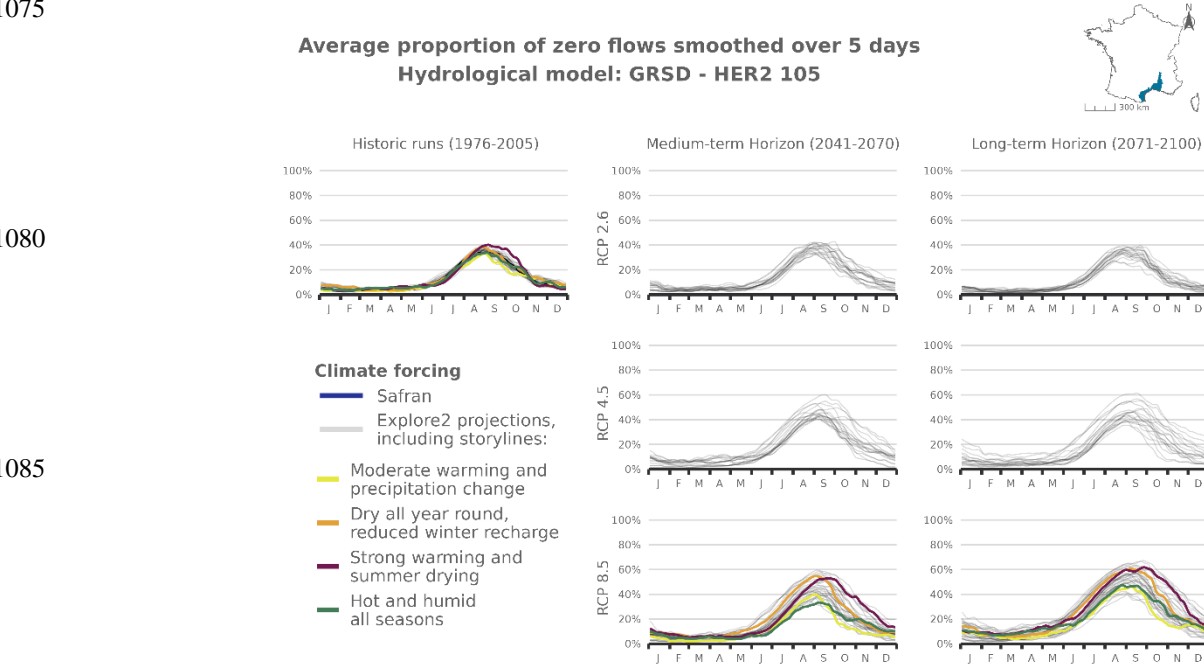

**Figure J19: Time series of the evolution of *PFI* dynamics at different horizons under different RCP scenarios for GRSD hydrological model in HER2 105**

**Average proportion of zero flows smoothed over 5 days**
**Hydrological model: ORCHIDEE - HER2 105**

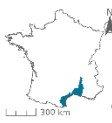

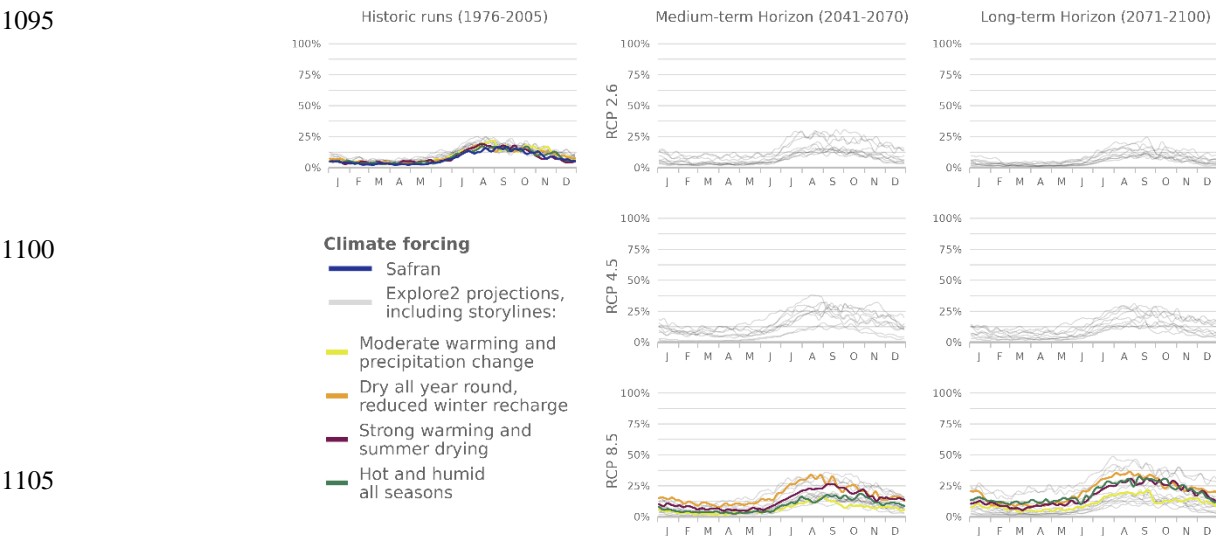

**Figure J20: Time series of the evolution of *PFI* dynamics at different horizons under different RCP scenarios for ORCHIDEE hydrological model in HER2 105**

**Average proportion of zero flows smoothed over 5 days**
**Hydrological model: SMASH - HER2 105**

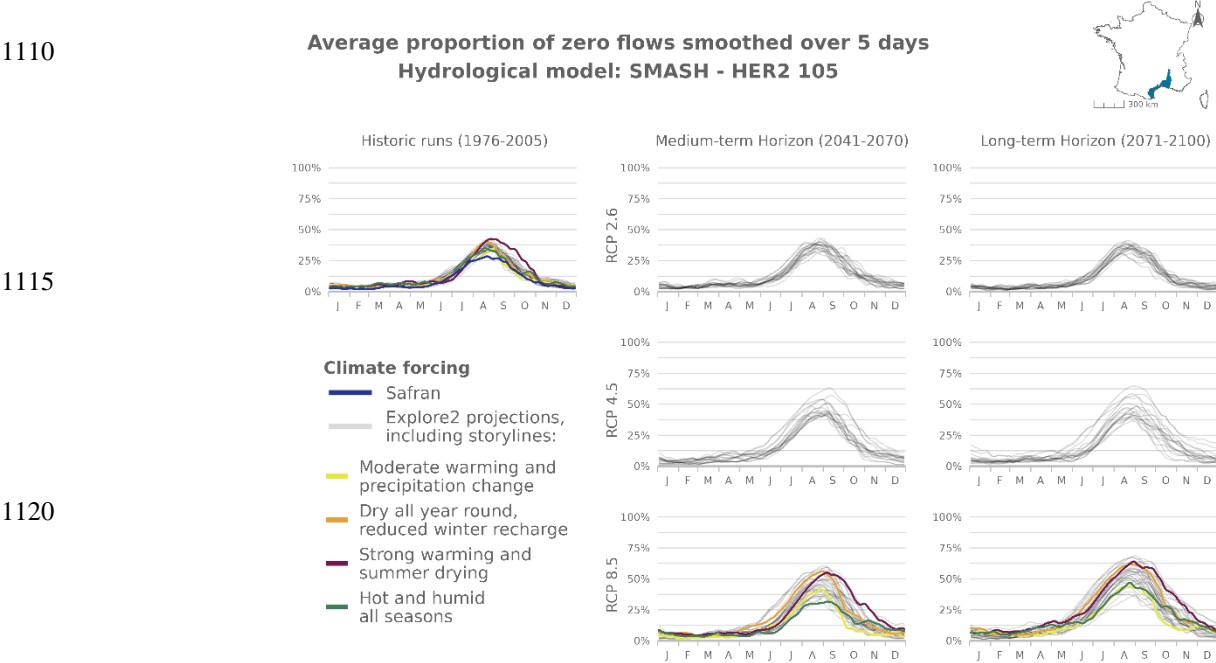

**Figure J21: Time series of the evolution of *PFI* dynamics at different horizons under different RCP scenarios for SMASH hydrological model in HER2 105**

**Appendix K: Agreement between changes under RCP 4.5**

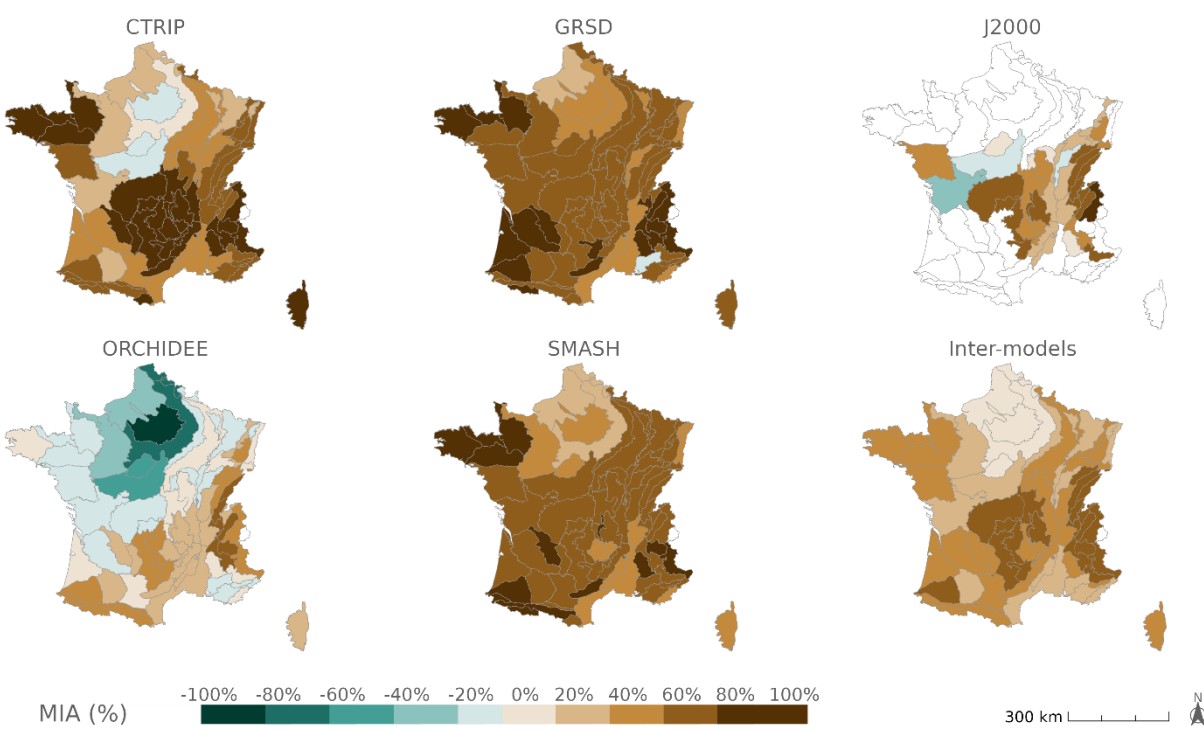

**Figure K1: Agreement between projections of *mPFI₇₋₁₀* for each hydrological model and inter-model agreement on the change signal of *mPFI₇₋₁₀* under the RCP 4.5 scenario.**

**Code availability**

The codes are available at https://github.com/tjaouen/PostDocINRAE

**Data availability**

The Explore2 streamflow projections are available at https://entrepot.recherche.data.gouv.fr/dataverse/explore2 and are described in detail by Sauquet et al. (in prep).

ONDE dataset is available at https://onde.eaufrance.fr/acces-aux-donnees.

.

## Author contribution

TJ, LB and ES designed the experiments and TJ carried them out. LH organized the data from the Explore2 project and made it available. TJ and ES developed the model code. TJ performed the simulations. TJ prepared the manuscript with contributions from all co-authors.

## Competing interests

The authors declare that they have no conflict of interest.

## Acknowledgements

This research was funded by the Rhône-Méditerranée-Corse Water Agency under the supervision of Dr. Benoît Terrier. Scientific oversight was provided within INRAE by Dr. Éric Sauquet and Dr. Lionel Benoît.

This work builds on an initial exploration led by Dr. Aurélien Beaufort and Dr. Éric Sauquet, along with contributions from all participants in the Explore2 project. The archiving and management of Explore2 data, expertly handled by Louis Héraut, also played a crucial role in this study.

Additionally, this research directly applies the QUALYPSO method, developed and adapted to Explore2 by Dr. Guillaume Evin.

We extend our deep gratitude to all these individuals for their invaluable contributions to this work.

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
