# Peer review of "Will rivers become more intermittent in France? Learning from an extended set of hydrological projections"

_EGUsphere, 2024_

## Author Response (AR1)

**RC1**

**GENERAL COMMENTS**

**The manuscript tries to understand how river intermittency across France could change over time as a consequence of climate change. To do so, the authors relate the number of dry locations within a predetermined area (PFI) to the average exceedance probability of the available flowrates within the same area (being them either measured or modeled). Then, a number of climatic and hydrological models are employed to provide future scenarios of streamflow, and from that estimate future scenarios of PFI. Despite the huge uncertainties that these types of works inevitably contain, this method shows how rivers are expected to become more intermittent in the future.**

**Overall, the manuscript is well written and the work seems well executed. However, a number of concerns arise, as reported in the specific comments below, mainly regarding the heterogeneity of the data used as a basis for extrapolating the future scenarios, and the techniques that have (or have not) been used to cope with the data limitations.**

Dear referee,

We thank you for agreeing to conduct this review. We hope that we will respond sufficiently and comprehensively to each of your comments. Before formulating our responses, we would like to express our gratitude for exploring the ONDE dataset and for raising the need to present it more precisely and to clarify how it is used by our method. The elements discussed below will be incorporated into the article. We also thank you for suggesting several references that improve the contextualization of the subject. The introduction has been substantially revised based on these elements.

**SPECIFIC COMMENTS**

**RC1.1 - The work heavily relies on the Explore2 project for future streamflow scenarios, which is explained in a manuscript that is not currently available to the public (Sauquet et al. in prep). While I gave a first review of this manuscript, access to Sauquet et al. (in prep) is a requirement in order to fully understand the soundness of the projections reported here and provide an informed decision on the manuscript.**

AC1.1 - Thank you for requesting access to the referenced document. The cited manuscript, Sauquet et al. (in prep), is a data paper specifically dedicated to presenting the multi-scenario, multi-model projections of the Explore2 project for the 21st century in France. At the time of submission of the present study, the writing of this paper was not finalized. However, it has since been submitted for peer review and will soon be publicly available during the peer-review process.

Once published, its reference will be as follows:

Title: A large transient multi-scenario multi-model ensemble of future streamflows and groundwater projections in France
Author(s): Eric SAUQUET, Guillaume EVIN, Sonia SIAUVE, Ryma AISSAT, Patrick ARNAUD, Maud BEREL, Jeremie BONNEAU, Flora BRANGER, François COLLEONI, Agnès DUCHARNE, Joël GAILHARD, Florence HABETS, Frédéric HENDRICKX, Louis HERAUT, Benoît HINGRAY, Peng HUANG, Tristan JAOUEN, Alexis JEANTET, Sandra LANINI, Matthieu LE LAY, Claire MAGAND, Louise MIMEAU, Céline MONTEIL, Simon MUNIER, Charles PERRIN, Olivier ROBELIN, Fabienne ROUSSET, Jean-Michel SOUBEYROUX, Laurent STROHMENGER, Guillaume THIREL, Flore TOCQUER, Yves TRAMBLAY, Jean-Pierre VERGNES, and Jean-Phillipe VIDAL
MS type: Data description paper

We appreciate your understanding and will update the references in this manuscript as soon as the data paper becomes publicly accessible. We agree that the present study is indeed firmly grounded in the Explore2 framework, which uses robust methodologies and state-of-the-art climate-hydrological modelling tools to obtain reliable projections, while transparently accounting for known uncertainties.

The Explore2 dataset can be accessed with the following digital object identifier https://doi.org/10.57745/JJWOYS. The hydrological data can be downloaded in netCDF file format through the open platform for French public data dedicated to the Explore2 project (https://entrepot.recherche.data.gouv.fr/dataverse/explore2). The Explore2 dataverse is the storage location for the technical documentation of the Explore2 dataset (written in French). Several scientific reports are of particular interest for users of the Explore2 dataset. One of them is an executive summary (Sauquet et al., 2024) with the main conclusions obtained for the two time slices 2041-2070 ('mid-century') and 2070-2099 ('end-of-century') with the two GHG emissions scenarios RCP4.5 and RCP8.5.

Sauquet, E., Evin, G., Siauve, S., Bornancin-Plantier, A., Jacquin, N., Arnaud, P., Bérel, M., Bernus, S., Bonneau, J., Branger, F., Caballero, Y., Colléoni, F., Collet, L., Corre, L., Drouin, A., Ducharne, A., Fournier, M., Gailhard, J., Habets, F., Hendrickx, F., Héraut, L., Hingray, B., Huang, P., Jaouen, T., Jeantet, A., Lanini, S., Le Lay, M., Loudin, S., Magand, C., Marson, P., Mimeau, L., Monteil, C., Munier, S., Perrin, C., Robin, Y., Rousset, F., Soubeyroux, J.-M., Strohmenger, L., Thirel, G., Tocquer, F., Tramblay, Y., Vergnes, J.-P., Vidal, J.-P., and Vrac, M.: Messages et enseignements du projet Explore2, Recherche Data Gouv, https://doi.org/10.57745/J3XIPW, 2024.

**RC1.2 - I am concerned about the biases in the datasets used for this analysis. Only 20% of ONDE stations have drainage area of <10km2, while most of the river network length is in the headwaters (give the power-law scaling of network length with contributing area). As such, there is a strong underrepresentation of headwaters. At the same time, ONDE stations are selected to highlight non-perennial conditions, so non-perennial streams are over-represented in this dataset. Given these biases, how representative is PFIh of the actual river intermittence in France, and its heterogeneity?**

AC1.2 - Thank you for raising these important points regarding the definition of headwaters in the ONDE dataset and their implications for the representativeness of the *PFI* indicators. Indeed, the disparity in drainage areas (with 20% of streams having drainage areas <10 km² and 15% >100 km²), as well as the intentional overrepresentation of intermittent streams, raises valid concerns about the characteristics of the monitored streams and the conclusions that can be drawn from this dataset. We would therefore like to clarify what we exactly mean by headwater streams and *PFI*, and how the observations from the ONDE network can be used to quantify *PFI*.

Headwater stream definition:

The ONDE network, managed by the French Biodiversity Office (OFB), was carefully designed to be "representative of the hydrographic context of French departments". It focuses on streams located "at the head of catchment basins," based on OFB terminology and these streams may be either perennial or intermittent.

This approach prioritizes hydrological characteristics over strict drainage area thresholds, as larger catchments can include upstream tributaries or sparsely developed hydrological networks resembling headwaters. To define headwater streams, the OFB uses the Strahler stream order classification. By this definition, 75% of ONDE sites are located on streams with a Strahler index of 1 or 2, corresponding to streams at the head of the basin, while less than 5% of sites have a Strahler index of 4 or 5.

We acknowledge that OFB headwater definition leads to an underrepresentation of very small drainage areas (<10km$^2$). Additionally, some ONDE sites have drainage areas exceeding 100 km$^2$. To avoid confusion, we propose replacing the term of "headwater streams" with "small streams" where appropriate, which better reflects the characteristics and focus of the ONDE dataset.

*PFI* representativeness:

The ONDE network is one of the oldest, most extensive, and most regularly monitored networks at a national scale for rivers of this type. It ensures a coherent representation of equivalent hydrological functioning all over France, despite inherent spatial distribution biases. Furthermore, the homogenous coverage of France and the systematic observation strategy adopted by ONDE, with monthly surveys conducted between 2012 and 2022, supports the derivation of a robust indicator of hydrological states at a regional scale. The potential overrepresentation of non-perennial streams in the ONDE dataset aligns with its specific objective to monitor intermittence. While this bias should be considered when interpreting *PFI* values, it does not diminish

their utility in capturing spatial and temporal trends. The systematic and regular observations across ONDE sites provide a reliable basis for comparisons and projections over time, including under future climate scenarios.

In summary, to address your comment, we will:

-   Explicitly define headwaters in the manuscript using Strahler orders rather than contributing area,

-   Include a distribution of Strahler orders in Appendix Section G to clarify the representativeness of ONDE stations,

-   Add a short paragraph at the end of section 2.1 to clearly explain what the *PFI* is and why it serves as a proxy (rather than an unbiased estimate) for regional intermittence,

-   Replace "headwater streams" with "small streams" when this terminology better reflects the characteristics of the ONDE sampling strategy and reduces potential ambiguity.

**RC1.3 - On the same note, the work refers multiple times to headwater streams (e.g., in line 231 "the 3302 ONDE sites are located in headwaters"). However, only 20% of the ONDE stations have contributing areas smaller than 10km2, and 15% have areas >100km2. I'm not sure about the author's definition of headwater stream, but I would refer to non-perennial rivers rather than headwater streams.**

AC1.3 - We acknowledge the need for clarification regarding the definition of headwater streams and hope the previous response provides adequate explanation and necessary corrections. While ONDE is one of the most extensive databases for monitoring small rivers at a national scale, we opted to use the entirety of the dataset to develop the *PFI* proxy indicator and track its evolution over time.

**RC1.4 - In the ONDE dataset, not all stations within one Hydro-ecological region are observed in the same day. In fact, some stations have been observed 40 times, others more than 120. The number of observed stations in a given day also ranges from 10 to more than 2000. Is this taken into account when estimating PFIh? As an example, let's say in one region there are 100 ONDE stations. 40 of these stations are observed on day A, and 20 on day B (let's imagine groups A and B do not overlap). If stations observed on day A are generally wetter than stations of day B, the corresponding PFI(A) and PFI(B) are not directly comparable (i.e., the same PFI will be related to significantly different values of FQ) and cannot be modeled with the same logistic regression. Further, the other 40 stations have not been observed in either date. This occurrence can create significant inaccuracies in the regional estimation of PFI and should be taken into account.**

AC1.4 - Your comments highlight the need to clarify an essential aspect of our methodology section: only the systematic campaigns from the ONDE dataset were used, while complementary campaigns were excluded. The revised version of the manuscript will therefore be amended based on the following clarifications:

Systematic campaigns involve singular observations of each site once a month, around the 25th (±5 days). Complementary campaigns, on the other hand, are typically organized in response to specific events, such as drought episodes, and could introduce a bias by over-representing regions or sites experiencing such conditions. By excluding these complementary campaigns, we ensured that the dataset maintained a globally constant number of sites observed each month, thereby allowing for the computation of *PFI* values that are consistent and robust over time.

Given this approach, the observations included in the analysis are discrete in time and are approximated to the mean date of all observations for a given HER around the 25th of each month. While it is true that not all stations within a Hydro-ecological Region (HER) are observed on the same day, the systematic campaigns provide a balanced and consistent sampling scheme that minimizes bias. This approach ensures that the regional *PFI* is representative of the HER as a whole, even when observation dates for individual stations vary slightly. A clarification will be added in Section 2.1 to address this point.

Our study focuses on flow intermittence, a phenomenon that evolves over relatively long time scales, except during periods of significant rainfall. To reflect this temporal scale, our explanatory variable—the non-exceedance curve—is calculated over a one-week window prior to the predicted date. This choice of a seven-day window was validated through sensitivity tests, which showed that the model's predictive performance is largely insensitive to minor adjustments in the window size (e.g., adding or removing a day). This consistency suggests that the sampling strategy is well-suited for predicting flow intermittence, which typically exhibits gradual transitions between flowing and dry states within a year. A clarification will be added in Section 3.1 to address this point.

Regarding the specific concern that varying observation dates and sample sizes could lead to inaccuracies, we recognize the potential for discrepancies if certain stations were systematically wetter or drier than others. However, the systematic sampling ensures that the number of observed sites within each HER remains consistent across campaigns. This consistency reduces the likelihood of significant bias in the regional $PFI$ estimation, as the observed sites are a stable representation of the HER over time. Additionally, while the stations observed on any given day may not cover the entire HER, the averaging process used to calculate $PFI$ smooths out short-term variations, making the predictions robust to the temporal discreteness of observations.

In summary, while there are inherent limitations to any observational, discrete and qualitative dataset, the methodological choices made in our study—relying exclusively on systematic campaigns, using a consistent temporal sampling approach, and focusing on a phenomenon with relatively slow temporal dynamics—were designed to ensure robustness in the estimation of $PFI$. We hope that this explanation of our methodological choices, along with the proposed modifications, addresses the potential inaccuracies raised in your comment and are consistent with the goals of capturing long-term trends in flow intermittence.

**RC1.5 - The entire available discharge time series of each gauging station is used to estimate the flow duration curve. I imagine different gauging stations have highly different time windows of data availability. How does this affect the estimation of FQ,HER2h? Since logistic regression models are fitted on data between 2012 and 2022, you should use the same time window for the flow duration curves.**

AC1.5 - We appreciate this observation and recognize the importance of clarifying how differences in data availability are managed in our approach. While it is true that the entire available daily discharge time series for each gauging station is used to estimate the non-exceedance curve, we acknowledge that the periods of data availability may vary significantly across stations. However, this potential inconsistency is relevant only during the validation phase, based on observed discharge data.

For the simulations conducted within the Explore2 framework (to calibrate the logistic regressions and make $PFI$ projections), the daily discharge time series are standardized. Specifically, all time series span from 1976 to 2005 considering discharge data simulated using the SAFRAN reanalysis data, and from 1976 to 2100 considering discharge data simulated under simulated climate conditions. To address potential issues with extrapolating non-exceedance frequencies for flows beyond the range simulated by SAFRAN, we constructed a synthetic flow time series of 169 years, combining daily discharge data from SAFRAN for the period 1976-2022 with projected discharge data for 1976-2100. This approach ensures consistency in calibration across all simulation points.

Furthermore, limiting the analysis to the period 2012–2022 would likely exclude critical information on hydrological extremes. This short time window does not fully capture the variability of high and low flow events, which are pivotal for understanding a river's behaviour during extreme conditions. Furthermore, the synthetic dataset accounts for potential flow extremes beyond historical records, as simulated under future climate change projections. By incorporating a wider temporal scope, this approach:
- Captures a more comprehensive range of hydrological variability, including historical and potential future extremes.
- Minimizes overfitting to recent data (2012–2022) and ensures the model's robustness for long-term trends.

The manuscript will be updated to account for the above discussion. This remark aligns with comment RC3.6 and highlights the need for greater clarity in our explanation of the derivation of non-exceedance curves. To address this, we propose restructuring the presentation of the methodology by dedicating a specific subsection to the derivation of non-exceedance curves at the gauging stations or simulation points. Following this, we will detail the calibration of logistic regressions and the projection of $PFI$ using the previously calculated non-exceedance curves.

**RC1.6 - Finally, more citations could be added along the text, such as:**
**- Jensen et al, HP 2017, Lapides et al., HP 2021 (flow intermittence results from limited rainfall, freezing conditions, human alterations)**
**- Durighetto et al. RSOS 2022 could be cited in line 29 (climate is a primary driver of streamflow patterns)**
**- Other citations that link non-perennial streams with streamflow, e.g. Shaw et al., JoH 2017, Shaw, HP 2016**
**- Other models that try to estimate streamflow intermittency, e.g. Jaeger et al., JoHX 2019**
**- Other works on the importance of non-perennial streams beyond hydrology, such as Bertassello et al., RSOS 2022, Giezendanner et al., WRR 2021.**

AC1.6 - In response to your remark, as well as to comments RC2.1 and RC3.2 on the introduction of the study, we will provide additional context and background on the literature regarding flow intermittence before elaborating on our research. We appreciate the references you suggested and have also included several additional ones to further enrich our work. A revised version of the introduction is specifically attached at the end of this document.

**TECHNICAL CORRECTIONS**

**RC1.7 - Figure 2, panel "logistic regression": what is the vertical axis representing in this plot?**

AC1.7 - The title of this axis (*PFI*) will be added.

**RC1.8 - Figure 3, line 2: how many of the gauging stations define nested catchments? How are the overlapping areas accounted for? I guess you count the overlapping areas more than once, as ratio also goes above 1. I think a more descriptive ratio should not double count overlapping areas in nested catchments.**

AC1.8 - We thank you for this question, which highlights an unclear aspect of this figure. The title has been revised to clarify the method used to compute the ratio.

**RC1.9 - Figure 3, line 3: the unit of the density is actually km^{-2}.**

AC1.9 - The unit will be updated on the figure.

**RC1.10 - Figure 4, panel "bias": it seems there are no biases > 1 or < -1. I suggest you to reduce the span of the colorbar accordingly, in order to better represent the bias variability on the map.**

AC1.10 - Thank you for pointing this out; the legend of this figure will be updated accordingly.

**RC1.11 - Line 148: the $ sign is visible (I guess it should indicate an equation within the text)**

AC1.11 - We thank you for bringing this formatting error to our attention.

**RC1.12 - Line 201: how are the four HER2 regions used for illustration been selected?**

AC1.12 - The text will specify that this choice was made deliberately to represent contrasting hydro-climatic conditions and behaviours.

**RC1.13 - Line 410: missing verb**

AC1.13 - The sentence will be modified for better clarity.

**RC1.14 - Line 413: how could groundwater levels improve projections? Can you quickly elaborate on what are the expected results if groundwater levels are included in the simulation (with proper citations if available)**

AC1.14 - Thank you for this interesting question. Currently, groundwater levels have only been tested during the calibration process of our method, as referenced in the studies of Beaufort et al. (2018) and Sauquet et al. (2019). While their incorporation did not drastically improve model performance, it provided additional useful information by increasing the density of the monitoring network in regions with few available gauging stations. These aspects will be further elaborated in the corresponding paragraph of the discussion.

In any case, incorporating groundwater levels into the projection process was not possible at the time of this project due to the lack of homogeneously distributed groundwater level projections across France. Yet, we are now working on local-scale projections and considering the integration of groundwater levels, as some data has recently become available through the Explore2 project.

**RC1.15 - Data availability: provide also the link to the ONDE dataset**

AC1.15 - Thank you for this remark, the link will be integrated into the Data availability section:
https://onde.eaufrance.fr/acces-aux-donnees

**RC2**

This paper details modelling and projections of the proportion of dry headwater streams in France at the scale of hydro-ecoregions. The topic is relevant, and the analyses appears to be sound. However, this paper is somewhat hard to comprehend for readers outside the specific research field or unfamiliar with France. I have the following comments to improve the paper:

Dear referee,

We thank you for agreeing to conduct this review. We hope that we will respond sufficiently and comprehensively to each of your comments. Before formulating our responses, we would like to express our gratitude for exploring a perspective outside your specific research. We believe your advice will enhance the understanding of the article, particularly Figure 2, which is central to explain the method and for which we have taken all your suggestions into account.

**MAJOR COMMENTS**

**RC2.1 - The second, third and fourth paragraphs of the introduction delves into details of the datasets and methods used in this study. This content would ideally be in the data and methods sections, and I suggest covering it there. In the introduction, I suggest that the authors provide more context and background about the literature on flow intermittence. Please include insights into the various methods that have been used to study intermittence in various regions of the globe (including outside France), and a summary of the outcomes. I suggest ending the introduction with the clear statement of the research gap this paper is addressing.**

AC2.1 - Thank you for your comment, which echoes feedback from other reviewers. The introduction has been thoroughly revised, with its structure adjusted to better position this study within the broader context of research on flow intermittence under climate change. Additionally, the originality of this study has been clarified. The updated version of the introduction can be found at the end of this document.

**RC2.2 - The reader is referred to (Sauquet et al. in prep). for the hydrological model simulations that underpin the analyses, so it difficult to understand the implications of these hydrological projections on the results presented here. I suggest including the implications of hydrological modelling assumptions also in the discussion.**

AC2.2 - Thank you for requesting access to the referenced document. The cited manuscript, Sauquet et al. (in prep), is a data paper specifically dedicated to presenting the multi-scenario, multi-model projections of the Explore2 project for the 21st century in France. At the time of submission of the present study, the writing of this paper was not finalized. However, it has since been submitted for peer review and will soon be publicly available during the peer-review process.

Once published, its reference will be as follows:

Title: A large transient multi-scenario multi-model ensemble of future streamflows and groundwater projections in France
Author(s): Eric SAUQUET, Guillaume EVIN, Sonia SIAUVE, Ryma AISSAT, Patrick ARNAUD, Maud BEREL, Jeremie BONNEAU, Flora BRANGER, François COLLEONI, Agnès DUCHARNE, Joël GAILHARD, Florence HABETS, Frédéric HENDRICKX, Louis HERAUT, Benoît HINGRAY, Peng HUANG, Tristan JAOUEN, Alexis JEANTET, Sandra LANINI, Matthieu LE LAY, Claire MAGAND, Louise MIMEAU, Céline MONTEIL, Simon MUNIER, Charles PERRIN, Olivier ROBELIN, Fabienne ROUSSET, Jean-Michel SOUBEYROUX, Laurent STROHMENGER, Guillaume THIREL, Flore TOCQUER, Yves TRAMBLAY, Jean-Pierre VERGNES, and Jean-Phillipe VIDAL
MS type: Data description paper

We appreciate your understanding and will update the references in this manuscript as soon as the data paper becomes publicly accessible. We agree that the present study is indeed firmly grounded in the Explore2 framework,

which uses robust methodologies and state-of-the-art climate-hydrological modeling tools to obtain reliable projections, while transparently accounting for known uncertainties. A short description of the models will be given in the revised version of the article.

Note that the Explore2 dataset has been published and can be accessed with the following digital object identifier https://doi.org/10.57745/JJWOYS. The hydrological data can be downloaded in netCDF file format through the open platform for French public data dedicated to the Explore2 project (https://entrepot.recherche.data.gouv.fr/dataverse/explore2). The Explore2 dataverse is the storage location for the technical documentation of the Explore2 dataset (written in French). Several scientific reports are of particular interest for users of the Explore2 dataset. One of them is an executive summary (Sauquet et al., 2024) with the main conclusions obtained for the two time slices 2041-2070 ('mid-century') and 2070-2099 ('end-of-century') with the two GHG emissions scenarios RCP4.5 and RCP8.5.

Sauquet, E., Evin, G., Siauve, S., Bornancin-Plantier, A., Jacquin, N., Arnaud, P., Bérel, M., Bernus, S., Bonneau, J., Branger, F., Caballero, Y., Colléoni, F., Collet, L., Corre, L., Drouin, A., Ducharne, A., Fournier, M., Gailhard, J., Habets, F., Hendrickx, F., Héraut, L., Hingray, B., Huang, P., Jaouen, T., Jeantet, A., Lanini, S., Le Lay, M., Loudin, S., Magand, C., Marson, P., Mimeau, L., Monteil, C., Munier, S., Perrin, C., Robin, Y., Rousset, F., Soubeyroux, J.-M., Strohmenger, L., Thirel, G., Tocquer, F., Tramblay, Y., Vergnes, J.-P., Vidal, J.-P., and Vrac, M.: Messages et enseignements du projet Explore2,  Recherche Data Gouv, https://doi.org/10.57745/J3XIPW, 2024.

**RC2.3 - The last two sentences of the abstract touch on the causes of uncertainties in northern France and the regime changes in mountainous regions. This gives the incorrect impression that these aspects are part of the analyses. While these points are touched on in the discussion, they are not the results of the analyses presented in this study. This study solely focuses on validation, projections, uncertainty partitioning and agreement in PFI, and I suggest the results summary sentences in the abstract to be consistent with that. If the authors wish to convey the additional aspects, it should be clear that those are the implications and not results of the analyses.**

AC2.3 - Thank you for pointing out that the end of our abstract extends beyond the core analyses of our study and gives interpretations derived from our results. We agree that it could mislead readers into thinking that the causes of uncertainties in northern France and the changes in mountainous regions were directly explained by this work. We have revised the abstract to ensure it aligns with the actual results of the study while clearly separating results from implications.

This revised version ensures that:

- The scope of the study is clearly communicated, emphasizing on results such as inter-model variability driving uncertainty in northern France.
- Broader implications, such as shifts in intermittence dynamics in mountainous regions, are appropriately framed as potential outcomes rather than direct findings of the study.

**MINOR COMMENTS**

**RC2.4 - Lines 107-108: "On average, each simulation point has discharge projections simulated by four HM (Interquartile Q1 and Q3 (IQ): 3-5)." It is unclear what the text inside the parenthesis means.**

AC2.4 - The supplemental information in brackets was nonessential and has been removed. The abbreviations for quartiles 1 and 3 (Q1-Q3) will be revised and standardized throughout the paper.

**RC2.5 - Lines 119-120: "In the end, the bias-corrected climate projections were used as inputs for the hydrological models: 10 GCM-RCM projections (resp. 9 and 17) are used.." Again, unclear what the text inside the parenthesis means. Also are there 17 GCM-RCM simulations, or 10 of them?**

AC2.5 - Thank you for this remark. Indeed, the explanation was not clear. All 17 GCM-RCM projections were used for the RCP8.5 scenario, whereas only 10 of these projections were used for the RCP2.6 scenario and 9 for the RCP4.5 scenario. The sentence will be improved in the revised manuscript and will read as: « In the end, the bias-corrected climate projections were used as inputs for the hydrological models: all 17 GCM-RCM projections are used under RCP8.5 scenario, whereas 10 of these projections are used under RCP2.6 and 9 under RCP4.5».

**RC2.6 - Figure 2 and Section 3.1: The explanation of how non-exceedance probabilities (of what flow?) are used as the explanatory variable is unclear. I think the authors mean that the flow duration curves at each gauge are used to obtain the non-exceedance probability of the observed/modelled daily flow for each day during the seven-day window. If so, please clarify this in the text as well as in the schematic. In Figure 2, it can be shown how the probabilities are read from the flow duration curve for an example campaign date which is then used as one of the data points for the logistic regression fit.**

AC2.6 - We thank you for this very helpful remark regarding this central aspect of our method. Your description of the process is indeed correct, and we agree that our original figure lacked clarity and pedagogical value. The non-exceedance curve is more appropriate than the flow duration curve and will replace it in Figure 2, making it easier to interpret. The terminology will be harmonized throughout the article. Following your suggestions, we will incorporate the example of the ONDE campaign of Septembre 25, 2012. We also enhanced the figure's description to provide additional explanations, making the protocol easier to understand.

[Figure]

*Figure 2: Schematic view of the approach adapted to derive the regional Probability of Flow Intermittence (*PFI*) from ONDE sites using daily flows either gauged or modelled within a given HER2. The daily flows simulated by Explore2 at three simulation points generate flow time series (column 1) and their corresponding non-exceedance curves (column 2). A logistic regression is then used to link daily flows to ONDE observations, as*

*illustrated here for the campaign of September 25, 2012. For this campaign, daily discharge values simulated between September 18 and September 25, 2012, are extracted for each simulation point whose drainage area intersects the HER2. These seven flow values are matched to their respective exceedance probabilities using the non-exceedance curve. The mean exceedance probability across all simulation points is then associated with the proportion of ONDE sites exhibiting "dry conditions" during this campaign (observed PFI, column 3). The logistic regression is calibrated using data from all ONDE campaigns. Once calibrated, these models allow for the conversion of daily discharge time series into daily PFI time series for the HER2 (column 4).*

**RC2.7 - Figure 3: It is hard to make out any difference between the columns presented for the gauging stations, and the CTRIP, GRSD, ORCHIDEE, and SMASH models (columns 2, 3, 4, 6 and 7). If the information is the same, consider presenting one set of plots for all five. If they are not the same, consider how the figure can be changed to convey that. If the differences between them is not the main point, maybe you need only one of the columns to be in the main text, and the rest can be in the appendix?**

AC2.7 - Thank you for pointing out the limited readability of this figure. As the differences between maps of CTRIP, GRSD, ORCHIDEE and SMASH are minimal yet distinct, we have followed your recommendation and retained only GRSD in the main text. The maps for the other HM have been moved to Appendix Section E.

**RC2.8 - Use of abbreviations H0, H1, H2: Please consider if it is necessary to name the time periods as it seems simpler to just use the years instead. The paper has quite a lot of abbreviations, and I'm having to go back and check what H0, H1, H2 stands for.**

AC2.8 - Your comment seems relevant to us, and, to improve readability, we now propose a new version where H0 is directly replaced by 1976–2005, H1 by 2041–2070, and H2 by 2070–2099.

**RC2.9 - The QUALYPSO method used to characterise uncertainty in not detailed well. The authors could consider using some illustrative equations to explain how the uncertainty is partitioned, and also explain what the different uncertainties/variabilities mean.**

AC2.9 - We thank you for this remark. To avoid overloading the main text with equations, the uncertainty decomposition method will be summarized in Sect. F of the appendices, using equations that adapt the notations from Evin et al. (2019). This addition will improve the clarity of our approach in characterizing uncertainties.

Furthermore, we have taken particular care to clarify the distinction between uncertainty and variability, as they represent different concepts in this context:

- Uncertainty refers to the lack of confidence in projections due to different sources, including scenario uncertainty (RCPs) and model uncertainty (GCMs, RCMs, and hydrological models).
- Variability, on the other hand, refers to the natural fluctuations in climate and hydrological processes, particularly internal variability, which is an inherent characteristic of the system rather than an uncertainty linked to model choices or future greenhouse gas emissions.

To enhance the accuracy of our terminology throughout the manuscript, we have systematically reviewed and corrected instances where "uncertainty" and "variability" were used, ensuring that each term is employed in accordance with these definitions.

**RC2.10 - Line 289-290: Are some words missing from this sentence?**

AC2.10 - The sentence will be modified as needed.

**RC2.11 - Line 317: "In contrast, under RCP8.5…". Unclear what the contrast is.**

AC2.11 - The connective expression 'In contrast' is not appropriate and will be removed.

**RC2.12 - Lines 355-357: HM also contributes to total variability?**

AC2.12 - We confirm that HMs contribute to the total variability, as illustrated in Figure 9. We acknowledge that we inadvertently failed to mention this and we will correct it in the revised manuscript.

**RC2.13 - Lines 450-452: Citation for the changes in rainfall and flows described.**

Concerning Explore2 projections for the northern part of France, information about the changes is available in the executive summary (Sauquet at al., 2024; in French) and the technical report dedicated to uncertainty (Evin et al., 2024; in French). These citations will be added here.

Evin, G., Hingray, B., Reverdy, A., Ducharne, A., and Sauquet, E.: Ensemble de projections Explore2 : Changements moyens et incertitudes associées, Recherche Data Gouv, https://doi.org/10.57745/KWH320, 2024

Sauquet, E., Evin, G., Siauve, S., Bornancin-Plantier, A., Jacquin, N., Arnaud, P., Bérel, M., Bernus, S., Bonneau, J., Branger, F., Caballero, Y., Colléoni, F., Collet, L., Corre, L., Drouin, A., Ducharne, A., Fournier, M., Gailhard, J., Habets, F., Hendrickx, F., Héraut, L., Hingray, B., Huang, P., Jaouen, T., Jeantet, A., Lanini, S., Le Lay, M., Loudin, S., Magand, C., Marson, P., Mimeau, L., Monteil, C., Munier, S., Perrin, C., Robin, Y., Rousset, F., Soubeyroux, J.-M., Strohmenger, L., Thirel, G., Tocquer, F., Tramblay, Y., Vergnes, J.-P., Vidal, J.-P., and Vrac, M.: Messages et enseignements du projet Explore2, Recherche Data Gouv, https://doi.org/10.57745/J3XIPW, 2024

Concerning the sentence "The annual precipitation by the end of the century remains uncertain, due to the compensatory effect between increased winter recharge and an increased evapotranspiration", we propose to cite Ribes et al. (2022) which provides an updated assessment of past and future warming over France discussing how increased winter precipitation may be counterbalanced by higher evapotranspiration rates due to rising temperatures, particularly in northern France. Additionally, we propose citing Douville et al. (2021), which highlights enhanced precipitation seasonality across Europe.

**RC3**

**General**

   This paper represents a lot of work, and is a valuable contribution to the field of changing intermittence under climate change, particularly as a regional study, and with discussion of uncertainty;

   The style of writing is technical and although well structured, it is not very concise, and would benefit both from more contextual and explanatory text, and consistent terms throughout (e.g. intermittence and intermittency).

Dear referee,

We thank you for agreeing to conduct this review. We hope that we will respond sufficiently and comprehensively to each of your comments. Before formulating our responses, we appreciate your suggestion to improve the contextualization of the subject. The introduction has been substantially modified based on these elements. The consistency of terms will be verified and has already been corrected for "intermittence" and "non-exceedance curve", for instance. You have also raised points that require clarification in the discussion, as well as inaccuracies in the description of the method and the conclusion. We hope the corrections following your advice will enhance the understanding of the paper and improve the presentation of the gap covered by the paper on this topic. We thank you for this.

**Comments**

**RC3.1 - The second part of the title isn't really necessary and CMIP5 isn't mentioned again until the discussion. The paper may have broader appeal without the name of the research project, although it's recognized that acknowledgement is important.**

AC3.1 – It is true that CMIP5 does not need to appear in the title and we agree to remove it. In doing so, we propose to change the title to "Will rivers become more intermittent in France? Learning from an extended set of hydrological projections".

**RC3.2 - The introduction and literature review needs to be better focused on the novelty and content of this study. The broad introductory paragraph is well referenced, but the subject is not then developed, with only passing reference to other regional studies of climate change impacts on intermittence (line 41), and no mention of uncertainty. Specifically:**

**- The literature review on regional studies is held back until the discussion (section 5.2). The justification for the study citing these studies (e.g. Tramblay, Zipper) needs to be moved to the introduction, concluding with the novelty;**

**- The approach to uncertainty needs clarification. If it is a novelty of this paper, this gap in the literature should be covered in the Introduction (but see comment on Section 4.4 below);**

AC3.2 - We thank you for your comment, which aligns with those raised by other reviewers. The introduction will be extensively revised to better position this study within the context of research on the characterization of flow intermittence under climate change. The novelty of this study has also been explained. The revised version of the introduction is attached at the end of this document.

**RC3.3 - Section 3.1. The method used to link intermittence to streamflow would be easier to follow if there were more high-level explanatory sentences e.g. that logistic regressions are derived at regional level using observations of flow state (at ONDE sites) and daily flows (where gauged or modelled). Similarly, the text needs to explain the final step, of deriving daily modelled PFI from mean daily flow exceedances.**

AC3.3 - Thank you for this comment. Following your advice, Section 3.1 will be significantly revised to present the main ideas more clearly. These modifications will complement the corrections suggested by the first reviewer regarding the figure title to better illustrate the protocol. To allow you to assess the relevance of the proposed corrections, if you wish, we have outlined them below:

"The empirical method suggested by Beaufort et al. (2018; Eq. (4)) is applied here to estimate the Probability of Flow Intermittence (*PFI*) at the scale of HER2 regions. It consists in a two-step process that links the observed flow state at ONDE sites to daily streamflows (either gauged or modelled) using logistic regressions.

First, observations of flow states from ONDE sites are used to determine the percentage of sites with "dry conditions" for each HER2 (h) and each campaign date (j). This percentage of dry ONDE sites $PFI_h(j)$ is considered as representative of the regional *PFI*. Subsequently, a logistic regression model is calibrated for each HER2 in order to link the regional *PFI* values for day $j$ to streamflow conditions (Eq. (1), Fig. 2).

$$PFI_h(j) = \frac{e^{\beta_{0 \cdot HER2_h} + \beta_{1 \cdot HER2_h} \times F_{Q \cdot HER2_h}(j)}}{1 + e^{\beta_{0 \cdot HER2_h} + \beta_{1 \cdot HER2_h} \times F_{Q \cdot HER2_h}(j)}},$$
$$(1)$$

where $\beta_{0 \cdot HER2_h}$ and $\beta_{1 \cdot HER2_h}$ are respectively the logistic regression intercept and slope coefficient associated with the predictor $F_{Q \cdot HER2_h}$.

The model uses the mean non-exceedance frequencies of discharge $F_{Q \cdot HER2_h}(j)$ as explanatory variable (Eq. (2)). $F_{Q.HER2h}$ is regarded as a proxy for characterising current wet versus dry hydrological conditions at the regional scale. More specifically, for each specific date ($j$) corresponding to an ONDE field campaign and each HER2 ($h$), the daily empirical non-exceedance frequencies of discharge are spatially averaged over all available $n$ streams whose drainage areas intersect the HER2 of interest (weighted by their drainage area) and temporally averaged over the period $[j-6; j]$. This seven-day window allows the integration of non-simultaneous response times caused by propagation times in the underground and the river networks (the choice for a seven-day window is the result of an optimisation process, see Appendix Sect. B).

$$F_{Q \cdot HER2_h}(j) = \frac{1}{7 \times n} \sum_{k=j-6}^{j} \sum_{s=1}^{n} F_Q(k, s)$$
$$(2)$$

Once calibrated, logistic regressions are used to convert the hydrological projections from the Explore2 project into *PFI* projections."

**RC3.4 - Section 3.4. Consider splitting this into two sections, with the first devoted to recalibration of the PFI logistic regression models using simulated daily discharge data, and the second using these models to simulate PFI into the future.**

AC3.4 - We thank you for this insightful remark, which highlights a lack of clarity in our explanation. Indeed, the cited paragraph focuses solely on the recalibration of the *PFI* logistic regression models using simulated daily discharge data. This recalibration, however, requires calculating non-exceedance curves over sufficiently long time intervals, which explains the construction of flow time series with a period length of 169 years and may have contributed to the confusion.

This remark aligns with comment RC1.5 and underscores that our explanation regarding the derivation of non-exceedance curves is not sufficiently clear. Following your suggestion to restructure the presentation of the method, we propose dedicating a subsection specifically to the derivation of non-exceedance curves at the gauging stations or simulation points. Subsequently, we will address the calibration of logistic regressions and the projection of *PFI* based on the previously calculated non-exceedance curves.

**RC3.5 - Section 4. When referring to Appendices, it would be helpful to have figure panels numbered. Some Appendix numbers appear to need updating (e.g. line 243);**

AC3.5 - Thank you for noticing this oversight. References to the appendices will be clarified for sections containing multiple figures. The appendix numbers will be updated.

**RC3.6 - Table 2: Consider adding min, max and median duration;**

AC3.6 - The requested data (minimum, maximum, and median duration of the dry period) will be added to Table 2.

**RC3.7 - Section 4.4. I was expecting something on the contribution of the logistic regressions to the uncertainty. This could be clarified by changing the subheading.**

AC3.7 - We thank you for pointing out this confusion. Unfortunately, our method does not allow for redundancy in the logistic regressions, as there is one deterministic regression per combination of RCP, GCM, RCM, and hydrological model. The title of Section 4.4.1 will be revised to clearly reflect that the uncertainty analysis focuses exclusively on the uncertainties related to the climatic and hydrological modeling components and how these propagate to the *PFI* projections. The revised title will read: *Uncertainties in hydro-climatic projections and their impact on PFI projections.*

**RC3.8 - Section 5.3. The results indicate overestimation of PFI in the southwest, and underestimation in the northwest, north and southeastern parts, especially pronounced in dry years (lines 283-285). The discussion of northern France needs to refer to these results for context.**

AC3.8 - Thank you for your insightful comment regarding the discussion of uncertainties in the northern part of France. We will carefully revise the paragraph to address your concern. Specifically, we will refine the language to acknowledge and account for the bias of our model in this region during dry years, before addressing the uncertainty of climate projections in this region, which we believe remains the primary driver of uncertainties in this context.

Additionally, we will support the statement, "The annual precipitation by the end of the century remains uncertain, due to the compensatory effect between increased winter recharge and an increased evapotranspiration," by citing Ribes et al. (2022), which offers an updated assessment of past and future warming over France, emphasizing how increased winter precipitation may be counterbalanced by heightened evapotranspiration rates due to rising temperatures, particularly in northern France. We will also reference Douville et al. (2021), which highlights enhanced precipitation seasonality over Europe.

The revised paragraph will read as follows:

In the northern part of France, discrepancies between GCM-RCM-Hydrological Models projections result in higher uncertainties in *PFI* projections and pronounced geographical contrasts on the multi-model-based *MIA* map (Fig. 10). Part of these uncertainties can be attributed to the observed underestimation of *PFI* in this region, particularly during dry years (Sect. 4.2, Appendix Fig. H2 and Table H1). However, the primary source of these uncertainties likely stems from uncertainties in future rainfall patterns in this region where the majority of Explore2 projections indicate an increase in winter rainfall and winter mean flows (for 8 out of 9 hydrological models) and a decrease in summer precipitation. The annual precipitation by the end of the century remains uncertain, due to the compensatory effect between increased winter recharge and an increased evapotranspiration (Ribes et al., 2022; Douville et al., 2021).

**RC3.9 - Figure 7. The headings do not match the captions. It is difficult to follow what is being shown when inconsistent terms are used.**

AC3.9 - Our sincere apologies for the confusion. You are correct that the expression "change in $mPFI_{7\text{-}10}$" was inappropriate to describe columns 1 to 3, as each map individually represents the median $mPFI_{7\text{-}10}$, rather than its evolution. The figure should be clearer once this correction is taken into account.

**RC3.10 - Consider replacing the first two sentences, which set a rather negative tone, with a more positive statement of the novelty of the study e.g. for the first time.**

AC3.10 - We will modify the conclusion to end on a more positive note in line with the study carried out:

"This study assesses the changes in the intermittency of river flows across France in the context of climate change. For the first time, multi-model and multi-scenario hydro-climatic projections are used as a predictor to explore the possible evolution of the daily probability of flow intermittency at the scale of (level 2) Hydro-EcoRegions. Leveraging monthly monitoring of small streams over ten summers, we calibrated logistic regressions to transform hydrological projections of large watersheds into regional proportions of flow intermittence for the 21$^{st}$ century."

**Minor comments**

**RC3.11 - There is no need to abbreviate hydrological models to HM, which is not used consistently (e.g. lines 77, 182 and 383), and should not in any case be abbreviated in the Conclusion;**

AC3.11 - All instances of the abbreviation "HM" will be replaced with the full expression "hydrological model(s)". No abbreviations will be used in the conclusion.

**RC3.12 - There are some grammatical and spelling errors and proof-reading is recommended (e.g. in the abstract, lines 10 and 12, and caption of Figure 4);**

AC3.12 - Please accept our apologies for these significant grammatical and spelling errors, which likely left an unfavourable impression from the very first lines of the document. We will carefully review the manuscript to ensure that no errors remain in the revised version.

**RC3.13 - Sections 2.1.3 (line 177) and 2.1.4 (line 179) could not be found.**

AC3.13 - Thank you for this observation. All the elements listed here will be modified and all references to sections will be verified.

**Revised version of the Introduction**

[revised manuscript text omitted]